# Towards the Generalization of Contrastive Self-Supervised Learning

**Weiran Huang**[1*]   **Mingyang Yi**[2*]   **Xuyang Zhao**[3*]   **Zihao Jiang**[1]

[1] Qing Yuan Research Institute, Shanghai Jiao Tong University
[2] Huawei Noah's Ark Lab
[3] School of Mathematical Sciences, Peking University

## Abstract

Recently, self-supervised learning has attracted great attention, since it only requires unlabeled data for model training. Contrastive learning is one popular method for self-supervised learning and has achieved promising empirical performance. However, the theoretical understanding of its generalization ability is still limited. To this end, we define a kind of $(\sigma, \delta)$-measure to mathematically quantify the data augmentation, and then provide an upper bound of the downstream classification error rate based on the measure. It reveals that the generalization ability of contrastive self-supervised learning is related to three key factors: *alignment* of positive samples, *divergence* of class centers, and *concentration* of augmented data. The first two factors are properties of learned representations, while the third one is determined by pre-defined data augmentation. We further investigate two canonical contrastive losses, InfoNCE and cross-correlation, to show how they provably achieve the first two factors. Moreover, we conduct experiments to study the third factor, and observe a strong correlation between downstream performance and the concentration of augmented data.

## 1 Introduction

Contrastive Self-Supervised Learning (SSL) has attracted great attention for its fantastic data efficiency and generalization ability in computer vision (He et al., 2020; Chen et al., 2020a;b; Grill et al., 2020; Chen & He, 2021; Zbontar et al., 2021) and natural language processing (Fang et al., 2020; Wu et al., 2020; Giorgi et al., 2020; Gao et al., 2021; Yan et al., 2021). It learns the representation through a large number of unlabeled data and manually designed supervision signals (i.e., regarding the augmented views of a data sample as positive samples). The model is updated by encouraging the features of positive samples close to each other. To overcome the feature collapse issue, various losses (e.g., InfoNCE (Chen et al., 2020a; He et al., 2020) and cross-correlation (Zbontar et al., 2021)) and training strategies (e.g., stop gradient (Grill et al., 2020; Chen & He, 2021)) are proposed.

In spite of the empirical success of contrastive SSL in terms of their generalization ability on downstream tasks, the theoretical understanding is still limited. Arora et al. (2019) propose a theoretical framework to show the provable downstream performance of contrastive SSL based on the InfoNCE loss. However, their results rely on the assumption that positive samples are drawn from the same latent class, instead of the augmented views of a data point as in practice. Wang & Isola (2020) propose alignment and uniformity to explain the downstream performance, but they are empirical indicators and lack of theoretical generalization guarantees. Both of the above works avoid characterizing the important role of data augmentation, which is the key to the success of contrastive SSL, since the only human knowledge is injected via data augmentation. Recently, HaoChen et al. (2021) propose to model the augmented data as a graph and study contrastive SSL from a matrix decomposition perspective, but it is only applicable to their own spectral contrastive loss.

Besides the limitations of existing contrastive SSL theories, there are also some interesting empirical observations that have not been unraveled theoretically yet. For example, why does the richer data

---

*Equal contribution ($\alpha$-$\beta$ ordering). Correspondence to Weiran Huang (weiran.huang@outlook.com).

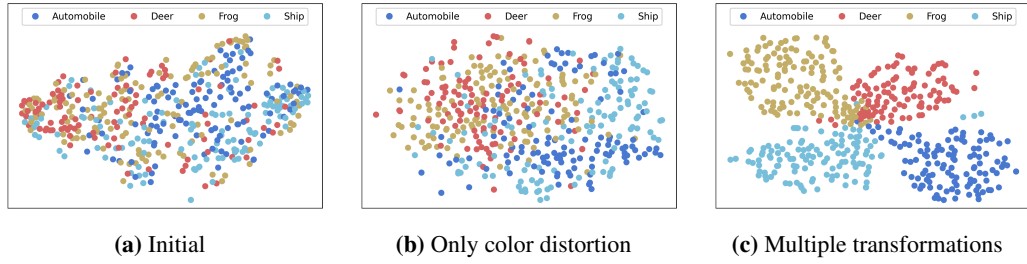

**(a)** Initial      **(b)** Only color distortion      **(c)** Multiple transformations

Figure 1: SimCLR's embedding space with different richnesses of data augmentations on CIFAR-10.

augmentation lead to the more clustered structure in the embedding space (Figure 1) as well as the better downstream performance (also observed by Chen et al. (2020a))? Why is aligning positive samples (augmented from the "same data point") able to gather the samples from the "same latent class" into a cluster (Figure 1c)? More interestingly, decorrelating components of representation like Barlow Twins (Zbontar et al., 2021) does not directly optimize the geometry of embedding space, but it still results in the clustered structure. Why is this?

In this paper, we focus on exploring the generalization ability of contrastive SSL *provably*, which can explain the above interesting observations. We start with understanding the role of data augmentation in contrastive SSL. Intuitively, samples from the same latent class are likely to have similar augmented views, which are mapped to the close locations in the embedding space. Since the augmented views of each sample are encouraged to be clustered in the embedding space by contrastive learning, different samples from the same latent class tend to be pulled closer. As an example, let's consider two images of dogs with different backgrounds (Figure 2). If we augment them with transformation "crop", we may get two similar views (dog heads), whose representations (gray points in the embedding space) are close. As the augmented views of

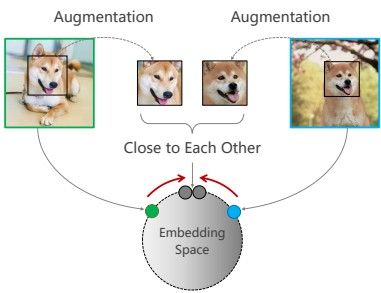

Figure 2: Mechanism of Clustering

each dog image are enforced to be close in the embedding space due to the objective of contrastive learning, the representations of two dog images (green and blue points) will be pulled closer to their augmented views (gray points). In this way, aligning positive samples is able to gather samples from the same class, and thus results in the clustered embedding space. Following the above intuition, we define the *augmented distance* between two samples as the minimum distance between their augmented views, and further introduce the $(\sigma, \delta)$-augmentation to measure the concentration of augmented data, i.e., for each latent class, the proportion of samples located in a ball with diameter $\delta$ (w.r.t. the augmented distance) is larger than $\sigma$.

With the mathematical description of data augmentation settled, we then prove an upper bound of downstream classification error rate in Section 3. It reveals that the generalization of contrastive SSL is related to three key factors. The first one is *alignment* of positive samples, which is a common objective that contrastive learning algorithms aim to optimize. The second one is *divergence* of class centers, which prevents the collapse of representation. The third factor is *concentration* of augmented data, i.e., a sharper concentration of augmented data indicates a better generalization error bound. We remark that the first two factors are properties of representations that can be optimized during the learning process. However, the third factor is determined by pre-defined data augmentation and is independent of the learning process. Thus, data augmentation plays a crucial role in contrastive SSL.

We then study the above three factors in more depth. In Section 4, we rigorously prove that not only the InfoNCE loss but also the cross-correlation loss (which does not directly optimize the geometry of embedding space) can satisfy the first two factors. For the third factor, we conduct various experiments on the real-world datasets and observe that the downstream performance of contrastive SSL is highly correlated to the concentration of augmented data in Section 5.

In summary, our contributions include: 1) proposing a novel $(\sigma, \delta)$-measure to quantify data augmentation; 2) presenting a theoretical framework for contrastive SSL that highlights alignment,

divergence, and concentration as key factors for generalization ability; 3) provably verifying that not only the InfoNCE loss but also the cross-correlation loss satisfy alignment and divergence; 4) showing a strong correlation between downstream performance and concentration of augmented data.

## RELATED WORK

**Algorithms of Contrastive SSL.** Early works such as MoCo (He et al., 2020) and SimCLR (Chen et al., 2020a), use the InfoNCE loss to pull the positive samples close while enforcing them away from the negative samples in the embedding space. These methods require large batch sizes (Chen et al., 2020a), memory banks (He et al., 2020), or carefully designed negative sampling strategies (Hu et al., 2021). To obviate these, some recent works get rid of negative samples and prevent representation collapse by cross-correlation loss (Zbontar et al., 2021; Bardes et al., 2021) or training strategies (Grill et al., 2020; Chen & He, 2021). In this paper, we mainly study the effectiveness of the InfoNCE loss and the cross-correlation loss, and do not enter the discussion of training strategies.

**Theoretical Understandings of Contrastive SSL.** Most theoretical analysis is based on the InfoNCE loss, and lack of understanding of recently proposed cross-correlation loss (Zbontar et al., 2021). Early works understand the InfoNCE loss based on maximizing the mutual information (MI) between positive samples (Oord et al., 2018; Bachman et al., 2019; Hjelm et al., 2018; Tian et al., 2019; 2020; Tschannen et al., 2019). However, a rigorous relationship between mutual information and downstream performance has not been established. Besides, Arora et al. (2019) directly analyze the generalization of InfoNCE loss based on the assumption that positive samples are drawn from the same latent classes, which is different from practical algorithms. Ash et al. (2021) study the role of negative samples and show an interesting collision-coverage trade-off theoretically. HaoChen et al. (2021) study contrastive SSL from a matrix decomposition perspective, but it is only applicable to their spectral contrastive loss. The behavior of InfoNCE is also studied from the perspective of alignment and uniformity (Wang & Isola, 2020), sparse coding model (Wen & Li, 2021), the expansion assumption (Wei et al., 2020), stochastic neighbor embedding (Hu et al., 2022), and augmentation robustness (Zhao et al., 2023).

## 2 PROBLEM FORMULATION

Given a number of unlabeled training data i.i.d. drawn from an unknown distribution, each sample belongs to one of $K$ latent classes $C_1, C_2, \ldots, C_K$. Based on an augmentation set $A$, the set of potential positive samples generated from a data point $\boldsymbol{x}$ is denoted as $A(\boldsymbol{x})$. We assume that $\boldsymbol{x} \in A(\boldsymbol{x})$ for any $\boldsymbol{x}$, and samples from different latent classes never transform into the same augmented sample, i.e., $A(C_k) \cap A(C_\ell) = \varnothing$ for any $k \neq \ell$. Notation $\| \cdot \|$ in this paper stands for $\ell_2$-norm or Frobenius norm for vectors and matrices, respectively.

Contrastive SSL aims to learn an encoder $f$, such that positive samples are closely aligned. In order to make the samples from different latent classes far away from each other, some methods such as (Chen et al., 2020a; He et al., 2020) use the InfoNCE loss[1] to push away negative pairs, formulated as

$$\mathcal{L}_{\text{InfoNCE}} = - \mathop{\mathbb{E}}_{\substack{\boldsymbol{x}, \boldsymbol{x}' \\ \boldsymbol{x}^- \in A(\boldsymbol{x}')}} \mathop{\mathbb{E}}_{\boldsymbol{x}_1, \boldsymbol{x}_2 \in A(\boldsymbol{x})} \log \frac{e^{f(\boldsymbol{x}_1)^\top f(\boldsymbol{x}_2)}}{e^{f(\boldsymbol{x}_1)^\top f(\boldsymbol{x}_2)} + e^{f(\boldsymbol{x}_1)^\top f(\boldsymbol{x}^-)}},$$

where $\boldsymbol{x}, \boldsymbol{x}'$ are two random data points. Some other methods such as Barlow Twins (Zbontar et al., 2021) use the cross-correlation loss to decorrelate the components of representation, formulated as

$$\mathcal{L}_{\text{Cross-Corr}} = \sum_{i=1}^{d} (1 - F_{ii})^2 + \lambda \sum_{i=1}^{d} \sum_{i \neq j} F_{ij}^2,$$

where $F_{ij} = \mathbb{E}_{\boldsymbol{x}} \mathbb{E}_{\boldsymbol{x}_1, \boldsymbol{x}_2 \in A(\boldsymbol{x})} [f_i(\boldsymbol{x}_1) f_j(\boldsymbol{x}_2)]$, $d$ is the dimension of encoder $f$, and encoder $f$ is normalized as $\mathbb{E}_{\boldsymbol{x}} \mathbb{E}_{\boldsymbol{x}' \in A(\boldsymbol{x})} [f_i(\boldsymbol{x}')^2] = 1$ for each dimension $i$.

The standard evaluation of contrastive SSL is to train a linear classifier over the learned representation using labeled data and regard its performance as the indicator. To simplify the analysis, we instead

---

[1]For simplicity in our analysis, we consider the InfoNCE loss with only one negative sample.

consider a non-parametric classifier – nearest neighbor (NN) classifier:

$$G_f(\boldsymbol{x}) = \arg\min_{k \in [K]} \|f(\boldsymbol{x}) - \mu_k\|,$$

where $\mu_k := \mathbb{E}_{\boldsymbol{x} \in C_k} \mathbb{E}_{\boldsymbol{x}' \in A(\boldsymbol{x})}[f(\boldsymbol{x}')]$ is the center of class $C_k$. In fact, the NN classifier is a special case of linear classifier, since it can be reformulated as $G_f(\boldsymbol{x}) = \arg\max_{k \in [K]} (Wf(\boldsymbol{x}) + b)_k$, where the $k$-th row of $W$ is $\mu_k$ and $b_k = -\frac{1}{2}\|\mu_k\|^2$ (See Appendix E). Therefore, the directly learned linear classifier used in practice should perform better than the NN classifier. In this paper, we use the classification error rate to quantify the performance of $G_f$, formulated as

$$\mathrm{Err}(G_f) = \sum_{k=1}^{K} \mathbb{P}[G_f(\boldsymbol{x}) \neq k, \forall \boldsymbol{x} \in C_k].$$

Our goal is to study why contrastive SSL is able to achieve a small $\mathrm{Err}(G_f)$.

## 3 GENERALIZATION GUARANTEE OF CONTRASTIVE SSL

Based on the NN classifier, if the samples are well clustered by latent classes in the embedding space, the error rate $\mathrm{Err}(G_f)$ should be small. Thus, one expects to have a small intra-class distance $\mathbb{E}_{\boldsymbol{x}_1, \boldsymbol{x}_2 \in C_k} \|f(\boldsymbol{x}_1) - f(\boldsymbol{x}_2)\|^2$ for an encoder $f$ learned by contrastive learning. However, contrastive algorithms can only control the alignment of positive samples $\mathbb{E}_{\boldsymbol{x}_1, \boldsymbol{x}_2 \in A(\boldsymbol{x})} \|f(\boldsymbol{x}_1) - f(\boldsymbol{x}_2)\|^2$. To bridge the gap between them, we need to investigate the role of data augmentation.

Motivated by Figure 2 introduced in Section 1, for a given augmentation set $A$, we define the *augmented distance* between two samples as the minimum distance between their augmented views:

$$d_A(\boldsymbol{x}_1, \boldsymbol{x}_2) = \min_{\boldsymbol{x}_1' \in A(\boldsymbol{x}_1), \boldsymbol{x}_2' \in A(\boldsymbol{x}_2)} \|\boldsymbol{x}_1' - \boldsymbol{x}_2'\|. \tag{1}$$

For the dog images in Figure 2 as an example, even though their pixel-level differences are significant, their semantic meanings are similar. Meanwhile, they also have a small augmented distance. Thus, the proposed augmented distance can partially capture the semantic distance. Based on the augmented distance, we now introduce the $(\sigma, \delta)$-augmentation to measure the concentration of augmented data.

**Definition 1** ($(\sigma, \delta)$-Augmentation). *The augmentation set $A$ is called a $(\sigma, \delta)$-augmentation, if for each class $C_k$, there exists a subset $C_k^0 \subseteq C_k$ (called a main part of $C_k$), such that both $\mathbb{P}[\boldsymbol{x} \in C_k^0] \geq \sigma \mathbb{P}[\boldsymbol{x} \in C_k]$ where $\sigma \in (0, 1]$ and $\sup_{\boldsymbol{x}_1, \boldsymbol{x}_2 \in C_k^0} d_A(\boldsymbol{x}_1, \boldsymbol{x}_2) \leq \delta$ hold.*

In other words, the main-part samples locate in a ball with diameter $\delta$ (w.r.t. the augmented distance) and its proportion is larger than $\sigma$. Larger $\sigma$ and smaller $\delta$ indicate the sharper concentration of augmented data. For any $A' \supseteq A$ with richer augmentations, one can verify that $d_{A'}(\boldsymbol{x}_1, \boldsymbol{x}_2) \leq d_A(\boldsymbol{x}_1, \boldsymbol{x}_2)$ for any $\boldsymbol{x}_1, \boldsymbol{x}_2$. Therefore, richer data augmentations lead to sharper concentration as $\delta$ gets smaller. With Definition 1, our analysis will focus on the samples in the main parts with good alignment, i.e., $(C_1^0 \cup \cdots \cup C_K^0) \cap S_\varepsilon$, where $S_\varepsilon := \{\boldsymbol{x} \in \cup_{k=1}^K C_k : \forall \boldsymbol{x}_1, \boldsymbol{x}_2 \in A(\boldsymbol{x}), \|f(\boldsymbol{x}_1) - f(\boldsymbol{x}_2)\| \leq \varepsilon\}$ is the set of samples with $\varepsilon$-close representations among augmented data. Furthermore, we let $R_\varepsilon := \mathbb{P}\left[\overline{S_\varepsilon}\right]$, which is provably small with good alignment (see Theorem 2).

**Lemma 3.1.** *For a $(\sigma, \delta)$-augmentation with main part $C_k^0$ of each class $C_k$, if all samples belonging to $(C_1^0 \cup \cdots \cup C_K^0) \cap S_\varepsilon$ can be correctly classified by a classifier $G$, then its classification error rate $\mathrm{Err}(G)$ is upper bounded by $(1 - \sigma) + R_\varepsilon$.*

The proof is deferred to the appendix. The above lemma presents a simple sufficient condition to guarantee the generalization ability on downstream tasks. Based on it, we need to further explore when samples in $(C_1^0 \cup \cdots \cup C_K^0) \cap S_\varepsilon$ can be all correctly classified by the NN classifier.

We assume that encoder $f$ is normalized by $\|f\| = r$, and it is $L$-Lipschitz continuity, i.e., for any $\boldsymbol{x}_1, \boldsymbol{x}_2, \|f(\boldsymbol{x}_1) - f(\boldsymbol{x}_2)\| \leq L \|\boldsymbol{x}_1 - \boldsymbol{x}_2\|$. We let $p_k := \mathbb{P}[\boldsymbol{x} \in C_k]$ for any $k \in [K]$.

**Lemma 3.2.** *Given a $(\sigma, \delta)$-augmentation used in contrastive SSL, for any $\ell \in [K]$, if $\mu_\ell^\top \mu_k < r^2 \left(1 - \rho_\ell(\sigma, \delta, \varepsilon) - \sqrt{2\rho_\ell(\sigma, \delta, \varepsilon)} - \frac{\Delta_\mu}{2}\right)$ holds for all $k \neq \ell$, then every sample $\boldsymbol{x} \in C_\ell^0 \cap S_\varepsilon$ can be correctly classified by the NN classifier $G_f$, where $\rho_\ell(\sigma, \delta, \varepsilon) = 2(1 - \sigma) + \frac{R_\varepsilon}{p_\ell} + \sigma \left(\frac{L\delta}{r} + \frac{2\varepsilon}{r}\right)$ and $\Delta_\mu = 1 - \min_{k \in [K]} \|\mu_k\|^2/r^2$.*

With Lemma 3.1 and 3.2, we can directly obtain the generalization guarantee of contrastive SSL:

**Theorem 1.** *Given a $(\sigma, \delta)$-augmentation used in contrastive SSL, if*

$$\mu_\ell^\top \mu_k < r^2 \left( 1 - \rho_{max}(\sigma, \delta, \varepsilon) - \sqrt{2\rho_{max}(\sigma, \delta, \varepsilon)} - \frac{\Delta_\mu}{2} \right) \tag{2}$$

*holds for any pair of $(\ell, k)$ with $\ell \neq k$, then the downstream error rate of NN classifier $G_f$*

$$\mathrm{Err}(G_f) \leq (1 - \sigma) + R_\varepsilon, \tag{3}$$

*where $\rho_{max}(\sigma, \delta, \varepsilon) = 2(1 - \sigma) + \frac{R_\varepsilon}{\min_\ell p_\ell} + \sigma \left( \frac{L\delta}{r} + \frac{2\varepsilon}{r} \right)$ and $\Delta_\mu = 1 - \min_{k \in [K]} \|\mu_k\|^2 / r^2$.*

The proof is deferred to the appendix. To better understand the above theorem, let us first consider a simple case that any two samples from the latent same class at least own a same augmented view ($\sigma = 1, \delta = 0$), and the positive samples are perfectly aligned after contrastive learning ($\varepsilon = 0, R_\varepsilon = 0$). In this case, the samples from the same latent class are embedded to a single point on the hypersphere, and thus arbitrarily small positive angle $\frac{\langle \mu_\ell, \mu_k \rangle}{\|\mu_\ell\| \cdot \|\mu_k\|} < 1$ is enough to distinguish them by the NN classifier. In fact, one can quickly verify that $\rho_{max}(\sigma, \delta, \varepsilon) = \Delta_\mu = 0$ holds in the above case. According to Theorem 1, if $\mu_\ell^\top \mu_k / r^2 < 1 - \rho_{max}(\sigma, \delta, \varepsilon) - \sqrt{2\rho_{max}(\sigma, \delta, \varepsilon)} - \frac{\Delta_\mu}{2} = 1$, then $\mathrm{Err}(G_f) = 0$, i.e., NN classifier can correctly recognize every sample when $\mu_\ell^\top \mu_k / r^2 < 1$. Thus, the condition suggested by Theorem 1 is exactly the same as the intuition.

Theorem 1 implies three key factors to the success of contrastive SSL. The first one is *alignment* of positive samples, which is a common objective that contrastive algorithms aim to optimize. Better alignment enables smaller $R_\varepsilon$, which directly decreases the generalization error bound (3). The second factor is *divergence* of class centers, i.e., the distance between class centers should be large enough (small $\mu_\ell^\top \mu_k$). The divergence condition (2) is related to the alignment ($R_\varepsilon$) and data augmentation ($\sigma, \delta$). Better alignment and sharper concentration indicate smaller $\rho_{max}(\sigma, \delta, \varepsilon)$, and hence looser divergence condition. The third factor is *concentration* of augmented data. When $\delta$ is given, sharper concentration implies larger $\sigma$, which directly affects the generalization error bound (3). For example, richer data augmentations lead to sharper concentration (see the paragraph below Definition 1), and hence better generalization error bound. Only the first two factors can be optimized during the learning process, and we will provably show how it can be achieved via two concrete examples in Section 4. In contrast, the third factor is priorly decided by the pre-defined data augmentation and is independent of the learning process. We will empirically study how the concentration of augmented data affects the downstream performance in Section 5. In summary, Theorem 1 provides a framework for different algorithms to analyze their generalization abilities.

Compared with the alignment and uniformity proposed by Wang & Isola (2020), both of the works have the same meaning of "alignment" since it is the objective that contrastive algorithms aim to optimize, but our "divergence" is fundamentally different from their "uniformity". Uniformity requires "all data" uniformly distributed on the embedding hypersphere, while our divergence characterizes the cosine distance between "class centers". We do not require the divergence to be as large as better, instead, the divergence condition can be loosened by better alignment and concentration properties. As an example, consider the case below Theorem 1. Since all the samples from the same latent class are embedded into a single point on the hypersphere, in that case, an arbitrarily small positive angle (arbitrarily small divergence) is enough to distinguish them. More importantly, alignment and uniformity are empirical predictors for downstream performance, while our alignment and divergence have explicit theoretical guarantees (Theorem 1) for the generalization of contrastive SSL. Moreover, Wang & Isola (2020) does not consider the crucial effect of data augmentation. In fact, with bad concentration (e.g., only using identity transformation as data augmentation), "perfect" alignment along with "perfect" uniformity still can not imply good downstream performance.

## 3.1 Upper Bound $R_\varepsilon$ via Alignment

We now upper bound $R_\varepsilon$ via the alignment

$$\mathcal{L}_{\mathrm{align}}(f) := \mathop{\mathbb{E}}_{\boldsymbol{x}} \mathop{\mathbb{E}}_{\boldsymbol{x}_1, \boldsymbol{x}_2 \in A(\boldsymbol{x})} \|f(\boldsymbol{x}_1) - f(\boldsymbol{x}_2)\|^2, \tag{4}$$

which is a common objective of contrastive losses. Recall that $R_\varepsilon$ can be rewritten as

$$R_\varepsilon = \mathbb{P} \left[ \boldsymbol{x} \in \cup_{k=1}^K C_k : \sup_{\boldsymbol{x}_1, \boldsymbol{x}_2 \in A(\boldsymbol{x})} \|f(\boldsymbol{x}_1) - f(\boldsymbol{x}_2)\| > \varepsilon \right].$$

Note that there is a gap between "sup operator" in $R_\varepsilon$ and "$\mathbb{E}$ operator" in $\mathcal{L}_{\text{align}}(f)$, which cannot be simply derived by concentration inequalities.

We separate the augmentation set $A$ as discrete transformations $\{A_\gamma(\cdot)\colon \gamma \in [m]\}$ and continuous transformations $\{A_\theta(\cdot)\colon \theta \in [0,1]^n\}$. For example, random cropping or flipping can be categorized into the discrete transformation, while the others like random color distortion or Gaussian blur can be regarded as the continuous transformation parameterized by the augmentation strength $\theta$. Without loss of generality, we assume that for any given $\boldsymbol{x}$, its augmented data are uniformly random sampled, i.e., $\mathbb{P}[\boldsymbol{x}' = A_\gamma(\boldsymbol{x})] = \frac{1}{2m}$ and $\mathbb{P}[\boldsymbol{x}' \in \{A_\theta(\boldsymbol{x})\colon \theta \in \Theta\}] = \frac{\text{vol}(\Theta)}{2}$ for any $\Theta \subseteq [0,1]^n$, where $\text{vol}(\Theta)$ denotes the volume of $\Theta$. For the continuous transformation, we further assume that the transformation is $M$-Lipschitz continuous w.r.t. $\theta$, i.e., $\|A_{\theta_1}(\boldsymbol{x}) - A_{\theta_2}(\boldsymbol{x})\| \leq M\|\theta_1 - \theta_2\|$ for any $\boldsymbol{x}, \theta_1, \theta_2$. With the above setting, we have the following theorem (proof is deferred to the appendix).

**Theorem 2.** *If encoder $f$ is $L$-Lipschitz continuous, then*

$$R_\varepsilon^2 \leq \eta(\varepsilon)^2 \cdot \mathop{\mathbb{E}}_{\boldsymbol{x}} \mathop{\mathbb{E}}_{\boldsymbol{x}_1, \boldsymbol{x}_2 \in A(\boldsymbol{x})} \|f(\boldsymbol{x}_1) - f(\boldsymbol{x}_2)\|^2 = \eta(\varepsilon)^2 \cdot \mathcal{L}_{\text{align}}(f),$$

*where* $\eta(\varepsilon) = \inf_{h \in \left(0, \frac{\varepsilon}{2\sqrt{n}LM}\right)} \frac{4\max\{1, m^2 h^{2n}\}}{h^{2n}(\varepsilon - 2\sqrt{n}LMh)}$.

The above theorem confirms that, with good alignment, $R_\varepsilon$ is guaranteed to be small.

## 4 CONTRASTIVE LOSSES MEET ALIGNMENT AND DIVERGENCE

We now study two canonical contrastive losses, the InfoNCE loss and the cross-correlation loss, to see how they can achieve good alignment (small $\mathcal{L}_{\text{align}}(f)$) and good divergence (small $\mu_k^\top \mu_\ell$).

### 4.1 INFONCE LOSS

The population loss of InfoNCE (Chen et al., 2020a; He et al., 2020) is well known as:

$$\mathcal{L}_{\text{InfoNCE}} = - \mathop{\mathbb{E}}_{\substack{\boldsymbol{x}, \boldsymbol{x}'}} \mathop{\mathbb{E}}_{\substack{\boldsymbol{x}_1, \boldsymbol{x}_2 \in A(\boldsymbol{x}) \\ \boldsymbol{x}^- \in A(\boldsymbol{x}')}} \log \frac{e^{f(\boldsymbol{x}_1)^\top f(\boldsymbol{x}_2)}}{e^{f(\boldsymbol{x}_1)^\top f(\boldsymbol{x}_2)} + e^{f(\boldsymbol{x}_1)^\top f(\boldsymbol{x}^-)}},$$

where encoder $f$ is normalized by $\|f\| = 1$. It can be divided into two parts:

$$\mathcal{L}_{\text{InfoNCE}} = \mathop{\mathbb{E}}_{\substack{\boldsymbol{x}, \boldsymbol{x}'}} \mathop{\mathbb{E}}_{\substack{\boldsymbol{x}_1, \boldsymbol{x}_2 \in A(\boldsymbol{x}) \\ \boldsymbol{x}^- \in A(\boldsymbol{x}')}} \left[ -f(\boldsymbol{x}_1)^\top f(\boldsymbol{x}_2) + \log\left(e^{f(\boldsymbol{x}_1)^\top f(\boldsymbol{x}_2)} + e^{f(\boldsymbol{x}_1)^\top f(\boldsymbol{x}^-)}\right) \right] \quad (5)$$

$$= \underbrace{\frac{1}{2} \mathop{\mathbb{E}}_{\boldsymbol{x}} \mathop{\mathbb{E}}_{\boldsymbol{x}_1, \boldsymbol{x}_2 \in A(\boldsymbol{x})} [\|f(\boldsymbol{x}_1) - f(\boldsymbol{x}_2)\|^2] - 1}_{=:\mathcal{L}_1^{\text{InfoNCE}}(f)} + \underbrace{\mathop{\mathbb{E}}_{\substack{\boldsymbol{x}, \boldsymbol{x}'}} \mathop{\mathbb{E}}_{\substack{\boldsymbol{x}_1, \boldsymbol{x}_2 \in A(\boldsymbol{x}) \\ \boldsymbol{x}^- \in A(\boldsymbol{x}')}} \left[\log\left(e^{f(\boldsymbol{x}_1)^\top f(\boldsymbol{x}_2)} + e^{f(\boldsymbol{x}_1)^\top f(\boldsymbol{x}^-)}\right)\right]}_{=:\mathcal{L}_2^{\text{InfoNCE}}(f)}.$$

Regardless of the constant factors, $\mathcal{L}_1^{\text{InfoNCE}}(f)$ is exactly the alignment term in (4). Next, we take a close look at $\mathcal{L}_2^{\text{InfoNCE}}(f)$ to see how it links to the divergence condition required by Theorem 1.

**Theorem 3.** *Assume that encoder $f$ with norm 1 is $L$-Lipschitz continuous. If the augmented data is $(\sigma, \delta)$-augmented, then for any $\varepsilon \geq 0$ and $k \neq \ell$, we have*

$$\mu_k^\top \mu_\ell \leq \log\left(\exp\left\{\frac{\mathcal{L}_2^{\text{InfoNCE}}(f) + \tau(\sigma, \delta, \varepsilon, R_\varepsilon)}{p_k p_\ell}\right\} - \exp(1 - \varepsilon)\right),$$

*where $\tau(\sigma, \delta, \varepsilon, R_\varepsilon)$ is a non-negative term, decreasing with smaller $\varepsilon$, $R_\varepsilon$ or sharper concentration of augmented data, and $\tau(\sigma, \delta, \varepsilon, R_\varepsilon) = 0$ when $\sigma = 1, \delta = 0, \varepsilon = 0, R_\varepsilon = 0$.*

The specific formulation of $\tau(\sigma, \delta, \varepsilon, R_\varepsilon)$ and the proof are deferred to the appendix. We remark that data augmentation $(\sigma, \delta)$, parameter $\varepsilon$, and $p_k, p_\ell$ are pre-determined before training procedure, and thus the upper bound of $\mu_k^\top \mu_\ell$ in Theorem 3 varies only with $\mathcal{L}_2^{\text{InfoNCE}}(f)$ and $R_\varepsilon$, positively.

Therefore, minimizing $\mathcal{L}_{\text{InfoNCE}} = \mathcal{L}_1^{\text{InfoNCE}}(f) + \mathcal{L}_2^{\text{InfoNCE}}(f)$ leads to both small $\mathcal{L}_1^{\text{InfoNCE}}(f)$ and small $\mathcal{L}_2^{\text{InfoNCE}}(f)$. Small $\mathcal{L}_1^{\text{InfoNCE}}(f)$ indicates good alignment $\mathcal{L}_{\text{align}}(f)$, as well as small $R_\varepsilon$

(Theorem 2). Small $\mathcal{L}_2^{\text{InfoNCE}}(f)$ along with small $R_\varepsilon$ indicates good divergence (small $\mu_k^\top \mu_\ell$) by Theorem 3. Hence, optimizing the InfoNCE loss can achieve both good alignment and good divergence. According to Theorem 1 and Theorem 2, the generalization ability of encoder $f$ on the downstream task is implied, i.e., $\text{Err}(G_f) \leq (1 - \sigma) + \eta(\varepsilon)\sqrt{2 + 2\mathcal{L}_1^{\text{InfoNCE}}(f)}$, when the upper bound of $\mu_k^\top \mu_\ell$ in Theorem 3 is smaller than the threshold in Theorem 1.

It is worth mentioning that the form of InfoNCE is critical to meeting the requirement of divergence, which is found when we prove Theorem 3. For example, let us consider the contrastive loss (5) formulated in a linear form[2] instead of `LogExp` such that

$$\mathcal{L}'(f) = \mathop{\mathbb{E}}_{\substack{\boldsymbol{x}, \boldsymbol{x}' \\ \boldsymbol{x}^- \in A(\boldsymbol{x}')}} \mathop{\mathbb{E}}_{\boldsymbol{x}_1, \boldsymbol{x}_2 \in A(\boldsymbol{x})} \left[ -f(\boldsymbol{x}_1)^\top f(\boldsymbol{x}_2) + \lambda f(\boldsymbol{x}_1)^\top f(\boldsymbol{x}^-) \right] = \mathcal{L}_1^{\text{InfoNCE}}(f) + \lambda \mathcal{L}_2'(f),$$

where $\mathcal{L}_2'(f)$ is the negative-pair term weighted by some $\lambda > 0$. Due to the independence between $\boldsymbol{x}$ and $\boldsymbol{x}'$, we have $\mathcal{L}_2'(f) = \| \mathbb{E}_{\boldsymbol{x}} \mathbb{E}_{\boldsymbol{x}_1 \in A(\boldsymbol{x})}[f(\boldsymbol{x}_1)]\|^2$. Therefore, minimizing $\mathcal{L}_2'(f)$ only leads to the representation with zero mean. Unfortunately, the objective of zero mean with $\|f\| = 1$ can not obviate the dimensional collapse (Hua et al., 2021) of the model. For example, the encoder $f$ can map the input data from multi classes into two points in the opposite directions on the hypersphere. This justifies the observation in (Wang & Liu, 2021): the uniformity of the encoder on the embedded hypersphere becomes worse when the temperature of the loss increases, where the loss degenerates to $\mathcal{L}'(f)$ with infinite temperature.

## 4.2 CROSS-CORRELATION LOSS

Cross-correlation loss is first introduced by Barlow Twins (Zbontar et al., 2021). In contrast to InfoNCE loss, it trains the model via decorrelating the components of representation instead of directly optimizing the geometry of embedding space, but it is still observed to have clustered embedding space. To explore this, we study the cross-correlation loss in detail and show how it implicitly optimizes the alignment and divergence required by Theorem 1.

The population loss of cross-correlation can be formulated as

$$\mathcal{L}_{\text{Cross-Corr}} = \sum_{i=1}^{d} \left( 1 - \mathop{\mathbb{E}}_{\boldsymbol{x}} \mathop{\mathbb{E}}_{\boldsymbol{x}_1, \boldsymbol{x}_2 \in A(\boldsymbol{x})}[f_i(\boldsymbol{x}_1)f_i(\boldsymbol{x}_2)] \right)^2 + \lambda \sum_{i \neq j} \left( \mathop{\mathbb{E}}_{\boldsymbol{x}} \mathop{\mathbb{E}}_{\boldsymbol{x}_1, \boldsymbol{x}_2 \in A(\boldsymbol{x})}[f_i(\boldsymbol{x}_1)f_j(\boldsymbol{x}_2)] \right)^2,$$

with normalization condition of $\mathbb{E}_{\boldsymbol{x}} \mathbb{E}_{\boldsymbol{x}_1 \in A(\boldsymbol{x})}[f_i(\boldsymbol{x}_1)] = 0$ and $\mathbb{E}_{\boldsymbol{x}} \mathbb{E}_{\boldsymbol{x}_1 \in A(\boldsymbol{x})}[f_i(\boldsymbol{x}_1)^2] = 1$ for each $i \in [d]$, where $d$ is the output dimension of encoder $f$. Positive coefficient $\lambda$ balances the importance between diagonal and non-diagonal elements of cross-correlation matrix. When $\lambda = 1$, the above loss is exactly the difference between the cross-correlation matrix and identity matrix. Similar to Section 4.1, we first divide the loss into two parts, by defining

$$\mathcal{L}_1^{\text{Cross}}(f) := \sum_{i=1}^{d} \left( 1 - \mathop{\mathbb{E}}_{\boldsymbol{x}} \mathop{\mathbb{E}}_{\boldsymbol{x}_1, \boldsymbol{x}_2 \in A(\boldsymbol{x})}[f_i(\boldsymbol{x}_1)f_i(\boldsymbol{x}_2)] \right)^2 \text{ and } \mathcal{L}_2^{\text{Cross}}(f) := \left\| \mathop{\mathbb{E}}_{\boldsymbol{x}} \mathop{\mathbb{E}}_{\boldsymbol{x}_1, \boldsymbol{x}_2 \in A(\boldsymbol{x})}[f(\boldsymbol{x}_1)f(\boldsymbol{x}_2)^\top] - I_d \right\|^2.$$

In this way, the cross-correlation loss becomes $\mathcal{L}_{\text{Cross-Corr}} = (1 - \lambda)\mathcal{L}_1^{\text{Cross}}(f) + \lambda \mathcal{L}_2^{\text{Cross}}(f)$. Then, we connect $\mathcal{L}_1^{\text{Cross}}(f)$ and $\mathcal{L}_2^{\text{Cross}}(f)$ with the alignment and divergence, respectively.

**Lemma 4.1.** *For a given encoder $f$, the alignment $\mathcal{L}_{\text{align}}(f)$ in (4) is upper bounded via $\mathcal{L}_1^{Cross}(f)$:*

$$\mathcal{L}_{\text{align}}(f) = \mathop{\mathbb{E}}_{\boldsymbol{x}} \mathop{\mathbb{E}}_{\boldsymbol{x}_1, \boldsymbol{x}_2 \in A(\boldsymbol{x})} \|f(\boldsymbol{x}_1) - f(\boldsymbol{x}_2)\|^2 \leq 2\sqrt{d \cdot \mathcal{L}_1^{\text{Cross}}(f)},$$

*where $d$ is the output dimension of encoder $f$.*

The above lemma connects $\mathcal{L}_1^{\text{Cross}}(f)$ with $\mathcal{L}_{\text{align}}(f)$, indicating that the diagonal elements of the cross-correlation matrix determine the alignment of positive samples. Next, we will link $\mathcal{L}_2^{\text{Cross}}(f)$ to the divergence $\mu_k^\top \mu_\ell$. It is challenging because $\mathcal{L}_2^{\text{Cross}}(f)$ is designed for reducing the redundancy between the encoder's output units, not for optimizing the geometry of embedding space.

---

[2]It is also called *simple contrastive loss* in some literature.

**Theorem 4.** *Assume that encoder $f$ with norm $\sqrt{d}$ is $L$-Lipschitz continuous. If the augmented data is $(\sigma, \delta)$-augmented, then for any $\varepsilon \geq 0$ and $k \neq \ell$, we have*

$$\mu_k^\top \mu_\ell \leq \sqrt{\frac{2}{p_k p_\ell} \left( \mathcal{L}_2^{\text{Cross}}(f) + \tau'(\sigma, \delta, \varepsilon, R_\varepsilon) - \frac{d - K}{2} \right)},$$

*where $\tau'(\sigma, \delta, \varepsilon, R_\varepsilon)$ is an upper bound of $\| \mathbb{E}_{\boldsymbol{x}} \mathbb{E}_{\boldsymbol{x}_1, \boldsymbol{x}_2 \in A(\boldsymbol{x})}[f(\boldsymbol{x}_1)f(\boldsymbol{x}_2)^\top] - \sum_{k=1}^{K} p_k \mu_k \mu_k^\top \|^2$.*

The specific formulation of $\tau'(\sigma, \delta, \varepsilon, R_\varepsilon)$ and proof are deferred to the appendix. Here we remark that $\tau'(\sigma, \delta, \varepsilon, R_\varepsilon)$ is a non-negative term, decreasing with smaller $\varepsilon$, $R_\varepsilon$ or sharper concentration of augmented data, and $\tau'(\sigma, \delta, \varepsilon, R_\varepsilon) = 0$ when $\sigma = 1, \delta = 0, \varepsilon = 0, R_\varepsilon = 0$. Since data augmentation $(\sigma, \delta)$, parameter $\varepsilon$, and $p_k, p_\ell$ are pre-determined before training procedure, the upper bound of $\mu_k^\top \mu_\ell$ in Theorem 4 varies only with $\mathcal{L}_2^{\text{Cross}}(f)$ and $R_\varepsilon$, positively.

Therefore, minimizing $\mathcal{L}_{\text{Cross-Corr}}$ leads to small $\mathcal{L}_1^{\text{Cross}}(f)$, as well as small $\mathcal{L}_2^{\text{Cross}}(f)$. Small $\mathcal{L}_1^{\text{Cross}}(f)$ indicates good alignment $\mathcal{L}_{\text{align}}(f)$ by Lemma 4.1 and small $R_\varepsilon$ by Theorem 2. Small $\mathcal{L}_2^{\text{Cross}}(f)$ along with small $R_\varepsilon$ indicates good divergence (small $\mu_k^\top \mu_\ell$) by Theorem 4. Hence, decorrelating the components of representation can achieve both good alignment and good divergence. According to Theorem 1 and Theorem 2, the generalization ability of encoder $f$ on the downstream task is implied, i.e., $\text{Err}(G_f) \leq (1 - \sigma) + \sqrt{2}\,\eta(\varepsilon)\,d^{\frac{1}{4}} \mathcal{L}_1^{\text{Cross}}(f)^{\frac{1}{4}}$, when the upper bound of $\mu_k^\top \mu_\ell$ in Theorem 4 is smaller than the threshold in Theorem 1.

Beyond the above two widely used contrastive learning losses, we further analyze a very recently proposed $t$-InfoNCE loss (Hu et al., 2022), which is a $t$-SNE style loss inspired by stochastic neighbor embedding. We show that it can also achieve good alignment and divergence in the appendix.

## 5 EMPIRICAL STUDY OF CONCENTRATION OF AUGMENTED DATA

Theorem 1 reveals that sharper concentration of augmented data w.r.t. the proposed augmented distance implies better generalization error bound regardless of algorithm. In this section, we empirically study the relationship between the concentration level and the real downstream performance.

**Basic Setup.** Our experiments are conducted on CIFAR-10 and CIFAR-100 (Krizhevsky, 2009). We consider 5 different kinds of transformations for performing data augmentations: (a) random cropping; (b) random Gaussian blur; (c) color dropping (i.e., randomly converting images to grayscale); (d) color distortion; (e) random horizontal flipping. We test different combinations of transformations via various SSL algorithms such as SimCLR (Chen et al., 2020a), Barlow Twins (Zbontar et al., 2021), MoCo (He et al., 2020), and SimSiam (Chen & He, 2021). We use ResNet-18 (He et al., 2016) as the encoder, and the other settings such as projection head remain the same as the original settings of algorithms. Each model is trained with a batch size of 512 and 800 epochs. To evaluate the quality of the encoder, we follow the KNN evaluation protocol (Wu et al., 2018).

**Different Richness of Augmentations.** We compose all 5 kinds of transformations together, and then successively drop one of the composed transformations from (e) to (b) to conduct 5 experiments for each dataset (Table 1). We observe that the downstream performance monotonously gets worse with the decrease of transformation number, under all four SSL algorithms, on both CIFAR-10 and CIFAR-100. Notice that richer augmentation implies sharper concentration (see the paragraph below Definition 1), and thus the concentration becomes less sharp from top to bottom for each dataset. Therefore, we observe that downstream performance becomes better with sharper concentration.

We also observe that (c) color dropping and (d) color distortion have a great impact on the performance of all algorithms. According to our theoretical framework, these two transformations enable the augmented data to vary in a very wide range, which makes the augmented distance (1) largely decrease. As an intuitive example, if the right dog image in Figure 2 is replaced by a Husky image, only with random cropping, one will get two dog heads with similar shapes but different colors, which still have a large augmented distance. Instead, if color distortion is further applied, one can get two similar dog heads both in shape and color. Therefore, these two dog images have similar augmented views, and thus their augmented distance (1) becomes very small. Notice that small augmented distance (1) indicates sharp concentration (small $\delta$ in Definition 1). Therefore, we observe that dramatic change in concentration leads to wildly fluctuating downstream performance.

Table 1: Downstream performance under different richness of augmentations.

| Dataset | Transformations | | | | | Accuracy | | | |
|---|---|---|---|---|---|---|---|---|---|
| | (a) | (b) | (c) | (d) | (e) | SimCLR | Barlow Twins | MoCo | SimSiam |
| CIFAR-10 | ✓ | ✓ | ✓ | ✓ | ✓ | **89.76 ± 0.12** | **86.91 ± 0.09** | **90.12 ± 0.12** | **90.59 ± 0.11** |
| | ✓ | ✓ | ✓ | ✓ | | 88.48 ± 0.22 | 85.38 ± 0.37 | 89.69 ± 0.11 | 89.34 ± 0.09 |
| | ✓ | ✓ | ✓ | | | 83.50 ± 0.14 | 82.00 ± 0.59 | 86.78 ± 0.07 | 85.38 ± 0.09 |
| | ✓ | ✓ | | | | 63.23 ± 0.05 | 67.83 ± 0.94 | 75.12 ± 0.28 | 63.27 ± 0.30 |
| | ✓ | | | | | 62.74 ± 0.18 | 67.77 ± 0.69 | 74.94 ± 0.22 | 61.47 ± 0.74 |
| CIFAR-100 | ✓ | ✓ | ✓ | ✓ | ✓ | **57.74 ± 0.12** | **57.99 ± 0.29** | **64.19 ± 0.14** | **63.48 ± 0.16** |
| | ✓ | ✓ | ✓ | ✓ | | 55.43 ± 0.10 | 55.22 ± 0.25 | 62.50 ± 0.28 | 60.31 ± 0.41 |
| | ✓ | ✓ | ✓ | | | 45.10 ± 0.25 | 50.40 ± 0.64 | 57.04 ± 0.21 | 51.42 ± 0.14 |
| | ✓ | ✓ | | | | 28.01 ± 0.18 | 34.11 ± 0.59 | 40.18 ± 0.04 | 26.26 ± 0.30 |
| | ✓ | | | | | 27.95 ± 0.09 | 34.05 ± 1.13 | 39.63 ± 0.31 | 25.90 ± 0.83 |

Table 2: Downstream performance under different strength of augmentations.

| Dataset | Color Distortion Strength | Accuracy | | | |
|---|---|---|---|---|---|
| | | SimCLR | Barlow Twins | MoCo | SimSiam |
| CIFAR-10 | 1 | **82.75 ± 0.24** | **82.58 ± 0.25** | **86.68 ± 0.05** | **82.50 ± 1.05** |
| | 1/2 | 78.76 ± 0.18 | 81.88 ± 0.25 | 84.30 ± 0.14 | 81.80 ± 0.15 |
| | 1/4 | 76.37 ± 0.11 | 79.64 ± 0.34 | 82.76 ± 0.09 | 78.80 ± 0.17 |
| | 1/8 | 74.23 ± 0.16 | 77.96 ± 0.16 | 81.20 ± 0.12 | 76.09 ± 0.50 |
| CIFAR-100 | 1 | **46.67 ± 0.42** | **50.39 ± 1.09** | **58.50 ± 0.51** | **49.94 ± 2.01** |
| | 1/2 | 40.21 ± 0.05 | 48.76 ± 0.25 | 55.08 ± 0.09 | 46.27 ± 0.46 |
| | 1/4 | 36.67 ± 0.08 | 46.22 ± 0.71 | 52.09 ± 0.18 | 42.02 ± 0.34 |
| | 1/8 | 34.75 ± 0.20 | 44.72 ± 0.26 | 49.43 ± 0.16 | 36.26 ± 0.34 |

**Different Strength of Augmentations.** We fix (a) random cropping and (d) color distortion as data augmentation, and vary the strength of (d) in $\{1, \frac{1}{2}, \frac{1}{4}, \frac{1}{8}\}$ to construct 4 groups of augmentations with different strength levels (Table 2). We observe that the downstream performance monotonously decreases with weaker color distortions, under all four SSL algorithms, on both CIFAR-10 and CIFAR-100. Recall that a stronger color distortion makes the augmented data vary in a wider range, leading to a smaller augmented distance (1) and thus sharper concentration. Therefore, we observe again that downstream performance becomes better with sharper concentration.

**Different Composed Pairs of Transformations.** To study the relationship between the concentration level and the corresponding downstream performance in a more fine-grained way, we compose transformations (a)-(e) in pairs to construct a total of $\binom{5}{2} = 10$ augmentations. Contrasted to the previous two groups of experiments, current composed augmentations do not have an apparent order of concentration levels. According to Definition 1, for a given $\delta$, a smaller $(1 - \sigma)$ corresponds to a sharper concentration. Thus, we mathematically compute $(1 - \sigma)$ (see appendix for details), and observe the correlation between classification error rate $\text{Err}(G_f)$ and $(1 - \sigma)$ under different $\delta$ on CIFAR-10, based on the SimCLR model trained with 200 epochs.

Interestingly, downstream performance is surprisingly highly correlated to the concentration level (Figure 3). Specifically, if we fix one of composed transformations as (a), we find that both $\text{Err}(G_f)$ and $(1 - \sigma)$ have the same order that $(a, d) < (a, c) < (a, e) \approx (a, b)$, under two values of $\delta$. Furthermore, among all 10 composed augmentations, augmentation $(a, d)$ has the smallest value of $(1 - \sigma)$, while the corresponding performance is also the best one. In addition, we observe that the choice of $\delta$ is not sensitive to the curve shape of $(1-\sigma)$. These observations suggest that sharper concentration is most likely to have better downstream performance. This also provides an explanation for Figure 5 in Sim-CLR paper (Chen et al., 2020a) of why the composition of "crop & color" performs the best.

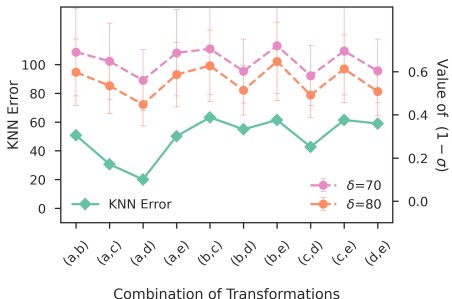

Figure 3: The correlation between observed $\text{Err}(G_f)$ and computed value of $(1 - \sigma)$.

ACKNOWLEDGMENT

We would like to express our sincere gratitude to the reviewers of ICLR 2023 for their insightful and constructive feedback. Their valuable comments have greatly contributed to improving the quality of our work.

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

# Appendix

## A    COMPUTE $\sigma$ FOR FIGURE 3

When $\delta$ is given, for each class $C_k$, we construct an auxiliary graph $G_k$ whose nodes correspond to the samples of $C_k$ and edge $(\boldsymbol{x}_1, \boldsymbol{x}_2)$ exists if $d_A(\boldsymbol{x}_1, \boldsymbol{x}_2) \leq \delta$. According to Definition 1, we can compute the main part of $C_k$ by finding the maximum clique of graph $G_k$. Then $\sigma$ can be estimated by $\min_{k \in [K]} |\text{MAXCLIQUE}(G_k)|/|C_k|$. We solve MAXCLIQUE via its dual problem – vertex cover, and adopt the Approx-Vertex-Cover (Papadimitriou & Steiglitz, 1998) to compute the solution.

## B    PROOFS FOR SECTION 3

**Lemma 3.1.** *For a $(\sigma, \delta)$-augmentation with main part $C_k^0$ of each class $C_k$, if all samples belonging to $(C_1^0 \cup \cdots \cup C_K^0) \cap S_\varepsilon$ can be correctly classified by a classifier $G$, then its classification error rate $\text{Err}(G)$ is upper bounded by $(1 - \sigma) + R_\varepsilon$.*

*Proof.* Since every sample $\boldsymbol{x} \in (C_1^0 \cup \cdots \cup C_K^0) \cap S_\varepsilon$ can be correctly classified by $G$, then the classification error rate

$$
\begin{aligned}
\text{Err}(G) &= \sum_{k=1}^{K} \mathbb{P}[G(\boldsymbol{x}) \neq k, \forall \boldsymbol{x} \in C_k] \\
&\leq \mathbb{P}\left[\overline{(C_1^0 \cup \cdots \cup C_K^0) \cap S_\varepsilon}\right] \\
&= \mathbb{P}\left[\overline{C_1^0 \cup \cdots \cup C_K^0} \cup \overline{S_\varepsilon}\right] \\
&\leq (1 - \sigma) + \mathbb{P}\left[\overline{S_\varepsilon}\right] \\
&= (1 - \sigma) + R_\varepsilon.
\end{aligned}
$$

This finishes the proof. $\square$

**Lemma 3.2.** *Given a $(\sigma, \delta)$-augmentation used in contrastive SSL, for any $\ell \in [K]$, if $\mu_\ell^\top \mu_k < r^2 \left(1 - \rho_\ell(\sigma, \delta, \varepsilon) - \sqrt{2\rho_\ell(\sigma, \delta, \varepsilon)} - \frac{\Delta_\mu}{2}\right)$ holds for all $k \neq \ell$, then every sample $\boldsymbol{x} \in C_\ell^0 \cap S_\varepsilon$ can be correctly classified by the NN classifier $G_f$, where $\rho_\ell(\sigma, \delta, \varepsilon) = 2(1 - \sigma) + \frac{R_\varepsilon}{p_\ell} + \sigma \left(\frac{L\delta}{r} + \frac{2\varepsilon}{r}\right)$ and $\Delta_\mu = 1 - \min_{k \in [K]} \|\mu_k\|^2 / r^2$.*

*Proof.* Without loss of generality, we consider $\ell = 1$. To show that every sample $\boldsymbol{x}_0 \in C_1^0 \cap S_\varepsilon$ can be correctly classified by $G_f$, we need to prove that for all $k \neq 1$, $\|f(\boldsymbol{x}_0) - \mu_1\| < \|f(\boldsymbol{x}_0) - \mu_k\|$. It is equivalent to prove that

$$
f(\boldsymbol{x}_0)^\top \mu_1 - f(\boldsymbol{x}_0)^\top \mu_k - \left(\frac{1}{2} \|\mu_1\|^2 - \frac{1}{2} \|\mu_k\|^2\right) > 0. \tag{6}
$$

Let $\tilde{f}(\boldsymbol{x}) := \mathbb{E}_{\boldsymbol{x}' \in A(\boldsymbol{x})}[f(\boldsymbol{x}')]$. Then $\|\tilde{f}(\boldsymbol{x})\| = \|\mathbb{E}_{\boldsymbol{x}' \in A(\boldsymbol{x})}[f(\boldsymbol{x}')]\| \leq \mathbb{E}_{\boldsymbol{x}' \in A(\boldsymbol{x})}[\|f(\boldsymbol{x}')\|] = r$.

On the one hand,

$$
\begin{aligned}
f(\boldsymbol{x}_0)^\top \mu_1 &= \frac{1}{p_1} f(\boldsymbol{x}_0)^\top \mathbb{E}_{\boldsymbol{x}}[\tilde{f}(\boldsymbol{x})\mathbb{I}(\boldsymbol{x} \in C_1)] \\
&= \frac{1}{p_1} f(\boldsymbol{x}_0)^\top \mathbb{E}_{\boldsymbol{x}}[\tilde{f}(\boldsymbol{x})\mathbb{I}(\boldsymbol{x} \in C_1 \cap C_1^0 \cap S_\varepsilon)] + \frac{1}{p_1} f(\boldsymbol{x}_0)^\top \mathbb{E}_{\boldsymbol{x}}\left[\tilde{f}(\boldsymbol{x})\mathbb{I}(\boldsymbol{x} \in C_1 \cap \overline{C_1^0 \cap S_\varepsilon})\right] \\
&= \frac{\mathbb{P}[C_1^0 \cap S_\varepsilon]}{p_1} f(\boldsymbol{x}_0)^\top \mathbb{E}_{\boldsymbol{x} \in C_1^0 \cap S_\varepsilon}[\tilde{f}(\boldsymbol{x})] + \frac{1}{p_1} \mathbb{E}_{\boldsymbol{x}}\left[f(\boldsymbol{x}_0)^\top \tilde{f}(\boldsymbol{x}) \cdot \mathbb{I}(\boldsymbol{x} \in C_1 \setminus C_1^0 \cap S_\varepsilon)\right] \\
&\geq \frac{\mathbb{P}[C_1^0 \cap S_\varepsilon]}{p_1} f(\boldsymbol{x}_0)^\top \mathbb{E}_{\boldsymbol{x} \in C_1^0 \cap S_\varepsilon}[\tilde{f}(\boldsymbol{x})] - \frac{r^2}{p_1} \mathbb{P}[C_1 \setminus C_1^0 \cap S_\varepsilon], \tag{7}
\end{aligned}
$$

where $\mathbb{I}(\cdot)$ is the indicator function. Note that

$$\mathbb{P}[C_1 \setminus C_1^0 \cap S_\varepsilon] = \mathbb{P}[(C_1 \setminus C_1^0) \cup (C_1^0 \cap \overline{S_\varepsilon})] \leq (1-\sigma)p_1 + R_\varepsilon, \tag{8}$$

and

$$\mathbb{P}[C_1^0 \cap S_\varepsilon] = \mathbb{P}[C_1] - \mathbb{P}[C_1 \setminus C_1^0 \cap S_\varepsilon] \geq p_1 - ((1-\sigma)p_1 + R_\varepsilon) = \sigma p_1 - R_\varepsilon. \tag{9}$$

Plugging to (7), we have

$$\begin{aligned}
f(\boldsymbol{x}_0)^\top \mu_1 &\geq \frac{\mathbb{P}[C_1^0 \cap S_\varepsilon]}{p_1} f(\boldsymbol{x}_0)^\top \mathop{\mathbb{E}}_{\boldsymbol{x} \in C_1^0 \cap S_\varepsilon} [\tilde{f}(\boldsymbol{x})] - \frac{r^2}{p_1} \mathbb{P}[C_1 \setminus C_1^0 \cap S_\varepsilon] \\
&\geq \left( \sigma - \frac{R_\varepsilon}{p_1} \right) f(\boldsymbol{x}_0)^\top \mathop{\mathbb{E}}_{\boldsymbol{x} \in C_1^0 \cap S_\varepsilon} [\tilde{f}(\boldsymbol{x})] - r^2 \left( 1 - \sigma + \frac{R_\varepsilon}{p_1} \right).
\end{aligned} \tag{10}$$

Notice that $\boldsymbol{x}_0 \in C_1^0 \cap S_\varepsilon$. For any $\boldsymbol{x} \in C_1^0 \cap S_\varepsilon$, we have $d_A(\boldsymbol{x}_0, \boldsymbol{x}) \leq \delta$. Let $(\boldsymbol{x}_0^*, \boldsymbol{x}^*) = \arg\min_{\boldsymbol{x}_0' \in A(\boldsymbol{x}_0), \boldsymbol{x}' \in A(\boldsymbol{x})} \|\boldsymbol{x}_0' - \boldsymbol{x}'\|$. We have $\|\boldsymbol{x}_0^* - \boldsymbol{x}^*\| \leq \delta$. Since $f$ is $L$-Lipschitz continuous, we have $\|f(\boldsymbol{x}_0^*) - f(\boldsymbol{x}^*)\| \leq L \cdot \|\boldsymbol{x}_0^* - \boldsymbol{x}^*\| \leq L\delta$. Since $\boldsymbol{x} \in S_\varepsilon$, for any $\boldsymbol{x}' \in A(\boldsymbol{x})$, $\|f(\boldsymbol{x}') - f(\boldsymbol{x}^*)\| \leq \varepsilon$. Similarly, since $\boldsymbol{x}_0 \in S_\varepsilon$ and $\boldsymbol{x}_0, \boldsymbol{x}_0^* \in A(\boldsymbol{x}_0)$, we have $\|f(\boldsymbol{x}_0) - f(\boldsymbol{x}_0^*)\| \leq \varepsilon$.

The first term of (10) can be bounded by

$$\begin{aligned}
& f(\boldsymbol{x}_0)^\top \mathop{\mathbb{E}}_{\boldsymbol{x} \in C_1^0 \cap S_\varepsilon} [\tilde{f}(\boldsymbol{x})] \\
&= \mathop{\mathbb{E}}_{\boldsymbol{x} \in C_1^0 \cap S_\varepsilon} \mathop{\mathbb{E}}_{\boldsymbol{x}' \in A(\boldsymbol{x})} [f(\boldsymbol{x}_0)^\top f(\boldsymbol{x}')] \\
&= \mathop{\mathbb{E}}_{\boldsymbol{x} \in C_1^0 \cap S_\varepsilon} \mathop{\mathbb{E}}_{\boldsymbol{x}' \in A(\boldsymbol{x})} [f(\boldsymbol{x}_0)^\top (f(\boldsymbol{x}') - f(\boldsymbol{x}_0) + f(\boldsymbol{x}_0))] \\
&= r^2 + \mathop{\mathbb{E}}_{\boldsymbol{x} \in C_1^0 \cap S_\varepsilon} \mathop{\mathbb{E}}_{\boldsymbol{x}' \in A(\boldsymbol{x})} [f(\boldsymbol{x}_0)^\top (f(\boldsymbol{x}') - f(\boldsymbol{x}_0))] \\
&= r^2 + \mathop{\mathbb{E}}_{\boldsymbol{x} \in C_1^0 \cap S_\varepsilon} \mathop{\mathbb{E}}_{\boldsymbol{x}' \in A(\boldsymbol{x})} [f(\boldsymbol{x}_0)^\top (\underbrace{f(\boldsymbol{x}') - f(\boldsymbol{x}^*)}_{\|\cdot\| \leq \varepsilon} + \underbrace{f(\boldsymbol{x}^*) - f(\boldsymbol{x}_0^*)}_{\|\cdot\| \leq L\delta} + \underbrace{f(\boldsymbol{x}_0^*) - f(\boldsymbol{x}_0)}_{\|\cdot\| \leq \varepsilon})] \\
&\geq r^2 - [r\varepsilon + rL\delta + r\varepsilon] \\
&= r^2 - r(L\delta + 2\varepsilon).
\end{aligned}$$

Therefore, (10) turns to

$$\begin{aligned}
f(\boldsymbol{x}_0)^\top \mu_1 &\geq \left( \sigma - \frac{R_\varepsilon}{p_1} \right) f(\boldsymbol{x}_0)^\top \mathop{\mathbb{E}}_{\boldsymbol{x} \in C_1^0 \cap S_\varepsilon} [\tilde{f}(\boldsymbol{x})] - r^2 \left( 1 - \sigma + \frac{R_\varepsilon}{p_1} \right) \\
&\geq \left( \sigma - \frac{R_\varepsilon}{p_1} \right) (r^2 - r(L\delta + 2\varepsilon)) - r^2 \left( 1 - \sigma + \frac{R_\varepsilon}{p_1} \right) \\
&= r^2 \left( (2\sigma - 1) - \frac{R_\varepsilon}{p_1} - \left( \sigma - \frac{R_\varepsilon}{p_1} \right) \left( \frac{L\delta}{r} + \frac{2\varepsilon}{r} \right) \right) \\
&= r^2 \left( 1 - 2(1-\sigma) - \frac{R_\varepsilon}{p_1} - \left( \sigma - \frac{R_\varepsilon}{p_1} \right) \left( \frac{L\delta}{r} + \frac{2\varepsilon}{r} \right) \right) \\
&= r^2(1 - \rho_1(\sigma, \delta, \varepsilon)).
\end{aligned} \tag{11}$$

On the other hand,

$$\begin{aligned}
f(\boldsymbol{x}_0)^\top \mu_k &= (f(\boldsymbol{x}_0) - \mu_1)^\top \mu_k + \mu_1^\top \mu_k \\
&\leq \|f(\boldsymbol{x}_0) - \mu_1\| \cdot \|\mu_k\| + \mu_1^\top \mu_k \\
&\leq r\sqrt{\|f(\boldsymbol{x}_0)\|^2 - 2f(\boldsymbol{x}_0)^\top \mu_1 + \|\mu_1\|^2} + \mu_1^\top \mu_k \\
&\leq r\sqrt{2r^2 - 2f(\boldsymbol{x}_0)^\top \mu_1} + \mu_1^\top \mu_k \\
&\leq \sqrt{2\rho_1(\sigma, \delta, \varepsilon)}r^2 + \mu_1^\top \mu_k.
\end{aligned} \tag{12}$$

Note that $\Delta_\mu = 1 - \min_k \|\mu_k\|^2 / r^2$, the LHS of (6) is

$$f(\boldsymbol{x}_0)^\top \mu_1 - f(\boldsymbol{x}_0)^\top \mu_k - \left( \frac{1}{2} \|\mu_1\|^2 - \frac{1}{2} \|\mu_k\|^2 \right)$$

$$\geq f(\boldsymbol{x}_0)^\top \mu_1 - f(\boldsymbol{x}_0)^\top \mu_k - \frac{1}{2} r^2 \Delta_\mu$$

$$\geq r^2 (1 - \rho_1(\sigma, \delta, \varepsilon)) - \sqrt{2\rho_1(\sigma, \delta, \varepsilon)} r^2 - \mu_1^\top \mu_k - \frac{1}{2} r^2 \Delta_\mu$$

$$= r^2 \left( 1 - \rho_1(\sigma, \delta, \varepsilon) - \sqrt{2\rho_1(\sigma, \delta, \varepsilon)} - \frac{1}{2} \Delta_\mu \right) - \mu_1^\top \mu_k > 0,$$

where the second ieequality is due to (11) and (12). This finishes the proof. □

**Theorem 1.** *Given a $(\sigma, \delta)$-augmentation used in contrastive SSL, if*

$$\mu_\ell^\top \mu_k < r^2 \left( 1 - \rho_{max}(\sigma, \delta, \varepsilon) - \sqrt{2\rho_{max}(\sigma, \delta, \varepsilon)} - \frac{\Delta_\mu}{2} \right) \tag{2}$$

*holds for any pair of $(\ell, k)$ with $\ell \neq k$, then the downstream error rate of NN classifier $G_f$*

$$\mathrm{Err}(G_f) \leq (1 - \sigma) + R_\varepsilon, \tag{3}$$

*where $\rho_{max}(\sigma, \delta, \varepsilon) = 2(1-\sigma) + \frac{R_\varepsilon}{\min_\ell p_\ell} + \sigma \left( \frac{L\delta}{r} + \frac{2\varepsilon}{r} \right)$ and $\Delta_\mu = 1 - \min_{k \in [K]} \|\mu_k\|^2 / r^2$.*

*Proof.* Since the augmentation $A$ is $(\sigma, \delta)$-augmented, there exists a main part $C_k^0$ for each class $C_k$ such that $\mathbb{P}[C_k^0] \geq \sigma p_k$ and $\sup_{\boldsymbol{x}_1, \boldsymbol{x}_2 \in C_k^0} d_A(\boldsymbol{x}_1, \boldsymbol{x}_2) \leq \delta$. Since for any $\ell \neq k$, we have $\mu_\ell^\top \mu_k < r^2 \left( 1 - \rho_{max}(\sigma, \delta, \varepsilon) - \sqrt{2\rho_{max}(\sigma, \delta, \varepsilon)} - \frac{\Delta_\mu}{2} \right) \leq r^2 \left( 1 - \rho_\ell(\sigma, \delta, \varepsilon) - \sqrt{2\rho_\ell(\sigma, \delta, \varepsilon)} - \frac{\Delta_\mu}{2} \right)$. According to Lemma 3.2, every sample $\boldsymbol{x} \in C_\ell^0 \cap S_\varepsilon$ can be correctly classified by $G_f$. Therefore, every sample $\boldsymbol{x} \in (C_1^0 \cap \cdots \cap C_K^0) \cap S_\varepsilon$ can be correctly classified by $G_f$. According to Lemma 3.1, the error rate $\mathrm{Err}(G_f) \leq 1 - \sigma + R_\varepsilon$. □

**Theorem 2.** *If encoder $f$ is $L$-Lipschitz continuous, then*

$$R_\varepsilon^2 \leq \eta(\varepsilon)^2 \cdot \mathbb{E}_{\boldsymbol{x}} \mathbb{E}_{\boldsymbol{x}_1, \boldsymbol{x}_2 \in A(\boldsymbol{x})} \|f(\boldsymbol{x}_1) - f(\boldsymbol{x}_2)\|^2 = \eta(\varepsilon)^2 \cdot \mathcal{L}_{\mathrm{align}}(f),$$

*where $\eta(\varepsilon) = \inf_{h \in \left( 0, \frac{\varepsilon}{2\sqrt{n}LM} \right)} \frac{4 \max\{1, m^2 h^{2n}\}}{h^{2n}(\varepsilon - 2\sqrt{n}LMh)}$.*

*Proof.* The parameter space $[0,1]^n$ of $\theta$ can be separated to cubes $\Theta_1, \ldots, \Theta_{m'}$ where $m' = 1/h^n$ and each cube's edge length is $h \in (0, \frac{\varepsilon}{2\sqrt{n}LM})$. Then for any given $\boldsymbol{x}$, we have

$$\mathbb{E}_{\boldsymbol{x}_1, \boldsymbol{x}_2 \in A(\boldsymbol{x})} \|f(\boldsymbol{x}_1) - f(\boldsymbol{x}_2)\| = \underbrace{\frac{1}{4m^2} \sum_{\gamma=1}^m \sum_{\beta=1}^m \|f(A_\gamma(\boldsymbol{x})) - f(A_\beta(\boldsymbol{x}))\|}_{\Lambda_1}$$

$$+ \underbrace{\frac{1}{2mm'} \sum_{\gamma=1}^m \sum_{j=1}^{m'} \int_{\Theta_j} \frac{1}{h^n} \|f(A_\gamma(\boldsymbol{x})) - f(A_\theta(\boldsymbol{x}))\| \, d\theta}_{\Lambda_2}$$

$$+ \underbrace{\frac{1}{4m'^2} \sum_{i=1}^{m'} \sum_{j=1}^{m'} \int_{\Theta_i} \int_{\Theta_j} \frac{1}{h^{2n}} \|f(A_{\theta_1}(\boldsymbol{x})) - f(A_{\theta_2}(\boldsymbol{x}))\| \, d\theta_2 \, d\theta_1}_{\Lambda_3}.$$

By Cauchy-Schwarz inequality,

$$\forall \theta, \|f(A_\gamma(\boldsymbol{x})) - f(A_{\theta'}(\boldsymbol{x}))\| \leq \|f(A_\gamma(\boldsymbol{x})) - f(A_\theta(\boldsymbol{x}))\| + \|f(A_\theta(\boldsymbol{x})) - f(A_{\theta'}(\boldsymbol{x}))\|.$$

Then for any given $\theta$,

$$\sup_{\theta'} \|f(A_\gamma(\boldsymbol{x})) - f(A_{\theta'}(\boldsymbol{x}))\| \leq \|f(A_\gamma(\boldsymbol{x})) - f(A_\theta(\boldsymbol{x}))\| + \sup_\theta \|f(A_\theta(\boldsymbol{x})) - f(A_{\theta'}(\boldsymbol{x}))\|$$

$$\leq \|f(A_\gamma(\boldsymbol{x})) - f(A_\theta(\boldsymbol{x}))\| + \sup_{\theta_1,\theta_2} \|f(A_{\theta_1}(\boldsymbol{x})) - f(A_{\theta_2}(\boldsymbol{x}))\|.$$

Therefore, for any $\gamma \in [m], j \in [m']$, we have

$$\sup_{\theta' \in \Theta_j} \|f(A_\gamma(\boldsymbol{x})) - f(A_{\theta'}(\boldsymbol{x}))\|$$

$$= \int_{\Theta_j} \frac{1}{h^n} \sup_{\theta' \in \Theta_j} \|f(A_\gamma(\boldsymbol{x})) - f(A_{\theta'}(\boldsymbol{x}))\| \, d\theta$$

$$\leq \int_{\Theta_j} \frac{1}{h^n} \|f(A_\gamma(\boldsymbol{x})) - f(A_\theta(\boldsymbol{x}))\| \, d\theta + \sup_{\theta_1,\theta_2 \in \Theta_j} \|f(A_{\theta_1}(\boldsymbol{x})) - f(A_{\theta_2}(\boldsymbol{x}))\|$$

$$\leq \int_{\Theta_j} \frac{1}{h^n} \|f(A_\gamma(\boldsymbol{x})) - f(A_\theta(\boldsymbol{x}))\| \, d\theta + L \sup_{\theta_1,\theta_2 \in \Theta_j} \|A_{\theta_1}(\boldsymbol{x}) - A_{\theta_2}(\boldsymbol{x})\|$$

$$\leq \int_{\Theta_j} \frac{1}{h^n} \|f(A_\gamma(\boldsymbol{x})) - f(A_\theta(\boldsymbol{x}))\| \, d\theta + LM \sup_{\theta_1,\theta_2 \in \Theta_j} \|\theta_1 - \theta_2\|$$

$$= \int_{\Theta_j} \frac{1}{h^n} \|f(A_\gamma(\boldsymbol{x})) - f(A_\theta(\boldsymbol{x}))\| \, d\theta + LM\sqrt{n}h$$

$$= \int_{\Theta_j} \frac{1}{h^n} \|f(A_\gamma(\boldsymbol{x})) - f(A_\theta(\boldsymbol{x}))\| \, d\theta + \sqrt{n}LMh.$$

Similarly, we can obtain

$$\sup_{\theta \in \Theta_i, \theta' \in \Theta_j} \|f(A_\theta(\boldsymbol{x})) - f(A_{\theta'}(\boldsymbol{x}))\|$$

$$= \int_{\Theta_i} \int_{\Theta_j} \frac{1}{h^{2n}} \sup_{\theta \in \Theta_i, \theta' \in \Theta_j} \|f(A_\theta(\boldsymbol{x})) - f(A_{\theta'}(\boldsymbol{x}))\| \, d\theta_2 \, d\theta_1$$

$$\leq \int_{\Theta_i} \int_{\Theta_j} \frac{1}{h^{2n}} \|f(A_{\theta_1}(\boldsymbol{x})) - f(A_{\theta_2}(\boldsymbol{x}))\| \, d\theta_2 \, d\theta_1$$

$$+ \int_{\Theta_i} \int_{\Theta_j} \frac{1}{h^{2n}} \sup_{\theta \in \Theta_i} \|f(A_\theta(\boldsymbol{x})) - f(A_{\theta_1}(\boldsymbol{x}))\| \, d\theta_2 \, d\theta_1$$

$$+ \int_{\Theta_i} \int_{\Theta_j} \frac{1}{h^{2n}} \sup_{\theta' \in \Theta_j} \|f(A_{\theta_2}(\boldsymbol{x})) - f(A_{\theta'}(\boldsymbol{x}))\| \, d\theta_2 \, d\theta_1$$

$$\leq \int_{\Theta_i} \int_{\Theta_j} \frac{1}{h^{2n}} \|f(A_{\theta_1}(\boldsymbol{x})) - f(A_{\theta_2}(\boldsymbol{x}))\| \, d\theta_2 \, d\theta_1$$

$$+ \sup_{\theta,\theta' \in \Theta_i} \|f(A_\theta(\boldsymbol{x})) - f(A_{\theta'}(\boldsymbol{x}))\| + \sup_{\theta,\theta' \in \Theta_j} \|f(A_\theta(\boldsymbol{x})) - f(A_{\theta'}(\boldsymbol{x}))\|$$

$$\leq \int_{\Theta_i} \int_{\Theta_j} \frac{1}{h^{2n}} \|f(A_{\theta_1}(\boldsymbol{x})) - f(A_{\theta_2}(\boldsymbol{x}))\| \, d\theta_2 \, d\theta_1 + 2\sqrt{n}LMh.$$

Therefore,

$$\sup_{\boldsymbol{x}_1,\boldsymbol{x}_2 \in A(\boldsymbol{x})} \|f(\boldsymbol{x}_1) - f(\boldsymbol{x}_2)\|$$

$$= \max \left\{ \begin{array}{c} \sup_{\gamma,\beta \in [m]} \|f(A_\gamma(\boldsymbol{x})) - f(A_\beta(\boldsymbol{x}))\| \\ \sup_{\gamma \in [m], j \in [m']} \sup_{\theta' \in \Theta_j} \|f(A_\gamma(\boldsymbol{x})) - f(A_{\theta'}(\boldsymbol{x}))\| \\ \sup_{i,j \in [m']} \sup_{\theta \in \Theta_i, \theta' \in \Theta_j} \|f(A_\theta(\boldsymbol{x})) - f(A_{\theta'}(\boldsymbol{x}))\| \end{array} \right\}$$

$$\leq \max \left\{ \begin{array}{c} \sup_{\gamma,\beta \in [m]} \|f(A_\gamma(\boldsymbol{x})) - f(A_\beta(\boldsymbol{x}))\| \\ \sup_{\gamma \in [m], j \in [m']} \int_{\Theta_j} \frac{1}{h^n} \|f(A_\gamma(\boldsymbol{x})) - f(A_\theta(\boldsymbol{x}))\| \, d\theta + \sqrt{n}LMh \\ \sup_{i,j \in [m']} \int_{\Theta_i} \int_{\Theta_j} \frac{1}{h^{2n}} \|f(A_{\theta_1}(\boldsymbol{x})) - f(A_{\theta_2}(\boldsymbol{x}))\| \, d\theta_2 \, d\theta_1 + 2\sqrt{n}LMh \end{array} \right\}$$

$$\leq \max \left\{ \begin{array}{c} \sum_{\gamma=1}^{m} \sum_{\beta=1}^{m} \|f(A_\gamma(\boldsymbol{x})) - f(A_\beta(\boldsymbol{x}))\| \\ \sum_{\gamma=1}^{m} \sum_{j=1}^{m'} \int_{\Theta_j} \frac{1}{h^n} \|f(A_\gamma(\boldsymbol{x})) - f(A_\theta(\boldsymbol{x}))\| \, d\theta + \sqrt{n} LMh \\ \sum_{i=1}^{m'} \sum_{j=1}^{m'} \int_{\Theta_i} \int_{\Theta_j} \frac{1}{h^{2n}} \|f(A_{\theta_1}(\boldsymbol{x})) - f(A_{\theta_2}(\boldsymbol{x}))\| \, d\theta_2 \, d\theta_1 + 2\sqrt{n} LMh \end{array} \right\}$$

$$\leq \max \left\{ \begin{array}{c} 4m^2 \Lambda_1 \\ 2mm' \Lambda_2 + \sqrt{n} LMh \\ 4m'^2 \Lambda_3 + 2\sqrt{n} LMh \end{array} \right\}$$

$$\leq \max \left\{ \begin{array}{c} 4m^2 \Lambda_1 \\ 2mm' \Lambda_2 \\ 4m'^2 \Lambda_3 \end{array} \right\} + 2\sqrt{n} LMh$$

$$\leq \max\{4m^2, 2mm', 4m'^2\}(\Lambda_1 + \Lambda_2 + \Lambda_3) + 2\sqrt{n} LMh$$

$$= \max\{4m^2, 2mm', 4m'^2\} \mathop{\mathbb{E}}_{\boldsymbol{x}_1, \boldsymbol{x}_2 \in A(\boldsymbol{x})} \|f(\boldsymbol{x}_1) - f(\boldsymbol{x}_2)\| + 2\sqrt{n} LMh.$$

Thus, the following set $S$ is a subset of $S_\varepsilon$:

$$S = \left\{ \boldsymbol{x} : \mathop{\mathbb{E}}_{\boldsymbol{x}_1, \boldsymbol{x}_2 \in A(\boldsymbol{x})} \|f(\boldsymbol{x}_1) - f(\boldsymbol{x}_2)\| \leq \frac{\varepsilon - 2\sqrt{n} LMh}{\max\{4m^2, 2mm', 4m'^2\}} \right\} \subseteq S_\varepsilon.$$

Then by Markov's inequality, we have

$$R_\varepsilon = \mathbb{P}\left[\overline{S_\varepsilon}\right] \leq \mathbb{P}\left[\overline{S}\right]$$
$$\leq \frac{\mathbb{E}_{\boldsymbol{x}} \mathbb{E}_{\boldsymbol{x}_1, \boldsymbol{x}_2 \in A(\boldsymbol{x})} \|f(\boldsymbol{x}_1) - f(\boldsymbol{x}_2)\|}{\frac{\varepsilon - 2\sqrt{n} LMh}{\max\{4m^2, 2mm', 4m'^2\}}}$$
$$= \frac{\max\{4, 2mh^n, 4m^2 h^{2n}\}}{h^{2n}(\varepsilon - 2\sqrt{n} LMh)} \mathop{\mathbb{E}}_{\boldsymbol{x}} \mathop{\mathbb{E}}_{\boldsymbol{x}_1, \boldsymbol{x}_2 \in A(\boldsymbol{x})} \|f(\boldsymbol{x}_1) - f(\boldsymbol{x}_2)\|$$
$$= \frac{4\max\{1, m^2 h^{2n}\}}{h^{2n}(\varepsilon - 2\sqrt{n} LMh)} \mathop{\mathbb{E}}_{\boldsymbol{x}} \mathop{\mathbb{E}}_{\boldsymbol{x}_1, \boldsymbol{x}_2 \in A(\boldsymbol{x})} \|f(\boldsymbol{x}_1) - f(\boldsymbol{x}_2)\|.$$

The above inequality holds for all $h \in (0, \frac{\varepsilon}{2\sqrt{n} LM})$, thus

$$R_\varepsilon \leq \inf_{0 < h < \frac{\varepsilon}{2\sqrt{n} LM}} \frac{4\max\{1, m^2 h^{2n}\}}{h^{2n}(\varepsilon - 2\sqrt{n} LMh)} \mathop{\mathbb{E}}_{\boldsymbol{x}} \mathop{\mathbb{E}}_{\boldsymbol{x}_1, \boldsymbol{x}_2 \in A(\boldsymbol{x})} \|f(\boldsymbol{x}_1) - f(\boldsymbol{x}_2)\| = \eta(\varepsilon) \cdot \mathop{\mathbb{E}}_{\boldsymbol{x}} \mathop{\mathbb{E}}_{\boldsymbol{x}_1, \boldsymbol{x}_2 \in A(\boldsymbol{x})} \|f(\boldsymbol{x}_1) - f(\boldsymbol{x}_2)\|.$$

Therefore, we have

$$R_\varepsilon^2 \leq \eta(\varepsilon)^2 \cdot (\mathop{\mathbb{E}}_{\boldsymbol{x}} \mathop{\mathbb{E}}_{\boldsymbol{x}_1, \boldsymbol{x}_2 \in A(\boldsymbol{x})} \|f(\boldsymbol{x}_1) - f(\boldsymbol{x}_2)\|)^2 \leq \eta(\varepsilon)^2 \cdot \mathop{\mathbb{E}}_{\boldsymbol{x}} \mathop{\mathbb{E}}_{\boldsymbol{x}_1, \boldsymbol{x}_2 \in A(\boldsymbol{x})} \|f(\boldsymbol{x}_1) - f(\boldsymbol{x}_2)\|^2.$$

This finishes the proof. $\qquad \square$

## C  PROOFS FOR SECTION 4

Before providing our proofs, we give the following useful lemma, which upper bounds the first and second order moment of intra-difference within each class $C_k$ via $\varepsilon$ and $R_\varepsilon$.

**Lemma C.1.** *Suppose that $\|f(\boldsymbol{x})\| = r$ for every $\boldsymbol{x}$. For each $k \in [K]$,*

$$\mathop{\mathbb{E}}_{\boldsymbol{x} \in C_k} \mathop{\mathbb{E}}_{\boldsymbol{x}_1 \in A(\boldsymbol{x})} \|f(\boldsymbol{x}_1) - \mu_k\| \leq 4r \left( 1 - \sigma \left( 1 - \frac{\varepsilon}{2r} - \frac{L\delta}{4r} \right) + \frac{R_\varepsilon}{p_k} \right),$$

*and*

$$\mathop{\mathbb{E}}_{\boldsymbol{x} \in C_k} \mathop{\mathbb{E}}_{\boldsymbol{x}_1 \in A(\boldsymbol{x})} \|f(\boldsymbol{x}_1) - \mu_k\|^2 \leq 4r^2 \left[ \left( 1 - \sigma + \frac{L}{2r}\delta \right) + \left( \frac{\varepsilon}{r} + \frac{R_\varepsilon}{p_k} \right) \right]^2 + 4r^2 \left( 1 - \sigma + \frac{R_\varepsilon}{p_k} \right).$$

*Proof.* For each $k \in [K]$,

$$\mathop{\mathbb{E}}_{\boldsymbol{x} \in C_k} \mathop{\mathbb{E}}_{\boldsymbol{x}_1 \in A(\boldsymbol{x})} \|f(\boldsymbol{x}_1) - \mu_k\|$$

$$= \frac{1}{p_k} \mathop{\mathbb{E}}_{\boldsymbol{x}} \mathop{\mathbb{E}}_{\boldsymbol{x}_1 \in A(\boldsymbol{x})} [\mathbb{I}(\boldsymbol{x} \in C_k) \|f(\boldsymbol{x}_1) - \mu_k\|]$$

$$= \frac{1}{p_k} \mathop{\mathbb{E}}_{\boldsymbol{x}} \mathop{\mathbb{E}}_{\boldsymbol{x}_1 \in A(\boldsymbol{x})} [\mathbb{I}(\boldsymbol{x} \in C_k^0 \cap S_\varepsilon) \|f(\boldsymbol{x}_1) - \mu_k\|] + \frac{1}{p_k} \mathop{\mathbb{E}}_{\boldsymbol{x}} \mathop{\mathbb{E}}_{\boldsymbol{x}_1 \in A(\boldsymbol{x})} [\mathbb{I}(\boldsymbol{x} \in C_k \setminus C_k^0 \cap S_\varepsilon) \|f(\boldsymbol{x}_1) - \mu_k\|]$$

$$\leq \frac{1}{p_k} \mathop{\mathbb{E}}_{\boldsymbol{x}} \mathop{\mathbb{E}}_{\boldsymbol{x}_1 \in A(\boldsymbol{x})} [\mathbb{I}(\boldsymbol{x} \in C_k^0 \cap S_\varepsilon) \|f(\boldsymbol{x}_1) - \mu_k\|] + \frac{2r\mathbb{P}[C_k \setminus C_k^0 \cap S_\varepsilon]}{p_k}$$

$$\leq \frac{1}{p_k} \mathop{\mathbb{E}}_{\boldsymbol{x}} \mathop{\mathbb{E}}_{\boldsymbol{x}_1 \in A(\boldsymbol{x})} [\mathbb{I}(\boldsymbol{x} \in C_k^0 \cap S_\varepsilon) \|f(\boldsymbol{x}_1) - \mu_k\|] + 2r \left(1 - \sigma + \frac{R_\varepsilon}{p_k}\right) \qquad \text{(using (8))}$$

$$\leq \frac{\mathbb{P}[C_k^0 \cap S_\varepsilon]}{p_k} \mathop{\mathbb{E}}_{\boldsymbol{x} \in C_k^0 \cap S_\varepsilon} \mathop{\mathbb{E}}_{\boldsymbol{x}_1 \in A(\boldsymbol{x})} \|f(\boldsymbol{x}_1) - \mu_k\| + 2r \left(1 - \sigma + \frac{R_\varepsilon}{p_k}\right)$$

$$\leq \mathop{\mathbb{E}}_{\boldsymbol{x} \in C_k^0 \cap S_\varepsilon} \mathop{\mathbb{E}}_{\boldsymbol{x}_1 \in A(\boldsymbol{x})} \|f(\boldsymbol{x}_1) - \mu_k\| + 2r \left(1 - \sigma + \frac{R_\varepsilon}{p_k}\right) \qquad (13)$$

where

$$\mathop{\mathbb{E}}_{\boldsymbol{x} \in C_k^0 \cap S_\varepsilon} \mathop{\mathbb{E}}_{\boldsymbol{x}_1 \in A(\boldsymbol{x})} \|f(\boldsymbol{x}_1) - \mu_k\|$$

$$= \mathop{\mathbb{E}}_{\boldsymbol{x} \in C_k^0 \cap S_\varepsilon} \mathop{\mathbb{E}}_{\boldsymbol{x}_1 \in A(\boldsymbol{x})} \|f(\boldsymbol{x}_1) - \mathop{\mathbb{E}}_{\boldsymbol{x}' \in C_k} \mathop{\mathbb{E}}_{\boldsymbol{x}_2 \in A(\boldsymbol{x}')} f(\boldsymbol{x}_2)\|$$

$$= \mathop{\mathbb{E}}_{\boldsymbol{x} \in C_k^0 \cap S_\varepsilon} \mathop{\mathbb{E}}_{\boldsymbol{x}_1 \in A(\boldsymbol{x})} \left\| f(\boldsymbol{x}_1) - \frac{\mathbb{P}[C_k^0 \cap S_\varepsilon]}{p_k} \mathop{\mathbb{E}}_{\boldsymbol{x}' \in C_k^0 \cap S_\varepsilon} \mathop{\mathbb{E}}_{\boldsymbol{x}_2 \in A(\boldsymbol{x}')} f(\boldsymbol{x}_2) - \frac{\mathbb{P}[C_k \setminus C_k^0 \cap S_\varepsilon]}{p_k} \mathop{\mathbb{E}}_{\boldsymbol{x}' \in C_k \setminus C_k^0 \cap S_\varepsilon} \mathop{\mathbb{E}}_{\boldsymbol{x}_2 \in A(\boldsymbol{x}')} f(\boldsymbol{x}_2) \right\|$$

$$= \mathop{\mathbb{E}}_{\boldsymbol{x} \in C_k^0 \cap S_\varepsilon} \mathop{\mathbb{E}}_{\boldsymbol{x}_1 \in A(\boldsymbol{x})} \left\| \frac{\mathbb{P}[C_k^0 \cap S_\varepsilon]}{p_k} \left( f(\boldsymbol{x}_1) - \mathop{\mathbb{E}}_{\boldsymbol{x}' \in C_k^0 \cap S_\varepsilon} \mathop{\mathbb{E}}_{\boldsymbol{x}_2 \in A(\boldsymbol{x}')} f(\boldsymbol{x}_2) \right) \right.$$

$$\left. + \frac{\mathbb{P}[C_k \setminus C_k^0 \cap S_\varepsilon]}{p_k} \left( f(\boldsymbol{x}_1) - \mathop{\mathbb{E}}_{\boldsymbol{x}' \in C_k \setminus C_k^0 \cap S_\varepsilon} \mathop{\mathbb{E}}_{\boldsymbol{x}_2 \in A(\boldsymbol{x}')} f(\boldsymbol{x}_2) \right) \right\|$$

$$\leq \frac{\mathbb{P}[C_k^0 \cap S_\varepsilon]}{p_k} \mathop{\mathbb{E}}_{\boldsymbol{x} \in C_k^0 \cap S_\varepsilon} \mathop{\mathbb{E}}_{\boldsymbol{x}_1 \in A(\boldsymbol{x})} \left\| f(\boldsymbol{x}_1) - \mathop{\mathbb{E}}_{\boldsymbol{x}' \in C_k^0 \cap S_\varepsilon} \mathop{\mathbb{E}}_{\boldsymbol{x}_2 \in A(\boldsymbol{x}')} f(\boldsymbol{x}_2) \right\| + \frac{\mathbb{P}[C_k \setminus C_k^0 \cap S_\varepsilon]}{p_k} \cdot 2r$$

$$\leq \sup_{\boldsymbol{x}, \boldsymbol{x}' \in C_k^0 \cap S_\varepsilon} \sup_{\substack{\boldsymbol{x}_1 \in A(\boldsymbol{x}) \\ \boldsymbol{x}_2 \in A(\boldsymbol{x}')}} \|f(\boldsymbol{x}_1) - f(\boldsymbol{x}_2)\| + 2r \left(1 - \sigma + \frac{R_\varepsilon}{p_k}\right). \qquad (14)$$

For any $\boldsymbol{x}, \boldsymbol{x}' \in C_1^0 \cap S_\varepsilon$, we have $d_A(\boldsymbol{x}, \boldsymbol{x}') \leq \delta$. Let $(\boldsymbol{x}_1^*, \boldsymbol{x}_2^*) = \arg \min_{\boldsymbol{x}_1 \in A(\boldsymbol{x}), \boldsymbol{x}_2 \in A(\boldsymbol{x}')} \|\boldsymbol{x}_1 - \boldsymbol{x}_2\|$. We have $\|\boldsymbol{x}_1^* - \boldsymbol{x}_2^*\| \leq \delta$. Since $f$ is $L$-Lipschitz continuous, we have $\|f(\boldsymbol{x}_1^*) - f(\boldsymbol{x}_2^*)\| \leq L \cdot \|\boldsymbol{x}_1^* - \boldsymbol{x}_2^*\| \leq L\delta$. Since $\boldsymbol{x} \in S_\varepsilon$, for any $\boldsymbol{x}_1 \in A(\boldsymbol{x})$, $\|f(\boldsymbol{x}_1) - f(\boldsymbol{x}_1^*)\| \leq \varepsilon$. Similarly, since $\boldsymbol{x}' \in S_\varepsilon$, for any $\boldsymbol{x}_2 \in A(\boldsymbol{x}')$, we have $\|f(\boldsymbol{x}_2) - f(\boldsymbol{x}_2^*)\| \leq \varepsilon$. Therefore, for any $\boldsymbol{x}, \boldsymbol{x}' \in C_1^0 \cap S_\varepsilon$ and $\boldsymbol{x}_1 \in A(\boldsymbol{x}), \boldsymbol{x}_2 \in A(\boldsymbol{x}')$,

$$\|f(\boldsymbol{x}_1) - f(\boldsymbol{x}_2)\| \leq \|f(\boldsymbol{x}_1) - f(\boldsymbol{x}_1^*)\| + \|f(\boldsymbol{x}_1^*) - f(\boldsymbol{x}_2^*)\| + \|f(\boldsymbol{x}_2^*) - f(\boldsymbol{x}_2)\| \leq 2\varepsilon + L\delta.$$

Plugging into (13) and (14), we obtain

$$\mathop{\mathbb{E}}_{\boldsymbol{x} \in C_k} \mathop{\mathbb{E}}_{\boldsymbol{x}_1 \in A(\boldsymbol{x})} \|f(\boldsymbol{x}_1) - \mu_k\| \leq (2\varepsilon + L\delta) + 4r \left(1 - \sigma + \frac{R_\varepsilon}{p_k}\right)$$

$$= 4r \left(1 - \sigma + \frac{L}{4r}\delta\right) + 2 \left(\varepsilon + \frac{2r}{p_k}R_\varepsilon\right).$$

Similar to (13) and (14), we have

$$\mathop{\mathbb{E}}_{\boldsymbol{x} \in C_k} \mathop{\mathbb{E}}_{\boldsymbol{x}_1 \in A(\boldsymbol{x})} \|f(\boldsymbol{x}_1) - \mu_k\|^2 \leq \mathop{\mathbb{E}}_{\boldsymbol{x} \in C_k^0 \cap S_\varepsilon} \mathop{\mathbb{E}}_{\boldsymbol{x}_1 \in A(\boldsymbol{x})} \|f(\boldsymbol{x}_1) - \mu_k\|^2 + 4r^2 \left(1 - \sigma + \frac{R_\varepsilon}{p_k}\right),$$

and

$$\mathop{\mathbb{E}}_{\boldsymbol{x}\in C_k^0\cap S_\varepsilon}\mathop{\mathbb{E}}_{\boldsymbol{x}_1\in A(\boldsymbol{x})}\|f(\boldsymbol{x}_1)-\mu_k\|^2$$

$$=\mathop{\mathbb{E}}_{\boldsymbol{x}\in C_k^0\cap S_\varepsilon}\mathop{\mathbb{E}}_{\boldsymbol{x}_1\in A(\boldsymbol{x})}\left\|\frac{\mathbb{P}[C_k^0\cap S_\varepsilon]}{p_k}\left(f(\boldsymbol{x}_1)-\mathop{\mathbb{E}}_{\boldsymbol{x}'\in C_k^0\cap S_\varepsilon}\mathop{\mathbb{E}}_{\boldsymbol{x}_2\in A(\boldsymbol{x}')}f(\boldsymbol{x}_2)\right)\right.$$

$$\left.+\frac{\mathbb{P}[C_k\setminus C_k^0\cap S_\varepsilon]}{p_k}\left(f(\boldsymbol{x}_1)-\mathop{\mathbb{E}}_{\boldsymbol{x}'\in C_k\setminus C_k^0\cap S_\varepsilon}\mathop{\mathbb{E}}_{\boldsymbol{x}_2\in A(\boldsymbol{x}')}f(\boldsymbol{x}_2)\right)\right\|^2$$

$$\leq\mathop{\mathbb{E}}_{\boldsymbol{x}\in C_k^0\cap S_\varepsilon}\mathop{\mathbb{E}}_{\boldsymbol{x}_1\in A(\boldsymbol{x})}\left[\left\|f(\boldsymbol{x}_1)-\mathop{\mathbb{E}}_{\boldsymbol{x}'\in C_k^0\cap S_\varepsilon}\mathop{\mathbb{E}}_{\boldsymbol{x}_2\in A(\boldsymbol{x}')}f(\boldsymbol{x}_2)\right\|+2r\left(1-\sigma+\frac{R_\varepsilon}{p_k}\right)\right]^2$$

$$\leq\left[(2\varepsilon+L\delta)+2r\left(1-\sigma+\frac{R_\varepsilon}{p_k}\right)\right]^2.$$

Therefore,

$$\mathop{\mathbb{E}}_{\boldsymbol{x}\in C_k}\mathop{\mathbb{E}}_{\boldsymbol{x}_1\in A(\boldsymbol{x})}\|f(\boldsymbol{x}_1)-\mu_k\|^2\leq\left[(2\varepsilon+L\delta)+2r\left(1-\sigma+\frac{R_\varepsilon}{p_k}\right)\right]^2+4r^2\left(1-\sigma+\frac{R_\varepsilon}{p_k}\right)$$

$$=4r^2\left[\left(1-\sigma+\frac{L}{2r}\delta\right)+\left(\frac{\varepsilon}{r}+\frac{R_\varepsilon}{p_k}\right)\right]^2+4r^2\left(1-\sigma+\frac{R_\varepsilon}{p_k}\right).$$

This finishes the proof. □

Now we are ready to give our proofs of theorems.

## C.1 INFONCE LOSS

**Theorem 3.** *Assume that encoder $f$ with norm 1 is $L$-Lipschitz continuous. If the augmented data is $(\sigma,\delta)$-augmented, then for any $\varepsilon\geq 0$ and $k\neq\ell$, we have*

$$\mu_k^\top\mu_\ell\leq\log\left(\exp\left\{\frac{\mathcal{L}_2^{\text{InfoNCE}}(f)+\tau(\sigma,\delta,\varepsilon,R_\varepsilon)}{p_kp_\ell}\right\}-\exp(1-\varepsilon)\right),$$

*where $\tau(\sigma,\delta,\varepsilon,R_\varepsilon)$ is a non-negative term, decreasing with smaller $\varepsilon, R_\varepsilon$ or sharper concentration of augmented data, and $\tau(\sigma,\delta,\varepsilon,R_\varepsilon)=0$ when $\sigma=1,\delta=0,\varepsilon=0,R_\varepsilon=0$.*

*Proof.* Given $\boldsymbol{x}\in S_\varepsilon$, for any $\boldsymbol{x}_1,\boldsymbol{x}_2\in A(\boldsymbol{x})$, we have

$$\log\left(e^{f(\boldsymbol{x}_1)^\top f(\boldsymbol{x}_2)}+e^{f(\boldsymbol{x}_1)^\top f(\boldsymbol{x}^-)}\right)=\log\left(e^{f(\boldsymbol{x}_1)^\top f(\boldsymbol{x}_1)}e^{f(\boldsymbol{x}_1)^\top(f(\boldsymbol{x}_2)-f(\boldsymbol{x}_1))}+e^{f(\boldsymbol{x}_1)^\top f(\boldsymbol{x}^-)}\right)$$

$$\geq\log\left(e^{\|f(\boldsymbol{x}_1)\|^2}e^{-\|f(\boldsymbol{x}_1)\|\cdot\varepsilon}+e^{f(\boldsymbol{x}_1)^\top f(\boldsymbol{x}^-)}\right)$$

$$=\log\left(e^{1-\varepsilon}+e^{f(\boldsymbol{x}_1)^\top f(\boldsymbol{x}^-)}\right).$$

Therefore, we have

$$\mathcal{L}_2^{\text{InfoNCE}}(f)=\mathop{\mathbb{E}}_{\boldsymbol{x},\boldsymbol{x}'}\mathop{\mathbb{E}}_{\substack{\boldsymbol{x}_1,\boldsymbol{x}_2\in A(\boldsymbol{x})\\\boldsymbol{x}^-\in A(\boldsymbol{x}')}}\left[\log\left(e^{f(\boldsymbol{x}_1)^\top f(\boldsymbol{x}_2)}+e^{f(\boldsymbol{x}_1)^\top f(\boldsymbol{x}^-)}\right)\right]$$

$$=\mathop{\mathbb{E}}_{\boldsymbol{x},\boldsymbol{x}'}\mathop{\mathbb{E}}_{\substack{\boldsymbol{x}_1,\boldsymbol{x}_2\in A(\boldsymbol{x})\\\boldsymbol{x}^-\in A(\boldsymbol{x}')}}\left[(\mathbb{I}(\boldsymbol{x}\in S_\varepsilon)+\mathbb{I}(\boldsymbol{x}\in\bar{S}_\varepsilon))\log\left(e^{f(\boldsymbol{x}_1)^\top f(\boldsymbol{x}_2)}+e^{f(\boldsymbol{x}_1)^\top f(\boldsymbol{x}^-)}\right)\right]$$

$$\geq\sum_{k=1}^K\sum_{\ell=1}^K\mathop{\mathbb{E}}_{\boldsymbol{x},\boldsymbol{x}'}\left[\mathbb{I}(\boldsymbol{x}\in S_\varepsilon\cap C_k)\mathbb{I}(\boldsymbol{x}'\in C_\ell)\mathop{\mathbb{E}}_{\substack{\boldsymbol{x}_1\in A(\boldsymbol{x})\\\boldsymbol{x}^-\in A(\boldsymbol{x}')}}\log\left(e^{1-\varepsilon}+e^{f(\boldsymbol{x}_1)^\top f(\boldsymbol{x}^-)}\right)\right]+\mathop{\mathbb{E}}_{\boldsymbol{x}}\left[\mathbb{I}(\boldsymbol{x}\in\bar{S}_\varepsilon)\log\left(e^{-1}+e^{-1}\right)\right]$$

$$
\begin{aligned}
&= \left( \sum_{k=1}^{K} \sum_{\ell=1}^{K} \mathop{\mathbb{E}}_{\boldsymbol{x},\boldsymbol{x}'} \left[ \mathbb{I}(\boldsymbol{x} \in C_k)\mathbb{I}(\boldsymbol{x}' \in C_\ell) \log\left( e^{1-\varepsilon} + e^{\mu_k^\top \mu_\ell} \right) \right] + \Delta_1 \right) - (1 - \log 2) R_\varepsilon \\
&= \sum_{k=1}^{K} \sum_{\ell=1}^{K} \left[ p_k p_l \log\left( e^{1-\varepsilon} + e^{\mu_k^\top \mu_\ell} \right) \right] - (1 - \log 2) R_\varepsilon + \Delta_1 \\
&\geq p_k p_\ell \log\left( e^{1-\varepsilon} + e^{\mu_k^\top \mu_\ell} \right) - (1 - \log 2) R_\varepsilon + \Delta_1,
\end{aligned}
\tag{15}
$$

where $\Delta_1$ is defined as

$$
\begin{aligned}
\Delta_1 :=& \sum_{k=1}^{K} \sum_{\ell=1}^{K} \mathop{\mathbb{E}}_{\boldsymbol{x},\boldsymbol{x}'} \left[ \mathbb{I}(\boldsymbol{x} \in S_\varepsilon \cap C_k)\mathbb{I}(\boldsymbol{x}' \in C_\ell) \mathop{\mathbb{E}}_{\substack{\boldsymbol{x}_1 \in A(\boldsymbol{x}) \\ \boldsymbol{x}^- \in A(\boldsymbol{x}')}} \log\left( e^{1-\varepsilon} + e^{f(\boldsymbol{x}_1)^\top f(\boldsymbol{x}^-)} \right) \right] \\
& - \sum_{k=1}^{K} \sum_{\ell=1}^{K} \mathop{\mathbb{E}}_{\boldsymbol{x},\boldsymbol{x}'} \left[ \mathbb{I}(\boldsymbol{x} \in C_k)\mathbb{I}(\boldsymbol{x}' \in C_\ell) \log\left( e^{1-\varepsilon} + e^{\mu_k^\top \mu_\ell} \right) \right] \\
=& -\sum_{k=1}^{K} \sum_{\ell=1}^{K} \mathop{\mathbb{E}}_{\boldsymbol{x},\boldsymbol{x}'} \left[ \left( \mathbb{I}(\boldsymbol{x} \in C_k) - \mathbb{I}(\boldsymbol{x} \in S_\varepsilon \cap C_k) \right) \mathbb{I}(\boldsymbol{x}' \in C_\ell) \mathop{\mathbb{E}}_{\substack{\boldsymbol{x}_1 \in A(\boldsymbol{x}) \\ \boldsymbol{x}^- \in A(\boldsymbol{x}')}} \left[ \log\left( e^{1-\varepsilon} + e^{f(\boldsymbol{x}_1)^\top f(\boldsymbol{x}^-)} \right) \right] \right] \\
& + \sum_{k=1}^{K} \sum_{\ell=1}^{K} \mathop{\mathbb{E}}_{\boldsymbol{x},\boldsymbol{x}'} \left[ \mathbb{I}(\boldsymbol{x} \in C_k)\mathbb{I}(\boldsymbol{x}' \in C_\ell) \mathop{\mathbb{E}}_{\substack{\boldsymbol{x}_1 \in A(\boldsymbol{x}) \\ \boldsymbol{x}^- \in A(\boldsymbol{x}')}} \left[ \log\left( e^{1-\varepsilon} + e^{f(\boldsymbol{x}_1)^\top f(\boldsymbol{x}^-)} \right) - \log\left( e^{1-\varepsilon} + e^{\mu_k^\top \mu_\ell} \right) \right] \right].
\end{aligned}
$$

Then,

$$
\begin{aligned}
&|\Delta_1| \\
&\leq \log(2e) \sum_{k=1}^{K} \sum_{\ell=1}^{K} \mathop{\mathbb{E}}_{\boldsymbol{x},\boldsymbol{x}'} \left[ \left( \mathbb{I}(\boldsymbol{x} \in C_k) - \mathbb{I}(\boldsymbol{x} \in S_\varepsilon \cap C_k) \right) \mathbb{I}(\boldsymbol{x}' \in C_\ell) \right] \\
&\quad + \sum_{k=1}^{K} \sum_{\ell=1}^{K} \mathop{\mathbb{E}}_{\boldsymbol{x},\boldsymbol{x}'} \left[ \mathbb{I}(\boldsymbol{x} \in C_k)\mathbb{I}(\boldsymbol{x}' \in C_\ell) \mathop{\mathbb{E}}_{\substack{\boldsymbol{x}_1 \in A(\boldsymbol{x}) \\ \boldsymbol{x}^- \in A(\boldsymbol{x}')}} \left[ \log\left( e^{1-\varepsilon} + e^{f(\boldsymbol{x}_1)^\top f(\boldsymbol{x}^-)} \right) - \log\left( e^{1-\varepsilon} + e^{\mu_k^\top \mu_\ell} \right) \right] \right] \\
&\leq (1 + \log 2) R_\varepsilon + \sum_{k=1}^{K} \sum_{\ell=1}^{K} \mathop{\mathbb{E}}_{\boldsymbol{x},\boldsymbol{x}'} \left[ \mathbb{I}(\boldsymbol{x} \in C_k)\mathbb{I}(\boldsymbol{x}' \in C_\ell) \mathop{\mathbb{E}}_{\substack{\boldsymbol{x}_1 \in A(\boldsymbol{x}) \\ \boldsymbol{x}^- \in A(\boldsymbol{x}')}} \left[ \log\left( e^{1-\varepsilon} + e^{f(\boldsymbol{x}_1)^\top f(\boldsymbol{x}^-)} \right) - \log\left( e^{1-\varepsilon} + e^{\mu_k^\top \mu_\ell} \right) \right] \right] \\
&\leq (1 + \log 2) R_\varepsilon + \sum_{k=1}^{K} \sum_{\ell=1}^{K} \mathop{\mathbb{E}}_{\boldsymbol{x},\boldsymbol{x}'} \left[ \mathbb{I}(\boldsymbol{x} \in C_k)\mathbb{I}(\boldsymbol{x}' \in C_\ell) \mathop{\mathbb{E}}_{\substack{\boldsymbol{x}_1 \in A(\boldsymbol{x}) \\ \boldsymbol{x}^- \in A(\boldsymbol{x}')}} \left[ \frac{e^\xi}{e^{1-\varepsilon} + e^\xi} \left| f(\boldsymbol{x}_1)^\top f(\boldsymbol{x}^-) - \mu_k^\top \mu_\ell \right| \right] \right] \\
&\qquad\qquad\qquad\qquad\qquad\qquad\qquad\qquad\qquad\qquad\qquad\qquad\qquad \text{(mean value theorem, } \xi \in [-1,1]) \\
&\leq (1 + \log 2) R_\varepsilon + \sum_{k=1}^{K} \sum_{\ell=1}^{K} \mathop{\mathbb{E}}_{\boldsymbol{x},\boldsymbol{x}'} \left[ \mathbb{I}(\boldsymbol{x} \in C_k)\mathbb{I}(\boldsymbol{x}' \in C_\ell) \mathop{\mathbb{E}}_{\substack{\boldsymbol{x}_1 \in A(\boldsymbol{x}) \\ \boldsymbol{x}^- \in A(\boldsymbol{x}')}} \left| f(\boldsymbol{x}_1)^\top f(\boldsymbol{x}^-) - \mu_k^\top \mu_\ell \right| \right] \\
&\qquad\qquad\qquad\qquad\qquad\qquad\qquad\qquad\qquad\qquad\qquad\qquad\qquad\qquad\qquad\qquad\qquad\quad (16) \\
&\leq (1 + \log 2) R_\varepsilon + \sum_{k=1}^{K} \sum_{\ell=1}^{K} \mathop{\mathbb{E}}_{\boldsymbol{x},\boldsymbol{x}'} \left[ \mathbb{I}(\boldsymbol{x} \in C_k)\mathbb{I}(\boldsymbol{x}' \in C_\ell) \mathop{\mathbb{E}}_{\substack{\boldsymbol{x}_1 \in A(\boldsymbol{x}) \\ \boldsymbol{x}^- \in A(\boldsymbol{x}')}} \left[ \left| (f(\boldsymbol{x}_1) - \mu_k)^\top (f(\boldsymbol{x}^-) - \mu_\ell) \right| \right. \right. \\
&\qquad + \|f(\boldsymbol{x}_1) - \mu_k\| \cdot \|\mu_\ell\| + \|\mu_k\| \cdot \|f(\boldsymbol{x}^-) - \mu_\ell\| \Big]\Big]
\end{aligned}
$$

$$\leq (1 + \log 2)R_\varepsilon + \sum_{k=1}^{K} \sum_{\ell=1}^{K} \mathop{\mathbb{E}}_{\boldsymbol{x}, \boldsymbol{x}'} \left[ \mathbb{I}(\boldsymbol{x} \in C_k) \mathbb{I}(\boldsymbol{x}' \in C_\ell) \mathop{\mathbb{E}}_{\substack{\boldsymbol{x}_1 \in A(\boldsymbol{x}) \\ \boldsymbol{x}^- \in A(\boldsymbol{x}')}} \left| (f(\boldsymbol{x}_1) - \mu_k)^\top (f(\boldsymbol{x}^-) - \mu_\ell) \right| \right]$$

$$+ 2 \sum_{k=1}^{K} \mathop{\mathbb{E}}_{\boldsymbol{x}} \left[ \mathbb{I}(\boldsymbol{x} \in C_k) \mathop{\mathbb{E}}_{\boldsymbol{x}_1 \in A(\boldsymbol{x})} \| f(\boldsymbol{x}_1) - \mu_k \| \right]$$

$$\leq (1 + \log 2)R_\varepsilon + \left[ \sum_{k=1}^{K} p_k \mathop{\mathbb{E}}_{\boldsymbol{x} \in C_k} \mathop{\mathbb{E}}_{\boldsymbol{x}_1 \in A(\boldsymbol{x})} \| f(\boldsymbol{x}_1) - \mu_k \| \right]^2 + 2 \sum_{k=1}^{K} p_k \mathop{\mathbb{E}}_{\boldsymbol{x} \in C_k} \mathop{\mathbb{E}}_{\boldsymbol{x}_1 \in A(\boldsymbol{x})} \| f(\boldsymbol{x}_1) - \mu_k \|$$

$$\leq (1 + \log 2)R_\varepsilon + \left[ \sum_{k=1}^{K} p_k \cdot \left( 2\varepsilon + L\delta + 4(1 - \sigma) + \frac{4R_\varepsilon}{p_k} \right) \right]^2 + 2 \sum_{k=1}^{K} p_k \cdot \left( 2\varepsilon + L\delta + 4(1 - \sigma) + \frac{4R_\varepsilon}{p_k} \right)$$

$$\text{(Lemma C.1)}$$

$$= (1 + \log 2)R_\varepsilon + (2\varepsilon + L\delta + 4(1 - \sigma) + 4KR_\varepsilon)^2 + 2 (2\varepsilon + L\delta + 4(1 - \sigma) + 4KR_\varepsilon).$$

Then (15) turns to

$$p_k p_\ell \log \left( e^{1-\varepsilon} + e^{\mu_k^\top \mu_\ell} \right)$$

$$\leq \mathcal{L}_2^{\text{InfoNCE}}(f) + (1 - \log 2)R_\varepsilon + |\Delta_1|$$

$$\leq \mathcal{L}_2^{\text{InfoNCE}}(f) + (2\varepsilon + L\delta + 4(1 - \sigma) + 4KR_\varepsilon)^2 + (4\varepsilon + 2L\delta + 8(1 - \sigma) + 8KR_\varepsilon) + 2R_\varepsilon.$$

Let

$$\tau(\sigma, \delta, \varepsilon, R_\varepsilon) := (2\varepsilon + L\delta + 4(1 - \sigma) + 4KR_\varepsilon)^2 + (4\varepsilon + 2L\delta + 8(1 - \sigma) + 8KR_\varepsilon) + 2R_\varepsilon,$$

and we obtain

$$\mu_k^\top \mu_\ell \leq \log \left( \exp \left\{ \frac{\mathcal{L}_2^{\text{InfoNCE}}(f) + \tau(\sigma, \delta, \varepsilon, R_\varepsilon)}{p_k p_\ell} \right\} - \exp(1 - \varepsilon) \right).$$

This finishes the proof. $\qquad \square$

## C.2 Cross-Correlation Loss

**Lemma 4.1.** *For a given encoder $f$, the alignment $\mathcal{L}_{\text{align}}(f)$ in (4) is upper bounded via $\mathcal{L}_1^{Cross}(f)$:*

$$\mathcal{L}_{\text{align}}(f) = \mathop{\mathbb{E}}_{\boldsymbol{x}} \mathop{\mathbb{E}}_{\boldsymbol{x}_1, \boldsymbol{x}_2 \in A(\boldsymbol{x})} \| f(\boldsymbol{x}_1) - f(\boldsymbol{x}_2) \|^2 \leq 2\sqrt{d \cdot \mathcal{L}_1^{Cross}(f)},$$

*where $d$ is the output dimension of encoder $f$.*

*Proof.* Since $\mathbb{E}_{\boldsymbol{x}} \mathbb{E}_{\boldsymbol{x}_1 \in A(\boldsymbol{x})} f_i(\boldsymbol{x}_1)^2 = 1$, for each coordinate component $i$, we have

$$1 - \mathop{\mathbb{E}}_{\boldsymbol{x}} \mathop{\mathbb{E}}_{\boldsymbol{x}_1, \boldsymbol{x}_2 \in A(\boldsymbol{x})} [f_i(\boldsymbol{x}_1) f_i(\boldsymbol{x}_2)] = \frac{1}{2} \mathop{\mathbb{E}}_{\boldsymbol{x}} \mathop{\mathbb{E}}_{\boldsymbol{x}_1, \boldsymbol{x}_2 \in A(\boldsymbol{x})} [f_i(\boldsymbol{x}_1)^2 + f_i(\boldsymbol{x}_2)^2] - \mathop{\mathbb{E}}_{\boldsymbol{x}} \mathop{\mathbb{E}}_{\boldsymbol{x}_1, \boldsymbol{x}_2 \in A(\boldsymbol{x})} [f_i(\boldsymbol{x}_1) f_i(\boldsymbol{x}_2)]$$

$$= \frac{1}{2} \mathop{\mathbb{E}}_{\boldsymbol{x}} \mathop{\mathbb{E}}_{\boldsymbol{x}_1, \boldsymbol{x}_2 \in A(\boldsymbol{x})} [f_i(\boldsymbol{x}_1) - f_i(\boldsymbol{x}_2)]^2.$$

Then

$$\mathop{\mathbb{E}}_{\boldsymbol{x}} \mathop{\mathbb{E}}_{\boldsymbol{x}_1, \boldsymbol{x}_2 \in A(\boldsymbol{x})} \| f(\boldsymbol{x}_1) - f(\boldsymbol{x}_2) \|^2 = \sum_{i=1}^{d} \mathop{\mathbb{E}}_{\boldsymbol{x}} \mathop{\mathbb{E}}_{\boldsymbol{x}_1, \boldsymbol{x}_2 \in A(\boldsymbol{x})} [f_i(\boldsymbol{x}_1) - f_i(\boldsymbol{x}_2)]^2$$

$$= 2 \sum_{i=1}^{d} \left( 1 - \mathop{\mathbb{E}}_{\boldsymbol{x}} \mathop{\mathbb{E}}_{\boldsymbol{x}_1, \boldsymbol{x}_2 \in A(\boldsymbol{x})} [f_i(\boldsymbol{x}_1) f_i(\boldsymbol{x}_2)] \right)$$

$$\leq 2 \left( d \sum_{i=1}^{d} \left( 1 - \mathop{\mathbb{E}}_{\boldsymbol{x}} \mathop{\mathbb{E}}_{\boldsymbol{x}_1, \boldsymbol{x}_2 \in A(\boldsymbol{x})} [f_i(\boldsymbol{x}_1) f_i(\boldsymbol{x}_2)] \right)^2 \right)^{\frac{1}{2}}$$

$$= 2d^{\frac{1}{2}} \mathcal{L}_1^{Cross}(f)^{\frac{1}{2}},$$

where the inequality holds due to the Cauchy inequality. $\qquad \square$

**Lemma C.2.** *Assume that encoder $f$ with norm $\sqrt{d}$ is $L$-Lipschitz continuous. If the augmented data is $(\sigma, \delta)$-augmented, then for any $\varepsilon \geq 0$,*

$$\left\| \mathbb{E}_{\boldsymbol{x}} \mathbb{E}_{\boldsymbol{x}_1, \boldsymbol{x}_2 \in A(\boldsymbol{x})} \left[ f(\boldsymbol{x}_1) f(\boldsymbol{x}_2)^\top \right] - \sum_{k=1}^{K} p_k \mu_k \mu_k^\top \right\|^2 \leq \tau'(\sigma, \delta, \varepsilon, R_\varepsilon),$$

*where $\tau'(\sigma, \delta, \varepsilon, R_\varepsilon)$ is defined as*

$$4d \left[ \left( 1 - \sigma + \frac{L\delta + 2\varepsilon}{2\sqrt{d}} \right)^2 + (1 - \sigma) + K R_\varepsilon \left( 3 - 2\sigma + \frac{L\delta + 2\varepsilon}{\sqrt{d}} \right) + R_\varepsilon^2 \left( \sum_{k=1}^{K} \frac{1}{p_k} \right) \right] + \sqrt{d} \left( \varepsilon^2 + 4dR_\varepsilon \right)^{\frac{1}{2}}.$$

*Proof.* We first decompose the LHS as

$$\mathbb{E}_{\boldsymbol{x}} \mathbb{E}_{\boldsymbol{x}_1, \boldsymbol{x}_2 \in A(\boldsymbol{x})} \left[ f(\boldsymbol{x}_1) f(\boldsymbol{x}_2)^\top \right] - \sum_{k=1}^{K} p_k \mu_k \mu_k^\top$$

$$= \sum_{k=1}^{K} p_k \mathbb{E}_{\boldsymbol{x} \in C_k} \mathbb{E}_{\boldsymbol{x}_1, \boldsymbol{x}_2 \in A(\boldsymbol{x})} \left[ f(\boldsymbol{x}_1) f(\boldsymbol{x}_2)^\top \right] - \sum_{k=1}^{K} p_k \mu_k \mu_k^\top$$

$$= \sum_{k=1}^{K} p_k \mathbb{E}_{\boldsymbol{x} \in C_k} \mathbb{E}_{\boldsymbol{x}_1 \in A(\boldsymbol{x})} \left[ f(\boldsymbol{x}_1) f(\boldsymbol{x}_1)^\top \right] - \sum_{k=1}^{K} p_k \mu_k \mu_k^\top + \sum_{k=1}^{K} p_k \mathbb{E}_{\boldsymbol{x} \in C_k} \mathbb{E}_{\boldsymbol{x}_1, \boldsymbol{x}_2 \in A(\boldsymbol{x})} \left[ f(\boldsymbol{x}_1) (f(\boldsymbol{x}_2)^\top - f(\boldsymbol{x}_1)^\top) \right]$$

$$= \sum_{k=1}^{K} p_k \mathbb{E}_{\boldsymbol{x} \in C_k} \mathbb{E}_{\boldsymbol{x}_1 \in A(\boldsymbol{x})} \left[ (f(\boldsymbol{x}_1) - \mu_k)(f(\boldsymbol{x}_1) - \mu_k)^\top \right] + \mathbb{E}_{\boldsymbol{x}} \mathbb{E}_{\boldsymbol{x}_1, \boldsymbol{x}_2 \in A(\boldsymbol{x})} \left[ f(\boldsymbol{x}_1) (f(\boldsymbol{x}_2)^\top - f(\boldsymbol{x}_1)^\top) \right].$$

Then its norm is

$$\left\| \mathbb{E}_{\boldsymbol{x}} \mathbb{E}_{\boldsymbol{x}_1, \boldsymbol{x}_2 \in A(\boldsymbol{x})} \left[ f(\boldsymbol{x}_1) f(\boldsymbol{x}_2)^\top \right] - \sum_{k=1}^{K} p_k \mu_k \mu_k^\top \right\|$$

$$\leq \sum_{k=1}^{K} p_k \mathbb{E}_{\boldsymbol{x} \in C_k} \mathbb{E}_{\boldsymbol{x}_1 \in A(\boldsymbol{x})} \left[ \left\| (f(\boldsymbol{x}_1) - \mu_k)(f(\boldsymbol{x}_1) - \mu_k)^\top \right\| \right] + \mathbb{E}_{\boldsymbol{x}} \mathbb{E}_{\boldsymbol{x}_1, \boldsymbol{x}_2 \in A(\boldsymbol{x})} \left[ \left\| f(\boldsymbol{x}_1) (f(\boldsymbol{x}_2)^\top - f(\boldsymbol{x}_1)^\top) \right\| \right]$$

$$\leq \sum_{k=1}^{K} p_k \mathbb{E}_{\boldsymbol{x} \in C_k} \mathbb{E}_{\boldsymbol{x}_1 \in A(\boldsymbol{x})} \left[ \| f(\boldsymbol{x}_1) - \mu_k \|^2 \right] + \mathbb{E}_{\boldsymbol{x}} \mathbb{E}_{\boldsymbol{x}_1, \boldsymbol{x}_2 \in A(\boldsymbol{x})} \left[ \| f(\boldsymbol{x}_1) \| \| f(\boldsymbol{x}_2) - f(\boldsymbol{x}_1) \| \right]$$

$$\leq \sum_{k=1}^{K} p_k \mathbb{E}_{\boldsymbol{x} \in C_k} \mathbb{E}_{\boldsymbol{x}_1 \in A(\boldsymbol{x})} \left[ \| f(\boldsymbol{x}_1) - \mu_k \|^2 \right] + \left[ \mathbb{E}_{\boldsymbol{x}} \mathbb{E}_{\boldsymbol{x}_1 \in A(\boldsymbol{x})} \| f(\boldsymbol{x}_1) \|^2 \right]^{\frac{1}{2}} \left[ \mathbb{E}_{\boldsymbol{x}} \mathbb{E}_{\boldsymbol{x}_1, \boldsymbol{x}_2 \in A(\boldsymbol{x})} \| f(\boldsymbol{x}_2) - f(\boldsymbol{x}_1) \|^2 \right]^{\frac{1}{2}}$$

$$\text{(Cauchy–Schwarz inequality)}$$

$$\leq \sum_{k=1}^{K} p_k \mathbb{E}_{\boldsymbol{x} \in C_k} \mathbb{E}_{\boldsymbol{x}_1 \in A(\boldsymbol{x})} \left[ \| f(\boldsymbol{x}_1) - \mu_k \|^2 \right] + \sqrt{d} \left( \varepsilon^2 + 4dR_\varepsilon \right)^{\frac{1}{2}} \qquad \text{(Lemma 4.1)}$$

$$\leq 4d \sum_{k=1}^{K} p_k \left[ \left( 1 - \sigma + \frac{L}{2\sqrt{d}}\delta + \frac{\varepsilon}{\sqrt{d}} + \frac{R_\varepsilon}{p_k} \right)^2 + \left( 1 - \sigma + \frac{R_\varepsilon}{p_k} \right) \right] + \sqrt{d} \left( \varepsilon^2 + 4dR_\varepsilon \right)^{\frac{1}{2}}$$

$$\text{(Lemma C.1)}$$

$$= 4d \left[ \left( 1 - \sigma + \frac{L\delta + 2\varepsilon}{2\sqrt{d}} \right)^2 + (1 - \sigma) + K R_\varepsilon \left( 3 - 2\sigma + \frac{L\delta + 2\varepsilon}{\sqrt{d}} \right) + R_\varepsilon^2 \left( \sum_{k=1}^{K} \frac{1}{p_k} \right) \right] + \sqrt{d} \left( \varepsilon^2 + 4dR_\varepsilon \right)^{\frac{1}{2}}$$

$$= \tau'(\sigma, \delta, \varepsilon, R_\varepsilon).$$

This finishes the proof. $\qquad \square$

**Theorem 4.** *Assume that encoder $f$ with norm $\sqrt{d}$ is $L$-Lipschitz continuous. If the augmented data is $(\sigma, \delta)$-augmented, then for any $\varepsilon \geq 0$ and $k \neq \ell$, we have*

$$\mu_k^\top \mu_\ell \leq \sqrt{\frac{2}{p_k p_\ell} \left( \mathcal{L}_2^{\text{Cross}}(f) + \tau'(\sigma, \delta, \varepsilon, R_\varepsilon) - \frac{d - K}{2} \right)},$$

*where $\tau'(\sigma, \delta, \varepsilon, R_\varepsilon)$ is an upper bound of $\| \mathbb{E}_{\boldsymbol{x}} \mathbb{E}_{\boldsymbol{x}_1, \boldsymbol{x}_2 \in A(\boldsymbol{x})} [f(\boldsymbol{x}_1) f(\boldsymbol{x}_2)^\top] - \sum_{k=1}^{K} p_k \mu_k \mu_k^\top \|^2$.*

*Proof.* Let $U = (\sqrt{p_1}\mu_1, \ldots, \sqrt{p_K}\mu_K) \in \mathbb{R}^{d \times K}$.

$$\left\| \sum_{k=1}^{K} p_k \mu_k \mu_k^\top - I_d \right\|^2 = \left\| UU^\top - I_d \right\|^2$$

$$= \mathrm{Tr}(UU^\top UU^\top - 2UU^\top + I_d) \qquad \text{(due to } \|A\|^2 = \mathrm{Tr}(A^\top A))$$

$$= \mathrm{Tr}(U^\top UU^\top U - 2U^\top U + I_K) + d - K$$

$$= \left\| U^\top U - I_K \right\|^2 + d - K$$

$$= \sum_{k=1}^{K} \sum_{\ell=1}^{K} (\sqrt{p_k p_\ell} \mu_k^\top \mu_\ell - \delta_{k\ell})^2 + d - K$$

$$\geq p_k p_\ell (\mu_k^\top \mu_\ell)^2 + d - K.$$

where $\delta_{kl}$ is the Dirichlet function.

Therefore,

$$\left(\mu_k^\top \mu_l\right)^2$$

$$\leq \frac{\left\| \sum_{k=1}^{K} p_k \mu_k \mu_k^\top - I_d \right\|^2 - (d - K)}{p_k p_\ell}$$

$$= \frac{\left\| \mathbb{E}_{\boldsymbol{x}} \mathbb{E}_{\boldsymbol{x}_1, \boldsymbol{x}_2 \in A(\boldsymbol{x})}[f(\boldsymbol{x}_1) f(\boldsymbol{x}_2)^\top] - I_d + \sum_{k=1}^{K} p_k \mu_k \mu_k^\top - \mathbb{E}_{\boldsymbol{x}} \mathbb{E}_{\boldsymbol{x}_1, \boldsymbol{x}_2 \in A(\boldsymbol{x})}[f(\boldsymbol{x}_1) f(\boldsymbol{x}_2)^\top] \right\|^2 - (d - K)}{p_k p_\ell}$$

$$\leq \frac{2 \left\| \mathbb{E}_{\boldsymbol{x}} \mathbb{E}_{\boldsymbol{x}_1, \boldsymbol{x}_2 \in A(\boldsymbol{x})}[f(\boldsymbol{x}_1) f(\boldsymbol{x}_2)^\top] - I_d \right\|^2 + 2 \left\| \sum_{k=1}^{K} p_k \mu_k \mu_k^\top - \mathbb{E}_{\boldsymbol{x}} \mathbb{E}_{\boldsymbol{x}_1, \boldsymbol{x}_2 \in A(\boldsymbol{x})}[f(\boldsymbol{x}_1) f(\boldsymbol{x}_2)^\top] \right\|^2 - (d - K)}{p_k p_\ell}$$

$$\leq \frac{2 \mathcal{L}_2^{\mathrm{Cross}}(f) + 2\tau'(\varepsilon, \sigma, \delta) - (d - K)}{p_k p_\ell} \qquad \text{(Lemma C.2)}$$

$$= \frac{2}{p_k p_\ell} \left( \mathcal{L}_2^{\mathrm{Cross}}(f) + \tau'(\varepsilon, \sigma, \delta) - \frac{d - K}{2} \right).$$

This finishes the proof. $\qquad\square$

## D  ANALYSIS OF $t$-INFONCE

The population loss of $t$-InfoNCE (Hu et al., 2022) can be written as:

$$\mathcal{L}_{t\text{-InfoNCE}} = - \mathbb{E}_{\substack{\boldsymbol{x}, \boldsymbol{x}' \\ }} \mathbb{E}_{\substack{\boldsymbol{x}_1, \boldsymbol{x}_2 \in A(\boldsymbol{x}) \\ \boldsymbol{x}^- \in A(\boldsymbol{x}')}} \log \frac{(1 + \|f(\boldsymbol{x}_1) - f(\boldsymbol{x}_2)\|^2)^{-1}}{(1 + \|f(\boldsymbol{x}_1) - f(\boldsymbol{x}_2)\|^2)^{-1} + (1 + \|f(\boldsymbol{x}_1) - f(\boldsymbol{x}^-)\|^2)^{-1}}.$$

It can be divided into two parts:

$$\mathcal{L}_{t\text{-InfoNCE}} = \underbrace{\mathbb{E}_{\boldsymbol{x}} \mathbb{E}_{\boldsymbol{x}_1, \boldsymbol{x}_2 \in A(\boldsymbol{x})} \log(1 + \|f(\boldsymbol{x}_1) - f(\boldsymbol{x}_2)\|^2)}_{=: \mathcal{L}_1(f)}$$

$$+ \underbrace{\mathbb{E}_{\substack{\boldsymbol{x}, \boldsymbol{x}' \\ }} \mathbb{E}_{\substack{\boldsymbol{x}_1, \boldsymbol{x}_2 \in A(\boldsymbol{x}) \\ \boldsymbol{x}^- \in A(\boldsymbol{x}')}} \log \left[ (1 + \|f(\boldsymbol{x}_1) - f(\boldsymbol{x}_2)\|^2)^{-1} + (1 + \|f(\boldsymbol{x}_1) - f(\boldsymbol{x}^-)\|^2)^{-1}) \right]}_{=: \mathcal{L}_2(f)}.$$

Similar to the InfoNCE loss and the cross-correlation loss, we can connect $\mathcal{L}_1(f)$ and $\mathcal{L}_2(f)$ with the alignment and divergence by the following Lemma D.1 and Theorem 5, respectively.

**Lemma D.1.** *For a given encoder $f$, the alignment $\mathcal{L}_{\mathrm{align}}(f)$ in (4) is upper bounded via $\mathcal{L}_1(f)$, i.e.,*

$$\mathcal{L}_{\mathrm{align}}(f) = \mathbb{E}_{\boldsymbol{x}} \mathbb{E}_{\boldsymbol{x}_1, \boldsymbol{x}_2 \in A(\boldsymbol{x})} \|f(\boldsymbol{x}_1) - f(\boldsymbol{x}_2)\|^2 \leq \frac{4}{\ln 5} \mathcal{L}_1(f).$$

*Proof.* It is easy to verify that $\log(1 + t^2) \geq \frac{\ln 5}{4} t^2$ for any $t \in [0, 2]$.

Since $\|f(\boldsymbol{x}_1) - f(\boldsymbol{x}_2)\| \in [0, 2]$, we have

$$\|f(\boldsymbol{x}_1) - f(\boldsymbol{x}_2)\|^2 \leq \frac{4}{\ln 5} \log(1 + \|f(\boldsymbol{x}_1) - f(\boldsymbol{x}_2)\|^2).$$

Thus,

$$\begin{aligned}
\mathcal{L}_{\text{align}}(f) &= \mathbb{E}_{\boldsymbol{x}} \mathbb{E}_{\boldsymbol{x}_1, \boldsymbol{x}_2 \in A(\boldsymbol{x})} \|f(\boldsymbol{x}_1) - f(\boldsymbol{x}_2)\|^2 \\
&\leq \frac{4}{\ln 5} \mathbb{E}_{\boldsymbol{x}} \mathbb{E}_{\boldsymbol{x}_1, \boldsymbol{x}_2 \in A(\boldsymbol{x})} \log(1 + \|f(\boldsymbol{x}_1) - f(\boldsymbol{x}_2)\|^2) \\
&= \frac{4}{\ln 5} \mathcal{L}_1(f).
\end{aligned}$$

This finishes the proof. $\qquad\square$

**Theorem 5.** *Assume that encoder $f$ with norm 1 is $L$-Lipschitz continuous. If the augmented data is $(\sigma, \delta)$-augmented, then for any $\varepsilon > 0$ and $k \neq \ell$,*

$$\mu_k^\top \mu_\ell \leq \frac{1}{2} \left( 3 - \frac{1}{\exp\left\{ \frac{\mathcal{L}_2(f) + \tau''(\sigma, \delta, \varepsilon, R_\varepsilon)}{p_k p_\ell} \right\} - \frac{1}{1+2\varepsilon}} \right),$$

*where $\tau''(\sigma, \delta, \varepsilon, R_\varepsilon)$ is a non-negative term, decreasing with smaller $\varepsilon$, $R_\varepsilon$ or sharper concentration of augmented data, and $\tau(\sigma, \delta, \varepsilon, R_\varepsilon) = 0$ when $\sigma = 1, \delta = 0, \varepsilon = 0, R_\varepsilon = 0$*

*Proof.* Given $\boldsymbol{x} \in S_\varepsilon$, for any $\boldsymbol{x}_1, \boldsymbol{x}_2 \in A(\boldsymbol{x})$, we have

$$\begin{aligned}
&\log[(1 + \|f(\boldsymbol{x}_1) - f(\boldsymbol{x}_2)\|_2^2)^{-1} + (1 + \|f(\boldsymbol{x}_1) - f(\boldsymbol{x}^-)\|_2^2)^{-1})] \\
&= \log \left[ \frac{1}{3 - 2f(\boldsymbol{x}_1)^\top f(\boldsymbol{x}_2)} + \frac{1}{3 - 2f(\boldsymbol{x}_1)^\top f(\boldsymbol{x}^-)} \right] \\
&= \log \left[ \frac{1}{3 - 2f(\boldsymbol{x}_1)^\top f(\boldsymbol{x}_1) - 2f(\boldsymbol{x}_1)^\top (f(\boldsymbol{x}_2) - f(\boldsymbol{x}_1))} + \frac{1}{3 - 2f(\boldsymbol{x}_1)^\top f(\boldsymbol{x}^-)} \right] \\
&\geq \log \left[ \frac{1}{3 - 2\|f(\boldsymbol{x}_1)\|^2 + 2\|f(\boldsymbol{x}_1)\| \cdot \varepsilon} + \frac{1}{3 - 2f(\boldsymbol{x}_1)^\top f(\boldsymbol{x}^-)} \right] \\
&= \log \left[ \frac{1}{1 + 2\varepsilon} + \frac{1}{3 - 2f(\boldsymbol{x}_1)^\top f(\boldsymbol{x}^-)} \right].
\end{aligned}$$

Therefore, we have

$$\begin{aligned}
\mathcal{L}_2(f) &= \mathbb{E}_{\boldsymbol{x}, \boldsymbol{x}'} \mathbb{E}_{\substack{\boldsymbol{x}_1, \boldsymbol{x}_2 \in A(\boldsymbol{x}) \\ \boldsymbol{x}^- \in A(\boldsymbol{x}')}} \log[(1 + \|f(\boldsymbol{x}_1) - f(\boldsymbol{x}_2)\|_2^2)^{-1} + (1 + \|f(\boldsymbol{x}_1) - f(\boldsymbol{x}^-)\|_2^2)^{-1})] \\
&= \mathbb{E}_{\boldsymbol{x}, \boldsymbol{x}'} \mathbb{E}_{\substack{\boldsymbol{x}_1, \boldsymbol{x}_2 \in A(\boldsymbol{x}) \\ \boldsymbol{x}^- \in A(\boldsymbol{x}')}} \log \left[ \frac{1}{3 - 2f(\boldsymbol{x}_1)^\top f(\boldsymbol{x}_2)} + \frac{1}{3 - 2f(\boldsymbol{x}_1)^\top f(\boldsymbol{x}^-)} \right] \\
&= \mathbb{E}_{\boldsymbol{x}, \boldsymbol{x}'} \mathbb{E}_{\substack{\boldsymbol{x}_1, \boldsymbol{x}_2 \in A(\boldsymbol{x}) \\ \boldsymbol{x}^- \in A(\boldsymbol{x}')}} [\mathbb{I}(\boldsymbol{x} \in S_\varepsilon) + \mathbb{I}(\boldsymbol{x} \in \overline{S_\varepsilon})] \log \left[ \frac{1}{3 - 2f(\boldsymbol{x}_1)^\top f(\boldsymbol{x}_2)} + \frac{1}{3 - 2f(\boldsymbol{x}_1)^\top f(\boldsymbol{x}^-)} \right] \\
&\geq \sum_{k=1}^K \sum_{\ell=1}^K \mathbb{E}_{\boldsymbol{x}, \boldsymbol{x}'} \left[ \mathbb{I}(\boldsymbol{x} \in S_\varepsilon \cap C_k) \mathbb{I}(\boldsymbol{x}' \in C_\ell) \mathbb{E}_{\substack{\boldsymbol{x}_1 \in A(\boldsymbol{x}) \\ \boldsymbol{x}^- \in A(\boldsymbol{x}')}} \log \left[ \frac{1}{1 + 2\varepsilon} + \frac{1}{3 - 2f(\boldsymbol{x}_1)^\top f(\boldsymbol{x}^-)} \right] \right] \\
&\quad + \mathbb{E}_{\boldsymbol{x}} \left[ \mathbb{I}(\boldsymbol{x} \in \overline{S_\varepsilon}) \log \left( \frac{1}{3+2} + \frac{1}{3+2} \right) \right] \\
&= \sum_{k=1}^K \sum_{\ell=1}^K \mathbb{E}_{\boldsymbol{x}, \boldsymbol{x}'} \left[ \mathbb{I}(\boldsymbol{x} \in C_k) \mathbb{I}(\boldsymbol{x}' \in C_\ell) \log \left( \frac{1}{1 + 2\varepsilon} + \frac{1}{3 - 2\mu_k^\top \mu_\ell} \right) \right] + \Delta_1 - (\log 2.5) R_\varepsilon
\end{aligned}$$

$$= \sum_{k=1}^{K} \sum_{\ell=1}^{K} p_k p_\ell \log \left( \frac{1}{1+2\varepsilon} + \frac{1}{3 - 2\mu_k^\top \mu_\ell} \right) - (\log 2.5) R_\varepsilon + \Delta_1$$

$$\geq p_k p_\ell \log \left( \frac{1}{1+2\varepsilon} + \frac{1}{3 - 2\mu_k^\top \mu_\ell} \right) - (\log 2.5) R_\varepsilon + \Delta_1, \tag{17}$$

where $\Delta_1$ is defined as

$$\Delta_1 := \sum_{k=1}^{K} \sum_{\ell=1}^{K} \mathop{\mathbb{E}}_{\boldsymbol{x},\boldsymbol{x}'} \left[ \mathbb{I}(\boldsymbol{x} \in S_\varepsilon \cap C_k) \mathbb{I}(\boldsymbol{x}' \in C_\ell) \mathop{\mathbb{E}}_{\substack{\boldsymbol{x}_1 \in A(\boldsymbol{x}) \\ \boldsymbol{x}^- \in A(\boldsymbol{x}')}} \log \left( \frac{1}{1+2\varepsilon} + \frac{1}{3 - 2f(\boldsymbol{x}_1)^\top f(\boldsymbol{x}^-)} \right) \right]$$

$$- \sum_{k=1}^{K} \sum_{\ell=1}^{K} \mathop{\mathbb{E}}_{\boldsymbol{x},\boldsymbol{x}'} \left[ \mathbb{I}(\boldsymbol{x} \in C_k) \mathbb{I}(\boldsymbol{x}' \in C_\ell) \log \left( \frac{1}{1+2\varepsilon} + \frac{1}{3 - 2\mu_k^\top \mu_\ell} \right) \right]$$

$$= - \sum_{k=1}^{K} \sum_{\ell=1}^{K} \mathop{\mathbb{E}}_{\boldsymbol{x},\boldsymbol{x}'} \left[ [\mathbb{I}(\boldsymbol{x} \in C_k) - \mathbb{I}(\boldsymbol{x} \in S_\varepsilon \cap C_k)] \mathbb{I}(\boldsymbol{x}' \in C_\ell) \mathop{\mathbb{E}}_{\substack{\boldsymbol{x}_1 \in A(\boldsymbol{x}) \\ \boldsymbol{x}^- \in A(\boldsymbol{x}')}} \log \left( \frac{1}{1+2\varepsilon} + \frac{1}{3 - 2f(\boldsymbol{x}_1)^\top f(\boldsymbol{x}^-)} \right) \right]$$

$$+ \sum_{k=1}^{K} \sum_{\ell=1}^{K} \mathop{\mathbb{E}}_{\boldsymbol{x},\boldsymbol{x}'} \mathbb{I}(\boldsymbol{x} \in C_k) \mathbb{I}(\boldsymbol{x}' \in C_\ell) \mathop{\mathbb{E}}_{\substack{\boldsymbol{x}_1 \in A(\boldsymbol{x}) \\ \boldsymbol{x}^- \in A(\boldsymbol{x}')}} \left[ \log \left( \frac{1}{1+2\varepsilon} + \frac{1}{3 - 2f(\boldsymbol{x}_1)^\top f(\boldsymbol{x}^-)} \right) - \log \left( \frac{1}{1+2\varepsilon} + \frac{1}{3 - 2\mu_k^\top \mu_\ell} \right) \right].$$

Then,

$$|\Delta_1|$$

$$\leq \sum_{k=1}^{K} \sum_{\ell=1}^{K} \mathop{\mathbb{E}}_{\boldsymbol{x},\boldsymbol{x}'} [[\mathbb{I}(\boldsymbol{x} \in C_k) - \mathbb{I}(\boldsymbol{x} \in S_\varepsilon \cap C_k)] \mathbb{I}(\boldsymbol{x}' \in C_\ell)] \cdot \log 2$$

$$+ \sum_{k=1}^{K} \sum_{\ell=1}^{K} \mathop{\mathbb{E}}_{\boldsymbol{x},\boldsymbol{x}'} \mathbb{I}(\boldsymbol{x} \in C_k) \mathbb{I}(\boldsymbol{x}' \in C_\ell) \mathop{\mathbb{E}}_{\substack{\boldsymbol{x}_1 \in A(\boldsymbol{x}) \\ \boldsymbol{x}^- \in A(\boldsymbol{x}')}} \left[ \log \left( \frac{1}{1+2\varepsilon} + \frac{1}{3 - 2f(\boldsymbol{x}_1)^\top f(\boldsymbol{x}^-)} \right) - \log \left( \frac{1}{1+2\varepsilon} + \frac{1}{3 - 2\mu_k^\top \mu_\ell} \right) \right]$$

$$\leq R_\varepsilon \log 2 + \sum_{k=1}^{K} \sum_{\ell=1}^{K} \mathop{\mathbb{E}}_{\boldsymbol{x},\boldsymbol{x}'} \mathbb{I}(\boldsymbol{x} \in C_k) \mathbb{I}(\boldsymbol{x}' \in C_\ell) \mathop{\mathbb{E}}_{\substack{\boldsymbol{x}_1 \in A(\boldsymbol{x}) \\ \boldsymbol{x}^- \in A(\boldsymbol{x}')}} \left[ \frac{1+2\varepsilon}{(3-2\xi)(2+\varepsilon-\xi)} |f(\boldsymbol{x}_1)^\top f(\boldsymbol{x}^-) - \mu_k^\top \mu_\ell| \right]$$

(mean value theorem, $\xi \in [-1, 1]$)

$$\leq R_\varepsilon \log 2 + 2 \sum_{k=1}^{K} \sum_{\ell=1}^{K} \mathop{\mathbb{E}}_{\boldsymbol{x},\boldsymbol{x}'} \left[ \mathbb{I}(\boldsymbol{x} \in C_k) \mathbb{I}(\boldsymbol{x}' \in C_\ell) \mathop{\mathbb{E}}_{\substack{\boldsymbol{x}_1 \in A(\boldsymbol{x}) \\ \boldsymbol{x}^- \in A(\boldsymbol{x}')}} |f(\boldsymbol{x}_1)^\top f(\boldsymbol{x}^-) - \mu_k^\top \mu_\ell| \right]$$

$$\leq R_\varepsilon \log 2 + 2 \left( 2\varepsilon + L\delta + 4(1-\sigma) + 4KR_\varepsilon \right)^2 + 4 \left( 2\varepsilon + L\delta + 4(1-\sigma) + 4KR_\varepsilon \right)$$

(using (16))

Therefore, according to (17),

$$p_k p_\ell \log \left( \frac{1}{1+2\varepsilon} + \frac{1}{3 - 2\mu_k^\top \mu_\ell} \right)$$

$$\leq \mathcal{L}_2(f) + R_\varepsilon \log 2.5 + |\Delta_1|$$

$$\leq \mathcal{L}_2(f) + R_\varepsilon \log 5 + 2 \left( 2\varepsilon + L\delta + 4(1-\sigma) + 4KR_\varepsilon \right)^2 + 4 \left( 2\varepsilon + L\delta + 4(1-\sigma) + 4KR_\varepsilon \right).$$

Let

$$\tau''(\sigma, \delta, \varepsilon, R_\varepsilon) := R_\varepsilon \log 5 + 2 \left( 2\varepsilon + L\delta + 4(1-\sigma) + 4KR_\varepsilon \right)^2 + 4 \left( 2\varepsilon + L\delta + 4(1-\sigma) + 4KR_\varepsilon \right),$$

and we obtain

$$\mu_k^\top \mu_\ell \leq \frac{1}{2} \left( 3 - \frac{1}{\exp\left\{ \frac{\mathcal{L}_2(f) + \tau''(\sigma, \delta, \varepsilon, R_\varepsilon)}{p_k p_\ell} \right\} - \frac{1}{1+2\varepsilon}} \right).$$

This finishes the proof. $\qquad\square$

# E  ADDITIONAL PROOFS

We give detailed proof of the linear reformulation of the nearest neighbor classifier in this section.

**Proposition E.1** (Linear reformulation of the NN classifier). *Let $G_f(\boldsymbol{x}) = \arg\min_{k \in [K]} \|f(\boldsymbol{x}) - \mu_k\|$ be the NN classifier. Then*

$$G_f(\boldsymbol{x}) = \arg\max_{k \in [K]} \left( \mu_k^\top f(\boldsymbol{x}) - \frac{1}{2}\|\mu_k\|^2 \right),$$

*which is also a linear classifier.*

*Proof.* $G_f(\boldsymbol{x}) = k$ means for each $l \in [K]$,

$$\|f(\boldsymbol{x}) - \mu_k\|^2 \leq \|f(\boldsymbol{x}) - \mu_\ell\|^2.$$

This is equivalent to

$$\mu_k^\top f(\boldsymbol{x}) - \frac{1}{2}\|\mu_k\|^2 \geq \mu_\ell^\top f(\boldsymbol{x}) - \frac{1}{2}\|\mu_\ell\|^2$$

holds for each $\ell \in [K]$. Therefore, $G_f(\boldsymbol{x}) = \arg\max_{k \in [K]} \left( \mu_k^\top f(\boldsymbol{x}) - \frac{1}{2}\|\mu_k\|^2 \right)$, which is a linear classifier. □

# F  AN EXTENSION TO $A(C_k) \cap A(C_\ell) \neq \varnothing$

In this section, we extend our theory to the case where $A(C_k) \cap A(C_\ell) \neq \varnothing$ for some $k \neq \ell$, i.e., augmentation could introduces wrong signals. To quantify this negative effect, we introduce the following definition.

**Definition 2** (Correctly augmented parts). We define the corrected augmented parts of augmentation $A$ by

$$\tilde{C}_k := \{x \in C_k : A(x) \subseteq C_k\}$$

for each $k \in [K]$.

We also denote the probability of their complement as $t := 1 - P(\cup_{k=1}^K \tilde{C}_k)$. By the definition, clearly $A(\tilde{C}_\ell) \cap A(\tilde{C}_k) = \varnothing$, hence our theory in the main body applies to $\cup_{k=1}^K \tilde{C}_k$. To see this, we first generalize the definition of $(\sigma, \delta)$-augmentation to the correctly augmented parts in the following.

**Definition 3** ($(\sigma, \delta)$-Augmentation on corrected augmented parts). The augmentation set $A$ is called a $(\sigma, \delta)$-augmentation on correctly augmented parts, if for each $\tilde{C}_k$, there exists a subset $C_k^0 \subseteq \tilde{C}_k$ (called a main part of $\tilde{C}_k$), such that both $\mathbb{P}[\boldsymbol{x} \in C_k^0] \geq \sigma \mathbb{P}[\boldsymbol{x} \in \tilde{C}_k]$ where $\sigma \in (0, 1]$ and $\sup_{\boldsymbol{x}_1, \boldsymbol{x}_2 \in C_k^0} d_A(\boldsymbol{x}_1, \boldsymbol{x}_2) \leq \delta$ hold.

Besides, we modify the definition of $\mu_k$ by $\mu_k := \mathbb{E}_{\boldsymbol{x} \in \tilde{C}_k} \mathbb{E}_{\boldsymbol{x}' \in A(\boldsymbol{x})}[f(\boldsymbol{x}')]$. Then we have a generalized version of Theorem 1.

**Theorem 6** (A generalized version of Theorem 1). *Given an augmentation $A$ that is a $(\sigma, \delta)$-augmentation on corrected augmented parts, if*

$$\mu_\ell^\top \mu_k < r^2 \left( 1 - \rho_{max}(\sigma, \delta, \varepsilon) - \sqrt{2\rho_{max}(\sigma, \delta, \varepsilon)} - \frac{\Delta_\mu}{2} \right) \tag{18}$$

*holds for any pair of $(\ell, k)$ with $\ell \neq k$, then the downstream error rate of NN classifier $G_f$*

$$\mathrm{Err}(G_f) \leq (1 - \sigma) + R_\varepsilon + t, \tag{19}$$

*where $\rho_{max}(\sigma, \delta, \varepsilon) = 2(1 - \sigma) + \frac{R_\varepsilon}{\min_\ell p_\ell} + \sigma\left(\frac{L\delta}{r} + \frac{2\varepsilon}{r}\right)$ and $\Delta_\mu = 1 - \min_{k \in [K]} \|\mu_k\|^2 / r^2$.*

We remark that the definition of $R_\varepsilon$ is unchanged. The above result gives a more general bound

$$\mathrm{Err}(G_f) \leq (1 - \sigma) + R_\varepsilon + t$$

by taking into account the correctly augmented part. An interesting trade-off between $t$ and $(\sigma, \delta)$ emerges. Increasing the strength of data augmentation leads to better concentration, but also a larger $t$. For extremely strong augmentations, $t$ could be large and dominate the above bound, hence the performance could decrease. We leave the detailed study of this trade-off to future work.

*Proof of Theorem 6.* Let $\tilde{R}_\varepsilon := \mathbb{P}\left[\overline{S_\varepsilon \cap (\cup_{k \in [K]}\tilde{C}_k)}\right]$. Then using Theorem 1 on $\cup_{k=1}^K \tilde{C}_k$ directly gives

$$\text{Err}(G_f) \leq (1 - \sigma) + \tilde{R}_\varepsilon.$$

Moreover,

$$
\begin{aligned}
\tilde{R}_\varepsilon &= \mathbb{P}\left[\overline{S_\varepsilon \cap (\cup_{k \in [K]}\tilde{C}_k)}\right] \\
&= \mathbb{P}\left[\overline{S_\varepsilon} \cup \overline{(\cup_{k \in [K]}\tilde{C}_k)}\right] \\
&\leq \mathbb{P}\left[\overline{S_\varepsilon}\right] + \mathbb{P}\left[\overline{\cup_{k \in [K]}\tilde{C}_k}\right] \\
&= R_\varepsilon + t,
\end{aligned}
$$

which completes the proof. □

## G   EXTENSIONS TO MAE, CLIP AND BYOL

In this section, we discuss how to apply our framework to MAE (He et al., 2022), CLIP (Radford et al., 2021) and BYOL (Grill et al., 2020).

### G.1   MAE

MAE learns representations by recovering the original image from its randomly masked version. By viewing random mask as data augmentation, Zhang et al. (2022) has shown that MAE implicitly aligns positive pairs as contrastive learning. Specifically, let $g$ and $f$ be the decoder and encoder of MAE respectively. The loss function of MAE is

$$\mathcal{L}_{\text{MAE}}(g \circ f) = \mathbb{E}_{\boldsymbol{x}} \mathbb{E}_{\boldsymbol{x}_1 \in A(\boldsymbol{x})} \|g(f(\boldsymbol{x}_1)) - \boldsymbol{x}\|^2,$$

where $A(x)$ denotes random masks of $x$. Then using Theorem 3.4 in (Zhang et al., 2022) under their conditions gives

$$\mathcal{L}_{\text{MAE}}(g \circ f) \geq C_1 \cdot \mathcal{L}_{\text{align}}(f) + C_2,$$

where $C_1$ and $C_2$ are constants. Based on this result, our framework applies to MAE naturally. We can use the $(\sigma, \delta)$-notion to characterize the concentration property of random mask, and use Theorem 1 to study how MAE ensures alignment.

As for the divergence term, we have the following result.

**Theorem 7.** *Assume that the decoder $g$ is $L$-bi-Lipschitz, i.e., $\forall (z_1, z_2)$ in the domain of $g$,* $1/L \|z_1 - z_2\|^2 \leq \|g(z_1) - g(z_2)\|^2 \leq L \|z_1 - z_2\|^2$. *Then for any $\ell, k \in [K]$*

$$\mathbb{E}_{\boldsymbol{x}_1 \in C_\ell} \mathbb{E}_{\boldsymbol{x}_2 \in C_k} \|f(\boldsymbol{x}_1) - f(\boldsymbol{x}_2)\|^2 \geq C \cdot \left[\mathbb{E}_{\boldsymbol{x}_1 \in C_\ell} \mathbb{E}_{\boldsymbol{x}_2 \in C_k} \|\boldsymbol{x}_1 - \boldsymbol{x}_2\|^2 - \mathcal{L}_{\text{MAE}}(g \circ f)\right],$$

*where $C$ is some constant. If we further assume $\|f(x)\| = 1$ for every $x$, we have*

$$\mu_k^\top \mu_\ell \leq 1 - C \cdot \left[\mathbb{E}_{\boldsymbol{x}_1 \in C_\ell} \mathbb{E}_{\boldsymbol{x}_2 \in C_k} \|\boldsymbol{x}_1 - \boldsymbol{x}_2\|^2 - \mathcal{L}_{\text{MAE}}(g \circ f)\right]$$

The $L$-bi-Lipschitz assumption follows (Zhang et al., 2022). This result shows that, the divergence bound of MAE contains both the MAE loss and an addition term $E_{\boldsymbol{x}_1 \in C_\ell} \mathbb{E}_{\boldsymbol{x}_2 \in C_k} \|\boldsymbol{x}_1 - \boldsymbol{x}_2\|^2$, which measures the class distances between original images. If the original images already have large class distances, the divergence can be ensured.

More refined results of MAE may need more additional effort.

*Proof of Theorem 7.* For any $\ell, k \in [K]$, by the $L$-bi-Lipschitz property of $g$ we have

$$\mathop{\mathbb{E}}_{\boldsymbol{x}_1 \in C_\ell} \mathop{\mathbb{E}}_{\boldsymbol{x}_2 \in C_k} \|f(\boldsymbol{x}_1) - f(\boldsymbol{x}_2)\|^2$$

$$\geq 1/L \cdot \mathop{\mathbb{E}}_{\boldsymbol{x}_1 \in C_\ell} \mathop{\mathbb{E}}_{\boldsymbol{x}_2 \in C_k} \|g(f(\boldsymbol{x}_1)) - g(f(\boldsymbol{x}_2))\|^2$$

$$= 1/L \cdot \mathop{\mathbb{E}}_{\boldsymbol{x}_1 \in C_\ell} \mathop{\mathbb{E}}_{\boldsymbol{x}_2 \in C_k} \|g(f(\boldsymbol{x}_1)) - \boldsymbol{x}_1 + \boldsymbol{x}_1 - \boldsymbol{x}_2 + \boldsymbol{x}_2 - g(f(\boldsymbol{x}_2))\|^2$$

$$\geq 1/(3L) \cdot \mathop{\mathbb{E}}_{\boldsymbol{x}_1 \in C_\ell} \mathop{\mathbb{E}}_{\boldsymbol{x}_2 \in C_k} \left[\|\boldsymbol{x}_1 - \boldsymbol{x}_2\|^2\right] - 1/L \cdot \mathop{\mathbb{E}}_{\boldsymbol{x}_1 \in C_\ell} \mathop{\mathbb{E}}_{\boldsymbol{x}_2 \in C_k} \left[\|g(f(\boldsymbol{x}_1)) - \boldsymbol{x}_1\|^2 + \|\boldsymbol{x}_2 - g(f(\boldsymbol{x}_2))\|^2\right]$$

$$\geq 1/(3L) \cdot \mathop{\mathbb{E}}_{\boldsymbol{x}_1 \in C_\ell} \mathop{\mathbb{E}}_{\boldsymbol{x}_2 \in C_k} \|\boldsymbol{x}_1 - \boldsymbol{x}_2\|^2 - 1/L \cdot \left(\frac{1}{p_k} + \frac{1}{p_\ell}\right) \cdot \mathcal{L}_{\mathrm{MAE}}(g \circ f).$$

If $\|f(x)\| = 1$ for any $x$, we have

$$\mathop{\mathbb{E}}_{\boldsymbol{x}_1 \in C_\ell} \mathop{\mathbb{E}}_{\boldsymbol{x}_2 \in C_k} \|f(\boldsymbol{x}_1) - f(\boldsymbol{x}_2)\|^2 = 2 - 2\mu_\ell^\top \mu_k.$$

Then we obtain

$$\mu_k^\top \mu_\ell \leq 1 - \frac{1}{6L} \cdot \mathop{\mathbb{E}}_{\boldsymbol{x}_1 \in C_\ell} \mathop{\mathbb{E}}_{\boldsymbol{x}_2 \in C_k} \|\boldsymbol{x}_1 - \boldsymbol{x}_2\|^2 - \frac{1}{2} \cdot \left(\frac{1}{p_k} + \frac{1}{p_\ell}\right) \cdot \mathcal{L}_{\mathrm{MAE}}(g \circ f)$$

$\square$

## G.2 CLIP

CLIP firstly constructs positive samples by image-text pairs, and then minimizes InfoNCE loss. If we view texts as data augmentation of images, our theory applies directly. To be specific, let $T(x)$ denote the set of all possible texts corresponding to image $x$. In this case, the augmented distance between images (parallel to equation (1) in our paper) can be defined by

$$d_T(x_1, x_2) = \min_{t_1 \in T(x_1), t_2 \in T(x_2)} \|t_1 - t_2\|,$$

where $\|\cdot\|$ is some norm of the text space. Then the $(\sigma, \delta)$ notion can also be extended as follows

**Definition 4** $((\sigma, \delta)$-Concentration of image-text pair)**.** We say the image-text pair is $(\sigma, \delta)$-concentrated, if there exists $C_k^0 \subseteq C_k$ such that

$$P(x \in C_k^0) \geq \sigma P(x \in C_k) \text{ and } \sup_{x_1, x_2 \in C_k^0} d_T(x_1, x_2) \leq \delta.$$

Note that we treat texts and images asymmetrically, i.e., we view texts as augmentation of images. The reason is that the information density in texts is larger than that in images, namely, images contain more redundant information. Therefore, for two images in the same class, their corresponding texts are expected to be close to each other, but not vice versa. Based on this model, all of our theoretical results about InfoNCE loss apply to CLIP.

## G.3 BYOL

BYOL (and SimSiam) adopts training strategies to avoid feature collapse instead of involving an explicit $\mathcal{L}_2$ term in its loss function. Besides, its network architecture contains a predictor, so its loss can not be formulated to the common alignment loss. For these reasons, whether BYOL can optimize alignment and divergence cannot be answered directly by our theory. Nevertheless, our $(\sigma, \delta)$-notion and Thm 1 still apply to BYOL, since they are algorithm independent. Besides, our experiments for SimSiam (which is similar to BYOL) indeed verify this:

- Tables 1 and 2 show that the performance of SimSiam also gets better as the concentration of augmentation gets better, which meets our Theorem 1 and related conclusions.
- The new experimental results in Figure 4b show that SimSiam does implicitly optimize divergence during its optimization procedure.

Further theoretical study of BYOL and SimSiam requires additional effort.

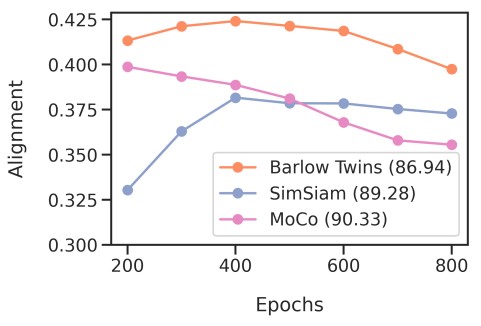

**(a)** Alignment after different training epochs

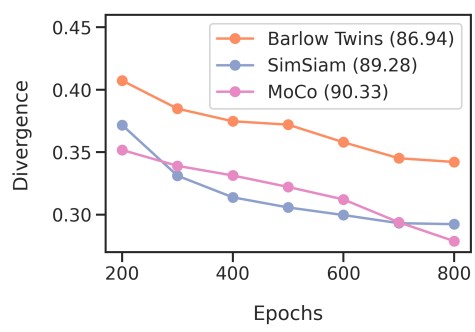

**(b)** Divergence after different training epochs

Figure 4: Alignment and divergence vary with training epochs.

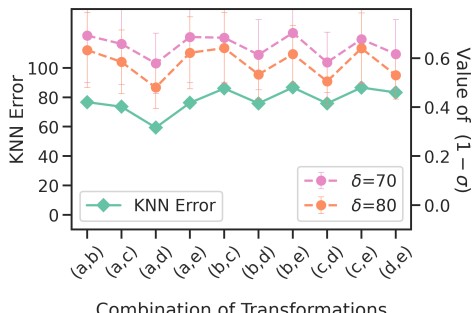

Figure 5: The correlation between observed $\mathrm{Err}(G_f)$ and computed value of $(1 - \sigma)$ on CIFAR-100.

## H ADDITIONAL EXPERIMENTS

**Alignment and Divergence.** We choose the models with three different loss functions (i.e., Barlow Twins, SiamSiam and MoCo) and observe how alignment and divergence change during the training procedure. Each model is trained on CIFAR-10 with a batch size of 512 and 800 epochs. The setting of data augmentation is fixed as the one that Chen et al. (2020a) used. In the experiments, the alignment is quantified by $\frac{1}{|X|} \sum_{x \in X} \|f(A_1(x)) - f(A_2(x))\|^2$ and the divergence is quantified by the average of $\left( \frac{1}{|C_k|} \sum_{x \in C_k} f(x) \right)^\top \left( \frac{1}{|C_\ell|} \sum_{x \in C_\ell} f(x) \right)$ among all $k \neq \ell$, where $f$ is the encoder, $A_1(x)$ and $A_2(x)$ are two augmented data of $x$, $X$ is the training set, $C_k$ contains all the training data with label $k$. At the end of the training procedure, Barlow Twins, SimSiam, and MoCo achieve a KNN accuracy of 86.94, 89.28, and 90.33, respectively.

We have the following observations:

- At the end of the training, both the alignment and divergence are ordered as MoCo < SimSiam < Barlow Twins, from small to large (i.e., good to bad). We also observe that MoCo, Simsiam, and Barlow Twins achieve a KNN accuracy of 90.33, 89.28, and 86.94, respectively. This suggests that better alignment and divergence result in better performance when the setting of data augmentation is fixed. This empirical result is as expected: as long as good alignment and divergence are achieved, no matter whether it is due to the loss functions (e.g., SimCLR and Barlow Twins, proved in Section 4) or other unknown reasons (e.g., SimSiam), the generalization error should be small according to our Thm 1.

- During the training procedure, the divergence factor always gets better (i.e., smaller) for all three kinds of algorithms. It decreases more quickly at the early stage of training. Meanwhile, the alignment factor starts to get better monotonously after several training epochs for MoCo and Barlow Twins. But for SimSiam, it becomes increasingly large. This is because SimSiam does not directly minimize the alignment, instead, a predictor is involved to transform the feature of one view and matches it to the other view.

**Different Composed Pairs of Transformations on CIFAR-100.** Similar to the experiments on CIFAR-10, we compose transformations (a)-(e) in pairs to construct a total of 10 augmentations, and observe the correlation between classification error rate $\text{Err}(G_f)$ and $(1 - \sigma)$ under different $\delta$ on CIFAR-100, based on the SimCLR model trained with 200 epochs. We find that downstream performance is also highly correlated to the concentration level on CIFAR-100, which has the similar result to Figure 3.

