# OpenReview forum: "Towards the Generalization of Contrastive Self-Supervised Learning"
_ICLR.cc/2023/Conference — ICLR 2023 poster_

### Official Review · Reviewer_Lkqf · 2022-10-17

**Confidence:** 4
**Correctness:** 3
**Technical Novelty And Significance:** 3
**Empirical Novelty And Significance:** Not applicable
**Recommendation:** 5

**Clarity, Quality, Novelty And Reproducibility:**

The paper is relatively novel, clear with reasonable quality.

The paper can be reproduced relatively easily.

**Strength And Weaknesses:**

Strength
1. The paper works on an important problem in contrastive learning: how the data augmentation plays the role in learning good representation.
2. The paper builds theoretical connections between different parts of loss functions to be optimized and the corresponding latent class structure.
3. The paper is well-written with clear motivations and explanation.

Weakness
1. Claim in the intro doesn't quite match the content.

While the author claims that they also consider a third factor concerning the concentration of data augmentation, in addition to the existing two factors, i.e., alignment and diversity / uniformity in the sphere, as in (Wang and Isola, 2020). In the analysis, it seems that the contrastive loss is still decomposed into two terms, and concentration of data augmentation serves as the additional terms in the upper bounds in theorems. It would be great if the paper can be revised to make the contribution more clear and precise.

2. No comparison between InfoNCE, Barlow Twins and t-InfoNCE.

From the analysis, it looks like all losses have two components that can bound the property of latent class models. Then a natural question is what's the pros and cons of InfoNCE, versus Barlow Twins and t-InfoNCE? In which scenarios one is better than others? It would be great if the authors could give empirical guidance.

3. More experiments can be done to verify the points.

The experiments in Tab. 1-2 are largely common sense: researchers in SSL know that large augmentation helps downstream tasks, without referring to the proposed theory. While Fig. 3 is good, more ablation studies are needed to verify components of the theory. It would be great if the authors could provide experiments on synthetic datasets with the analyzed loss functions (InfoNCE, Barlow Twins and t-InfoNCE), in which the data exactly follow the latent classes assumption with ground truth (and known) concentration, and verify the theory in more details.

4. Issues in mathematical rigidity.

The are some issues in the definition. $\cap_{k=1}^K A(C_k) = \emptyset$ doesn't mean $A(C_k)$ and $A(C_j)$ are pairwise disjoint.

The nearest neighbor classifier (last equation in Page 3) CANNOT be reformulated to be linear classifier, because what if several class centers $\{\mu_k\}$ are co-linear (i.e., there exists a common vector $v$ so that $\mu_k = \lambda_k v$) , then linear classifier can only predict the two "end-points" (i.e., $\arg\max_k \lambda_k$ and $\arg\min_k \lambda_k$) since the score of other classes will already be dominated by the two end-points. However, NN can predict these classes with ease.

Some notations are really confusing. E.g., the definition of $\mathcal{L}_1$ and $\mathcal{L}_2$ should be different for InfoNCE (Eqn. 5) and for Barlow Twins, yet they use the same notation, making it hard to understand.

**Summary Of The Paper:**

The paper works on contrastive self-supervised learning (SSL) and gives several bounds between the assumed latent class structure of the dataset in terms of different parts of the loss functions. For example, bounds of angles between two cluster centers $\mu_j^\top\mu_k$ (Theorem 1, Theorem 3, Theorem 4), size of the regions where the representation of the input x is different from its view (Theorem 2)), in terms of different loss terms, such as (1) feature alignment within augmentation, and (2) divergence metric across views of different samples. Note that (1) and (2) are largely defined in (Wang and Isola, 2020), and the paper claims that they further study a third term $R_\epsilon$ that represents by the role played by data augmentation, through the lens of a newly proposed concentration metric for augmentation data, i.e., $(\sigma, \delta)$-augmentation. Experiments verify the relationship of the downstream KNN classification performance and the concentration of learned feature for each latent class (Eqn. 3).

**Summary Of The Review:**

Overall the paper has pros and cons and I am on the boundary. If authors address my concerns, I will raise the score.

---

> ### Author Response · Authors · 2022-11-16
> **Response to Reviewer Lkqf (2/2)**
>
> **Q3.** The experiments in Tab. 1-2 are largely common sense. While Fig. 3 is good, more ablation studies are needed to verify components of the theory. More experiments can be done to verify the points.
>
> **A3.** Thanks for the suggestion. First of all, we would like to remark that our empirical results **still provide some interesting insights** beyond the existing works (e.g., SimCLR paper):
>
> 1. The downstream performance monotonously increases with both the richness and strength of augmentations. The above observation can be directly explained by our theory that **richer** and **stronger** transformations lead to **sharper concentration** of augmentations (according to Def 1), hence the better downstream performance (according to Thm 1).
>
> 1. Color **dropping** and **distortion** have a great impact on the performance. **This is also predictable**, since our theory suggests that these two enable the augmented data to vary in a very wide range, making the augmented distance (Eq 1) largely decrease. Note that largely reduced augmented distance (Eq 1) indicates the **much sharper concentration** (Def 1), hence the much better downstream performance (according to Thm 1).
>
> 1. The above two empirical results are all **observed among different contrastive algorithms** (i.e., SimCLR, MoCo, Barlow Twins, SimSiam). This is because concentration of augmented data is pre-defined and can not be optimized by algorithms (Thm 1). Thus, for a given contrastive algorithm including the algorithms preventing collapse by training strategies instead of losses such as **SimSiam**, sharper concentration indicates better downstream performance.
>
>
> 1. Fig 3 shows that the concentration of augmented data is **highly correlated** to real downstream performance. This also provides a theoretical explanation for Figure 5 in SimCLR paper of **why the composition of "crop & color" performs the best**.
>
> Secondly, as suggested by the reviewer, we also conduct **new experiments on CIFAR-100** (Fig 5 in Appendix H) to verify how data augmentation affects performance.
>
> - Similar to the experiments on CIFAR-10, we compose transformations (a)-(e) in pairs to construct a total of 10 augmentations, and observe the correlation between classification error rate $\text{Err}(G_f)$ and $(1-\sigma)$ under different $\delta$ on CIFAR-100, based on the SimCLR model trained with 200 epochs. We find that **downstream performance is also highly correlated to the concentration level on CIFAR-100**, which has a similar result to Fig 3.
>
>
>
> Thirdly, **we also run additional experiments to empirically study the alignment and divergence factors**, which have been shown and discussed in above **A2**.
>
> ---
>
> **Q4.** (1) $\cap_{k=1}^KA(C_k)=\varnothing$ doesn't mean $A(C_k)$ and $A(C_j)$ are pairwise disjoint.
>
> (2) The nearest neighbor classifier (last equation in Page 3) CANNOT be reformulated to be a linear classifier.
>
> (3) The definition of $\mathcal{L}_1$ and $\mathcal{L}_2$ should be different for InfoNCE (Eqn. 5) and for Barlow Twins, yet they use the same notation, making it hard to understand.
>
> **A4.** Thanks for your careful reading and detailed comments!
>
> (1) Yes, it is a typo. We have replaced it with "$A(C_k)\cap A(C_j)=\varnothing$ for any $k\not=j$" in the revised paper.
>
> (2) In fact, the nearest neighbor (NN) classifier **CAN** be reformulated to be a linear classifier, even for your example. If several class centers $\mu_k$ are co-linear, it's true that a single linear classifier can only predict the two "end-points". However, **$K$ parallel linear classifiers ($K$ parallel linear boundaries) can indeed classify them just as NN classifier.** To avoid confusion, we give a detailed formulation and proof of this equivalence here and add them to the appendix in the revised paper.
>
> **Proposition**. Let $G_f(\boldsymbol{x})=\underset{k \in[K]}{\arg \min }\left\\|f(\boldsymbol{x})-\mu_k\right\\|$ be the nearest neighbor (NN) classifier, then we have $G_f(\boldsymbol{x})=\underset{k \in[K]}{\arg \max }\left( \mu_k^\top f(\boldsymbol{x})-\frac{1}{2}\\|\mu_k\\|^2\right)$, which is a linear classifier.
>
> **Proof**. $G_f(\boldsymbol{x})=k$ means that
>     $
> \\|f(\boldsymbol{x}) - \mu_k\\|^2 \leq \\|f(\boldsymbol{x}) - \mu_{\ell}\\|^2
>     $
> for each $\ell\in[K]$. This is equivalent to
>     $
> \mu_k^\top f(\boldsymbol{x}) - \frac{1}{2} \\|\mu_k\\|^2 \geq \mu_{\ell}^\top f(\boldsymbol{x}) - \frac{1}{2} \\|\mu_{\ell}\\|^2
>     $
> holds for each $\ell\in[K]$. Therefore, we have $G_f(\boldsymbol{x})=\underset{k \in[K]}{\arg \max }\left( \mu_k^\top f(\boldsymbol{x})-\frac{1}{2}\\|\mu_k\\|^2\right)$.
>
> (3) Sorry for the confusion. We have modified them to $\mathcal{L}_1^{\text{Info}},\mathcal{L}_2^{\text{Info}}$ and $\mathcal{L}_1^{\text{Cross}},\mathcal{L}_2^{\text{Cross}}$ in the revised paper.

---

> > ### Comment · Reviewer_Lkqf · 2022-12-11
> > **Thanks for your rebuttal**
> >
> > Thanks for your rebuttal. This partly addressed my concerns but I still have additional concerns.
> >
> > The key confusion I have, is whether this paper can be reduced to a simple logic that "more augmentation leads to better results in contrastive learning"? Did your theory give more insightful results than that simple logic? Please give concrete example.
> >
> > Also you mentioned that "richer and stronger transformations lead to sharper concentration of augmentations" (Def. 1), which is true since your $\delta$ is the maximal distances between two augmented samples from the same latent class, but on the other hand, will that lead to violation of your assumption that $A(C_k) \cap A(C_j) = \emptyset$? Please explain.
> >
> > Note that SimSiam is largely not considered as a contrastive algorithm, but more like a non-contrastive method. I am not sure how your theoretical analysis (either InfoNCE Loss or Cross-Correlation Loss) links to it?
> >
> > Also in the conclusion, you still list the incorrect assumption $\bigcap_{k=1}^K A(C_k) = \emptyset$ as the conclusion. Please correct.

---

> > > ### Author Response · Authors · 2022-12-13
> > > **Thanks for your further comments**
> > >
> > > Thanks for your detailed reading and further comments!
> > >
> > > ---
> > >
> > > **Q1.** Whether this paper can be reduced to a simple logic that "more augmentation leads to better results in contrastive learning"
> > >
> > > **A1.** We think that our theory provides more intuition than that. Broadly speaking, our theoretical results are divided into two parts: (i) how data augmentation affects concentration and hence generalization (Def 1 & Thm 1); (ii) how InfoNCE and Cross-Correlation loss lead to better alignment and divergence (Thm 3 & 4).
> > >
> > > For (i), the simplest conclusion is indeed "more augmentation leads to better results in contrastive learning". However, **for different combinations of augmentation** (e.g., random cropping + Gaussian blur & color dropping + color distortion), **it is difficult to compare their strength intuitively.** We propose the $(\sigma,\delta)$-notion to **quantitatively** characterize the strength of augmentation. Both our theory (Thm 1) and experiments (Fig 3&5) illustrate the close relationship between this notion and the generalization error. We also think this notion may have the potential to be used to design better augmentations.
> > >
> > > For (ii), we think our analysis of InfoNCE and Cross-Correlation is also novel. Although there have been many analyses of InfoNCE loss, most of them **rely on the impractical assumption that positive samples are drawn from the same classes.** Our theory for InfoNCE does not need this assumption. Some insights emerge, e.g., how the temperature affects the performance of InfoNCE (the discussion at the end of Sec 4.1). Cross-Correlation loss is rarely discussed in previous works. We also give a detailed analysis of it (Lem 4.1 & Thm 4).
> > >
> > > ---
> > >
> > > **Q2.** More transformations lead to violation of the assumption $A(C_k)\cap A(C_j)=\varnothing$.
> > >
> > > **A2.** The conclusion "richer and stronger transformations lead to sharper concentration of augmentations" **still holds even when the assumption $A(C_k)\cap A(C_j)=\varnothing$ is violated**, since Def 1 does not depend on this assumption. However, when this assumption is not true, **sharper concentration may not always lead to better generalization error**, i.e., Theorem 1 does not hold, since too strong augmentation could lead to wrong signals, i.e., $A(C_k)\cap A(C_j)\not=\varnothing$.
> > >
> > > Nevertheless, we can extend our theory to the case where this assumption is violated. To be specific, considering the **correctly augmented part**, which is defined as
> > > $$
> > > \tilde{C_k} := \{x\in C_k: A(x)\subset C_k\}.
> > > $$
> > > We also define $t := 1 - P(\cup_{k=1}^K \tilde{C_k}),$ which **measures the level of wrong signal**. By this definition, clearly $A(\tilde{C_{\ell}}) \cap A(\tilde{C_k})=\varnothing$ for any $k\neq \ell$. Therefore, **for the part of $\cup_{k=1}^K \tilde{C}_k$, our current theory applies directly.** On its complement, correct classification can not be ensured since data augmentation introduces wrong signals on it. Therefore, $t$ appears as an additional term on the classification error bound, i.e., under the conditions of Theorem 1 we have a more general result:
> > > $$
> > > \operatorname{Err}\left(G_f\right) \leq (1-\sigma)+R_{\varepsilon} + t.
> > > $$
> > > Detailed proof of this generalized result is provided in Appendix F of the revised paper.
> > >
> > > With this result, an interesting **trade-off between $t$ and $(\sigma,\delta)$** emerges. Increasing the strength of data augmentation **still leads to better concentration**, but **also a larger $t$, i.e., more wrong signal**. For extremely strong augmentation, $t$ could be extremely large and dominate the error bound, hence the performance would decrease.
> > >
> > > ---
> > >
> > > **Q3.** How does our theoretical analysis (either InfoNCE Loss or Cross-Correlation Loss) link to SimSiam?
> > >
> > > **A3.** Compared to SimCLR and MoCo, SimSiam adopts training strategies to avoid feature collapse instead of involving an explicit regularization loss. Nevertheless, since **SimSiam also aims to close the distance between positive pairs**, **our theory about the relationship between augmentation and generalization** ($(\sigma,\delta)$-notion and Thm 1) still apply, since they are algorithm independent.
> > >
> > > On the other hand, our **other theoretical analysis (for InfoNCE Loss or Cross-Correlation Loss) cannot apply to SimSiam directly**, and whether SimSiam can optimize alignment and divergence cannot be answered by our existing theory. For this reason, we conduct additional experiments for SimSiam to see how alignment and divergence change during its training.The results show that SimSiam is also able to optimize alignment and divergence. A detailed analysis of these experiments is given in Appendix H.
> > >
> > > ---
> > >
> > > **Q4.** You still list the incorrect assumption in the conclusion.
> > >
> > > **A4.** We apologize for not noticing this typo. We have corrected it.

---

> ### Author Response · Authors · 2022-11-16
> **Response to Reviewer Lkqf (1/2)**
>
> **Q1.** Claim in the intro doesn't quite match the content.
>
> **A1.** Sorry for the confusion. Unlike the alignment and divergence factors which are **properties of representations** that can be optimized by algorithms, the concentration factor is a **property of data augmentation**, which has nothing to do with encoders and also **cannot be affected by algorithms (losses)**. Therefore, it is not involved in the analysis of contrastive losses (Sec 4), and we study it later in Sec 5 through experiments. We have revised the abstract and introduction to make our arguments more precise.
>
> ---
> **Q2.** No comparison between InfoNCE, Barlow Twins and t-InfoNCE.
>
> **A2.** Thanks for the suggestion. It might be meaningless to directly compare different methods by theoretical formula, since their real performance is also significantly influenced by the experimental environment, optimization algorithms, and various other settings.
>
> Nevertheless, **we add new experiments to empirically compare algorithms** in terms of the alignment and divergence properties. We choose the models with three different loss functions (i.e., Barlow Twins, SiamSiam and MoCo) and observe how alignment and divergence change during the training procedure when the setting of data augmentation is fixed (see Fig 4 of Appendix H in the revised pdf).
>
> We have the following interesting observations:
>
> - At the end of the training, both the alignment and divergence are ordered as MoCo < SimSiam < Barlow Twins, from small to large (i.e., good to bad). We also observe that MoCo, Simsiam, and Barlow Twins achieve a KNN accuracy of 90.33, 89.28, and 86.94, respectively. This suggests that **better alignment and divergence result in better performance** when the setting of data augmentation is fixed. This empirical result is as expected: as long as good alignment and divergence are achieved, **no matter** whether it is due to the loss functions (e.g., SimCLR and Barlow Twins, proved in Section 4) or other unknown reasons (e.g., SimSiam), the generalization error should be small according to our Thm 1.
>
> - During the training procedure, the **divergence factor always gets better** (i.e., smaller) for all three kinds of algorithms. It decreases more quickly at the early stage of training. Meanwhile, the **alignment factor starts to get better monotonously after several training epochs** for MoCo and Barlow Twins. But for SimSiam, its alignment gets worse. This is because SimSiam does not directly minimize the alignment, instead, a predictor is involved to transform the feature of one view and matches it to the other view.

---

### Official Review · Reviewer_R8Eq · 2022-10-23

**Confidence:** 3
**Correctness:** 3
**Technical Novelty And Significance:** 4
**Empirical Novelty And Significance:** 3
**Recommendation:** 6

**Clarity, Quality, Novelty And Reproducibility:**

### Clarity
- The authors defined alignment and divergence properties with well-defined formulas and showed InfoNCE and Cross-Correlation loss can be interpreted with these two terms. However, this paper only includes the empirical analysis of the concentration property, which may be more essential. Also, these experiments did not support the author's claims enough.

### Quality
- The paper included plenty of information to analyze the SSL framework in a theoretical way. Especially, the authors tried to prove the Cross-Correlation loss in BT, which is rarely discussed in previous works. I think the quality of this work can be much improved if the authors add more explanations and experiments to support their claims.

### Novelty
- As mentioned above, the three properties to measure the SSL's performance will bring valuable insights to further researchers. Recently, there are some other analyses on the InfoNCE loss, and updating recent works in this paper can make this work more novel.

### Reproducibility
- The authors used popular frameworks and well-known augmentation techniques in computer vision. The proofs of their claims are represented in the appendix, so the reader can follow them to understand the authors' claims.


**Strength And Weaknesses:**

## Strengths
+ Theoretical analyses on the effect of augmentations are less often than ones on the negative samples recently. This paper may give some insights into further research on both positive and negative samples in SSL.
+ There were few attempts to analyze the mechanism of SSL frameworks without negative samples such as BYOL and Barlow Twins (BT). To prove the robustness of SSL models, the integrated perspective which includes both InfoNCE and non-InfoNCE-based frameworks will be needed and this paper can be the pioneer of it.
+ The authors defined the three important properties which can represent the status of distribution of the features. Their concept can be widely used to further researchers who analyze similar problems.

## Weaknesses
- I think that lack of information on recent works may decline their work's novelty. Recently, some papers analyzed the relationship between SSL loss function and supervised learning loss with the collision phenomena such as [1]. In [1], the authors showed that the upper bound of supervised learning loss contains the intra-class variance term which is similar to the concentration property in this paper. I carefully suggest including more recent works which tried to solve a similar problem.
- The authors used popular, but somewhat old-fashioned augmentations to validate their claims. One of the main results of this paper is stronger augmentations will bring higher performance. However, another recent work [2] said the dramatic distortion of the data can harm the essential information for the downstream tasks (in this case, classification), and it will degrade the performance in the end. This implies that their claim can be right in their experiment settings, but it may not be applied always. Conducting more experiments with other datasets or stronger augmentations such as adversarial attack-based augmentations can be a good way to justify the authors' claims.
- The authors analyzed alignment and divergence properties in theoretical ways. However, there are no experiments that support their theory. Instead, there are only experiments for the empirical study of concentration property. In general, we cannot be sure that a property is always preserved when the model is trained with a loss with it. This means that there may still exist some performance drops caused by the former two properties (alignment and divergence) in the experiments on the concentration property. I suggest adding more analyses to show the trained model satisfies the former two properties in every augmentation set.

[1] Ash, Jordan, et al. "Investigating the Role of Negatives in Contrastive Representation Learning." International Conference on Artificial Intelligence and Statistics. PMLR, 2022.

[2] Yang, Kaiwen, et al. "Identity-Disentangled Adversarial Augmentation for Self-supervised Learning." International Conference on Machine Learning. PMLR, 2022.

**Summary Of The Paper:**

In this paper, the authors analyzed the Self-supervised learning frameworks such as SimCLR, MoCo, and Barlow Twins in perspective of augmentations. (sigma-delta) augmentation was defined to indicate how close the augmented data with the same class are. The authors suggested three important properties to analyze the performance of SSL: alignment, divergence, and concentration. Common SSL losses, InfoNCE, and Cross-Correlation loss can be split into smaller components which are represented as the optimization terms of the alignment and divergence properties. In experiments, various augmentation settings with different strengths were used for comparing the downstream performance with them. In CIFAR-10 and CIFAR-100, the model trained with stronger augmentations can be more robust than with weaker ones. The stronger augmentation can imply a more concentrated cluster of each class.


**Summary Of The Review:**

The authors defined (sigma-delta) augmentation and three important properties to measure the SSL model's performance. InfoNCE and Cross-Correlation loss functions can be separated into two components that optimize the alignment and divergence properties respectively. To show the correlations between concentration property and augmentation, the authors conducted several experiments with various augmentation sets with different intensities. Updating more recent works and adding more exact analyses or experiments on each property can dramatically enhance this work's novelty.

---

> ### Author Response · Authors · 2022-11-16
> **Response to Reviewer R8Eq (2/2)**
>
> **Q2.** One of the main results of this paper is stronger augmentations will bring higher performance. However, another recent work [3] said the dramatic distortion of the data can harm the essential information for the downstream tasks (in this case, classification), and it will degrade the performance in the end.
>
> **A2.** We thank the reviewer for providing the related recent work.
> It is true that dramatically strong augmentation will harm the downstream performance as [3] shown.
> However, **this does not contradict our theory**. This is because our theory assumes that samples from different classes never transfer to the same augmented sample, i.e., $A(C_k)\cap A(C_j)=\varnothing$ for any $k\not=j$.
> **This assumption does not hold any longer when the augmentation is too strong**, which is beyond the scope of this paper.
>
> In fact, **our theory can be extended to this case easily** by considering the **correctly augmented part**, which can be defined as
> $\tilde{C}\_k := \\{ x\in C\_k: A(x)\subset C\_k \\}$.
> We also define $t := 1 - P(\cup\_{k=1}^K \tilde{C}\_k).$ By this definition, clearly $A(\tilde{C}\_{\ell}) \cap A(\tilde{C}\_k)=\varnothing$ for any $k\neq \ell$. Therefore, **for the part of $\cup_{k=1}^K \tilde{C}_k$, our theory applies directly.** On its complement, correct classification can not be ensured since data augmentation introduces wrong signals on it. Therefore, $t$ appears as an additional term on the classification error bound, i.e., under the conditions of Theorem 1 we have a more general result:
> $
> \operatorname{Err}\left(G_f\right) \leq (1-\sigma)+R_{\varepsilon} + t.
> $
> Note that some definitions are modified here to apply to the correctly augmented parts $\tilde{C}_k$. Detailed proof of this generalized result is provided in the appendix of the revised paper.
>
> With the above result, an interesting **trade-off between $t$ and $(\sigma,\delta)$** emerges. Increasing the strength of data augmentation leads to better concentration, but also a larger $t$. For extremely strong augmentation, $t$ could be extremely large and dominate the error bound, hence the performance would decrease. **Through this extension, both of the experiments in [3] and our paper agree with our theory**.
>
>
> ---
>
> **Q3.** The authors analyzed alignment and divergence properties in theoretical ways. However, there are no experiments that support their theory. I suggest adding more analyses to show the trained model satisfies the former two properties in every augmentation set.
>
> **A3.** Thanks for the suggestion.  **we run additional experiments to empirically study the alignment and divergence factors**. We choose the models with three different loss functions (i.e., Barlow Twins, SiamSiam and MoCo) and observe how alignment and divergence change during the training procedure when the setting of data augmentation is fixed (see Fig 4 of Sec H in the revised pdf).
>
> We have the following interesting observations:
>
> - At the end of the training, both the alignment and divergence are ordered as MoCo < SimSiam < Barlow Twins, from small to large (i.e., good to bad). We also observe that MoCo, Simsiam, and Barlow Twins achieve a KNN accuracy of 90.33, 89.28, and 86.94, respectively. This suggests that **better alignment and divergence result in better performance** when the setting of data augmentation is fixed. This empirical result is as expected: as long as good alignment and divergence are achieved, no matter whether it is due to the loss functions (e.g., SimCLR and Barlow Twins, proved in Section 4) or other unknown reasons (e.g., SimSiam), the generalization error should be small according to our Thm 1.
>
> - During the training procedure, the **divergence factor always gets better** (i.e., smaller) for all three kinds of algorithms. It decreases more quickly at the early stage of training. Meanwhile, the **alignment factor starts to get better monotonously after several training epochs** for MoCo and Barlow Twins. But for SimSiam, it becomes increasingly large. This is because SimSiam does not directly minimize the alignment, instead, a predictor is involved to transform the feature of one view and matches it to the other view.
>
> ---
> **References:**
>
> [1] Ash, Jordan, et al. "Investigating the Role of Negatives in Contrastive Representation Learning." International Conference on Artificial Intelligence and Statistics. PMLR, 2022.
>
> [2] Arora, Sanjeev, et al. "A theoretical analysis of contrastive unsupervised representation learning." *arXiv preprint arXiv:1902.09229* (2019).
>
> [3] Yang, Kaiwen, et al. "Identity-disentangled adversarial augmentation for self-supervised learning." International Conference on Machine Learning. PMLR, 2022.

---

> > ### Comment · Reviewer_R8Eq · 2022-11-16
> > **Thank you for your response**
> >
> > Dear Authors, thanks for the response and the revised manuscript. These helped me a lot. I don't have any additional questions at this point (will update the score as soon as I am allowed to do that).

---

> > > ### Author Response · Authors · 2022-11-16
> > > **Response to Reviewer R8Eq**
> > >
> > > We are glad that our response solves your concern and thank you for appreciating our work!

---

> ### Author Response · Authors · 2022-11-16
> **Response to Reviewer R8Eq (1/2)**
>
> **Q1.** I think that a lack of information on recent works may decline their work's novelty. Recently, some papers analyzed the relationship between SSL loss function and supervised learning loss with the collision phenomena such as [1]. I carefully suggest including more recent works which tried to solve a similar problem.
>
> **A1.** Thanks for reminding us of this interesting related work. We have cited it and **carefully compared it with our work** below (also in our revised paper). Overall, there are several significant differences between them, so our novelty might not be declined.
>
> [1] mainly studies the role of negative samples in contrastive learning. Theoretically, they show an interesting collision-coverage trade-off, which suggests that the optimal number of negative examples should scale with the number of underlying concepts in the data. Although both [1] and our paper aim to study contrastive learning from a theoretical perspective, we highlight the following significant differences:
>
> (1) **Different focus.** As the abstract of [1] concluded, "the success of modern contrastive learning pipelines relies on many parameters such as the choice of data augmentation, the number of negative examples...". The focus of [1] is the **number of negative samples,** while ours is **data augmentation.** By taking the number of negative samples into account, [1] established a relationship between NCE loss and downstream supervised loss. We propose a **new framework to study data augmentation,** and use it to study the classification error associated with **InfoNCE, Cross-Correlation and t-InfoNCE.** Therefore, [1] and our paper study contrastive learning from two orthogonal perspectives.
>
> (2) **Different data generation models.** [1] assumed that **positive samples are drawn from the same latent classes,** which is an overly ideal model adopted by early contrastive learning theories [2]. Namely, it assumes that the positive samples are drawn i.i.d. from the same class. In practice, positive pairs are constructed by data augmentation instead of labels. **One of our contributions is to propose the $(\sigma,\delta)$-formulation to quantify data augmentation**, such that the above unrealistic assumption can be avoided.
>
> (3) **Difference between intra-class variance and concentration.** The theory in [1] contains an intra-class variance term
> $$
> s(f):=\underset{c \sim \rho}{\mathbb{E}}\left[\underset{x \sim D_c}{\mathbb{E}}[\|f(x)\|] \sqrt{\|\Sigma(f, c)\|_2}\right].
> $$
> This quantity is a function of the embedding $f$, and is used to quantify its intra-class variance. In contrast, our concentration is a property of data augmentation, and is used to measure how well the augmented data are concentrated. These two quantities are **significantly different**, since **$s(f)$ is a function of embedding $f$**, while **concentration is completely independent of $f$**.

---

### Official Review · Reviewer_kpZM · 2022-10-28

**Confidence:** 3
**Correctness:** 4
**Technical Novelty And Significance:** 4
**Empirical Novelty And Significance:** 2
**Recommendation:** 6

**Clarity, Quality, Novelty And Reproducibility:**

Clarity: the paper is mathematically heavy but clear and easy to follow as motivations, intuitions behind the theorems and exemples are provided.

Novelty: the paper clearly position itself in the theoretical self-supervised learning literature, and provide useful and new insights that are valuable for the community

**Strength And Weaknesses:**

Strength:

1) The problem is clearly stated and properly mathematically defined, which allows a formal analysis on the role of alignment, divergence, and data augmentation in the downstream performance of some popular self-supervised learning algorithms. The paper provide theoretical understanding on why these components are essential and suffisant to learn meaningful representations.

2) The proposed framework could be used to analyse other self-supervised methods based on an explicit collapse prevention term in the loss, by simply showing how the quantity $\mu_k^T \mu_l$ is bounded by the term. Garanties on the performance of the downstream classifier can be derived automatically.

3) Showing experimentally the correlation between $\mathrm{Err}(G_f)$ and $1 - \sigma$ is nice. Having more experimental results on the relation with $\delta$ as well would be even better.



Weaknesses:

1) Some self-supervised learning methods are hard to modelize in the proposed framework. For exemple BYOL and SimSiam which have no explicit collapse prevention mechanism in their loss function. How would you derive similar bounds for these methods ?

2) It is hard to tell how tight the bounds are in practice. Could you derive practical insights from your theoretical analysis ? For exemple design better data augmentations ?

3) The experimental analysis and the experimental results of Table 1 and Table 2 are very basic and already known results.


Remarks and Questions:

1) How would you tackle the case where the $A(C_k)$ intersect ?

2) What is the intuition behind the "Main part" in Definition 1 ? Is it a critical detail ?

3) "Thus, the semantic distance can be partially characterised by the proposed augmented distance". I believe semantic is much more complex than visual ressemblance. Making this assumption would not work for less fine-grained classification tasks where two samples from the same class are not necessarily visually similar. This might be an inherent shortcoming of these methods based on learning invariances to data augmentations.


**Summary Of The Paper:**

This paper study the generalisation capabilities of contrastive self-supervised learning models, showing that the data augmentation strategy is key to generalisation. They decompose the problem of learning visual representations with siamese networks into 3 crucial parts, alignment of positive samples, divergence of class centers, and concentration of augmented data. By defining the problem formally, they derive upper bounds on the performance of a downstream classifier, that depends on the expressivity of the chosen data augmentation.


**Summary Of The Review:**

The theoretical contribution of the paper is significant and valuable as the proposed framework could be used as a basis to analyse other self-supervised learning methods. However it is unclear how it could lead to practical insights that could improve these algorithms. For these reasons I recommend the score of 6.

---

> ### Author Response · Authors · 2022-11-16
> **Response to Reviewer kpZM (3/3)**
>
>
> **Q4.** How would you tackle the case where the $A(C_k)$ intersects?
>
> **A4.** If the $A(C_k)$ intersects, it means that the data augmentation can produce wrong positive pairs, e.g., an image of a dog is augmented to a cat.
> In fact, **our theory can be extended to this case easily** by considering the **correctly augmented part**, which can be defined as
> $\tilde{C}\_k := \\{ x\in C\_k: A(x)\subset C\_k \\}$.
> We also define $t := 1 - P(\cup\_{k=1}^K \tilde{C}\_k).$ By this definition, clearly $A(\tilde{C}\_{\ell}) \cap A(\tilde{C}\_k)=\varnothing$ for any $k\neq \ell$. Therefore, **for the part of $\cup_{k=1}^K \tilde{C}_k$, our theory applies directly.** On its complement, correct classification can not be ensured since data augmentation introduces wrong signals on it. Therefore, $t$ appears as an additional term on the classification error bound, i.e., under the conditions of Theorem 1 we have a more general result:
> $
> \operatorname{Err}\left(G_f\right) \leq (1-\sigma)+R_{\varepsilon} + t.
> $
> Note that some definitions are modified here to apply to the correctly augmented parts $\tilde{C}_k$. Detailed proof of this generalized result is provided in the appendix of the revised paper.
>
> With the above result, an interesting **trade-off between $t$ and $(\sigma,\delta)$** emerges. Increasing the strength of data augmentation leads to better concentration, but also a larger $t$. For extremely strong augmentation, $t$ could be extremely large and dominate the error bound, hence the performance would decrease.
>
>
> ---
>
> **Q5.** What is the intuition behind the "Main part" in Definition 1? Is it a critical detail?
>
> **A5.** The "main part" in Definition 1 is used to **make this notion more flexible** and is **not a critical detail**.
> In fact, one can discard the "main part" and define the $\delta$ for the whole class directly: for each $k$, $\delta$ satisfies $\sup _{\boldsymbol{x}_1, \boldsymbol{x}_2 \in C_k} d_A\left(\boldsymbol{x}_1, \boldsymbol{x}_2\right) \leq \delta$. This is equivalent to set $\sigma=1$ in Definition 1. Therefore, our theory still works.
>
> **We remark that introducing the main part is to make our definition more flexible.** For example, consider there are a few outlying data points such that the $d_A$ distances between them are very large. If we consider the whole class, the $\delta$ is also very large. However, if we introduce the "main part", we can obtain a small $\delta$ and small $1-\sigma$ simultaneously, and the corresponding classification error bound is much tighter.
>
> ---
>
> **Q6.** "Thus, the semantic distance can be partially characterized by the proposed augmented distance". I believe semantic is much more complex than visual resemblance. Making this assumption would not work for less fine-grained classification tasks where two samples from the same class are not necessarily visually similar. This might be an inherent shortcoming of these methods based on learning invariances to data augmentations.
>
> **A6.** We agree with you that for less fine-grained tasks, two samples from the same class may not be visually similar, i.e., **their augmented distance $d_A$ could be large**. Therefore, for these tasks, **the $(\sigma,\delta)$-concentration would also be bad**, leading to **poor performance** of contrastive learning. This suggests us that tailored augmentations can be used to capture more essential semantic information for improving the performance on these tasks.

---

> ### Author Response · Authors · 2022-11-16
> **Response to Reviewer kpZM (2/3)**
>
> **Q3.** The experimental analysis and the experimental results of Table 1 and Table 2 are very basic and already known results. Showing experimentally the correlation between $Err(G_f)$  and $1-\sigma$ is nice. Having more experimental results on the relation with $\delta$ as well would be even better.
>
> **A3.** Thanks for the suggestion. First of all, we would like to remark that our empirical results still provide some interesting insights beyond the existing works (e.g., SimCLR paper):
>
> 1. The downstream performance monotonously increases with both the richness and strength of augmentations. The above observation can be directly explained by our theory that **richer** and **stronger** transformations lead to **sharper concentration** of augmentations (according to Def 1), hence the better downstream performance (according to Thm 1).
>
> 1. Color **dropping** and **distortion** have a great impact on the performance. **This is also predictable**, since our theory suggests that these two enable the augmented data to vary in a very wide range, making the augmented distance (Eq 1) largely decrease. Note that largely reduced augmented distance (Eq 1) indicates the **much sharper concentration** (Def 1), hence the much better downstream performance (according to Thm 1).
>
> 1. The above two empirical results are all **observed among different contrastive algorithms** (i.e., SimCLR, MoCo, Barlow Twins, SimSiam). This is because concentration of augmented data is pre-defined and can not be optimized by algorithms (Thm 1). Thus, for a given contrastive algorithm including the algorithms preventing collapse by training strategies instead of losses such as **SimSiam**, sharper concentration indicates better downstream performance.
>
>
> 1. Fig 3 shows that the concentration of augmented data is **highly correlated** to real downstream performance. This also provides a theoretical explanation for Figure 5 in SimCLR paper of **why the composition of "crop & color" performs the best**.
>
> Secondly, we conduct **new experiments on CIFAR-100** (Fig 5 in Appendix H) to see more empirical results on the relationship between concentration and downstream performance.
>
> - Similar to the experiments on CIFAR-10, we compose transformations (a)-(e) in pairs to construct a total of 10 augmentations, and observe the correlation between classification error rate $\text{Err}(G_f)$ and $(1-\sigma)$ under different $\delta$ on CIFAR-100, based on the SimCLR model trained with 200 epochs. We find that **downstream performance is also highly correlated to the concentration level on CIFAR-100**, which has a similar result to Fig 3.
>
> Besides, as suggested by the reviewers, **we run additional experiments to empirically study the alignment and divergence factors**. We choose the models with three different loss functions (i.e., Barlow Twins, SiamSiam and MoCo) and observe how alignment and divergence change during the training procedure when the setting of data augmentation is fixed (see Fig 4 of Sec H in the revised pdf).
>
> We have the following interesting observations:
>
> - At the end of the training, both the alignment and divergence are ordered as MoCo < SimSiam < Barlow Twins, from small to large (i.e., good to bad). We also observe that MoCo, Simsiam, and Barlow Twins achieve a KNN accuracy of 90.33, 89.28, and 86.94, respectively. This suggests that **better alignment and divergence result in better performance** when the setting of data augmentation is fixed. This empirical result is as expected: as long as good alignment and divergence are achieved, no matter whether it is due to the loss functions (e.g., SimCLR and Barlow Twins, proved in Section 4) or other unknown reasons (e.g., SimSiam), the generalization error should be small according to our Thm 1.
>
> - During the training procedure, the **divergence factor always gets better** (i.e., smaller) for all three kinds of algorithms. It decreases more quickly at the early stage of training. Meanwhile, the **alignment factor starts to continuously get better after several training epochs** for MoCo and Barlow Twins. But for SimSiam, its alignment gets worse. This is because SimSiam does not directly minimize the alignment, instead, a predictor is involved to transform the feature of one view and matches it to the other view.

---

> ### Author Response · Authors · 2022-11-16
> **Response to Reviewer kpZM (1/3)**
>
> **Q1.** Some self-supervised learning methods such as BYOL and SimSiam have no explicit collapse prevention mechanism in their loss function. How would you derive similar bounds for these methods?
>
> **A1.** Thanks for the question. As you point out, BYOL and SimSiam adopt specific training strategies to avoid feature collapse, and thus their loss functions do not contain an explicit regularization term.
> Therefore, how BYOL and SimSiam optimize the divergence cannot be answered directly by our theory.
> Nevertheless, **our general framework**, such as $(\sigma,\delta)$-augmentation and Thm 1, **can still apply to them**, since they are algorithm-independent.
> In other words, as long as good alignment and divergence are achieved, **no matter** whether they are due to the loss functions (e.g., SimCLR, Barlow Twins, t-SimCLR) or other unknown reasons (e.g., BYOL, SimSiam), it is still guaranteed to have good downstream performance.
> Meanwhile, our experiments for SimSiam also provide some empirical verifications:
> - Tables 1 and 2 show that the performance of SimSiam also gets better as the concentration of augmented data gets sharper, which agrees with our Thm 1 and related conclusions.
> - The new experimental results in Fig 4 of Sec H show that SimSiam is ordered between MoCo and Barlow Twins in terms of both alignment and divergence, and its downstream performance is also in the middle, when the concentration is fixed. That means **the downstream error has the exact same order as alignment and divergence**.
>
> In addition, studying how the divergence is guaranteed for BYOL and SimSiam needs additional effort.
> We would like to leave it for future work.
>
> ---
>
> **Q2.** It is hard to tell how tight the bounds are in practice. Could you derive practical insights from your theoretical analysis? For example, design better data augmentations?
>
> **A2.** Firstly, although the tightness is hard to verify in practice, we find that **it may not affect the derived conclusions**.
> In particular, from Thm 1, we derive that alignment, divergence and concentration are three key factors for downstream performance.
> From Thm 3 and 4, we prove that good alignment and good divergence can be achieved in SimCLR, Barlow Twins and t-SimCLR.
> The above derived conclusions have been empirically verified by experiments:
>
> - **better alignment and divergence result in better performance**: For well-trained MoCo, SimSiam, Barlow Twins models, both the alignment and divergence are ordered as MoCo < SimSiam < Barlow Twins (Fig 4 of Sec H), from small to large (i.e., from good to bad). We also observe that MoCo, Simsiam, and Barlow Twins achieve a KNN accuracy of 90.33 > 89.28 > 86.94, respectively. That means the downstream error has the exact same order as alignment and divergence.
>
> - **better concentration results in better performance**: As Table 1 and 2 shown, richer and stronger data augmentations have better downstream performance, under different algorithms (including SimSiam).
>
> - **good alignment and good divergence can be achieved**: As Fig 4 shown, during the training procedure, the divergence factor always gets better (i.e., smaller) for all three kinds of algorithms. Meanwhile, the alignment factor starts to continuously get better after several training epochs for both MoCo and Barlow Twins.
>
>
> Secondly, if we are allowed to access a few extra labeled data from the same data distribution, we can have a rough estimate of concentration for different kinds of augmentations over this data distribution. Therefore, our theory suggests choosing the (combinations of) augmentations with the sharpest concentration to do contrastive learning. **This could be an example of practical guidance based on the theorems**. A specific implementation of the above idea could be future work.

---

### Official Review · Reviewer_qUte · 2022-10-28

**Confidence:** 4
**Correctness:** 4
**Technical Novelty And Significance:** 4
**Empirical Novelty And Significance:** 3
**Recommendation:** 10

**Clarity, Quality, Novelty And Reproducibility:**

- Clarity: This paper is well-written and easy to follow.
- Quality: I read the full paper, including all the theorems, lemmas and most of the proof in appendix. The hypothesis is sound and the math is solid.
- Novelty: This paper proposes a new theoretical framework for understanding contrastive learning. The formulation of data augmentation is novel and simple. Various interesting insights emerge in Section 3 and 4.


**Strength And Weaknesses:**

Strength:
- This paper is well motivated. I can easily get the idea and insights from the theorems. This paper well explains the main phenomenon of contrastive learning that only aligning positive samples is able to gather the samples from the same latent class into a cluster.
- The $(\sigma, \delta)$ formulation of data augmentation is novel. It provides a new yet simple modeling of augmented data compared to the previous graph modeling in Haochen et al [1]. More interestingly, the analysis of this paper can be directly applied to the realistic losses (eg. SimCLR, Barlow Twins, t-InfoNCE) used in practice, while [1] only suits their spectral contrastive loss.
- Based on the proposed $(\sigma, \delta)$ formulation, the generalization analysis of contrastive learning becomes much more natural. Three vital factors are derived from the main result (Thm 1): alignment, divergence, and concentration. The first two factors extend the well-known alignment and uniformity in [2], and match people's intuition better. The third concentration factor is a key concept of this paper. It reveals the prerequisite for contrastive learning to work, since it does not depend on learning algorithms.
- Section 4 is quite interesting in the sense that the analysis dissects various contrastive learning objectives from the view of the proposed theoretical framework. Various interesting insights emerge, such as why logsumexp is crucial to InfoNCE, and how to properly understand multiple contrastive losses commonly used in practice. Especially for Barlow Twins, it is very interesting to see that optimizing the statistical objective actually optimizes the geometric structure of embedding space.
- Although the concentration factor is drived from the upper bound of KNN error, it still can provide some predictions for real downstream performance. For example, it is surprising to see that $1-\sigma$ correlates with KNN error so well (Fig 3) in real-world experiments. It also provides an explaination for the observation in SimCLR paper that "crop&color" is the best (due to the sharp concentration). Moreover, the concentration concept can help to understand why (c) color dropping has a great impact on performance (Table 1).

Weaknesses:
- Compared with [2], the authors claim that the divergence condition can be loosened by better alignment and concentration properties. It seems this is a key difference from [2] but lacks of explainations. Can you provide more explanations on this to help me understand correctly?
- In Table 1&2, if I further increase the strength of augmentation, will the performance decrease? If so, can your theory explain it?
- Additional experiments on large-scale datasets such as ImageNet (especially Fig 3 setting) would be better. But the existing experiments are okay for a theory paper.
- I wonder if the theory can be extended to other self-supervised algorithms such as BYOL [3], MAE [4]. Is the theory applicable to multi-modal contrastive algorithms such as CLIP [5] or only applicable to vision data?

[1] Jeff Z HaoChen, Colin Wei, Adrien Gaidon, and Tengyu Ma. Provable guarantees for self-supervised deep learning with spectral contrastive loss. Advances in Neural Information Processing Systems, 34, 2021.
[2] Tongzhou Wang and Phillip Isola. Understanding contrastive representation learning through alignment and uniformity on the hypersphere. In International Conference on Machine Learning, pp. 9929–9939. PMLR, 2020.
[3] Jean-Bastien Grill, Florian Strub, Florent Altché, Corentin Tallec, Pierre H Richemond, Elena Buchatskaya, Carl Doersch, Bernardo Avila Pires, Zhaohan Daniel Guo, Mohammad Gheshlaghi Azar, et al. Bootstrap your own latent: A new approach to self-supervised learning. arXiv preprint arXiv:2006.07733, 2020.
[4] Kaiming He, et al. Masked autoencoders are scalable vision learners. Proceedings of the IEEE/CVF Conference on Computer Vision and Pattern Recognition. 2022.
[5] Alec Radford, et al. "Learning transferable visual models from natural language supervision." International Conference on Machine Learning. PMLR, 2021.


**Summary Of The Paper:**

This paper develops a new theoretical understanding of generalization for self-supervised contrastive methods. In particular, a $(\sigma, \delta)$ formulation is proposed to quantify the strength of data augmentation. A generalization bound on KNN error is derived based on the $(\sigma, \delta)$ formulation. Three key factors: alignment, divergence, and concentration were proposed, which are closely related to the generalization performance. Then the authors relate the first two factors with various contrastive objectives and prove that such different objectives all implicitly optimize the two factors. Finally, empirical experiments on CIFAR10 and CIFAR100 are conducted to study the effect of the third factor. The authors observe that the generalization performance is highly correlated to the third factor.


**Summary Of The Review:**

Overall, the paper provides a different perspective on the theoretical analysis of the generalization guarantees of contrastive learning by modeling the concentration of augmented data. It also reveals the important role of augmentation in contrastive learning. The hypothesis is sound and the mathematical proof looks good to me. The insights from theorems are very interesting, including the three key factors of generalization, why decorrelating the components of representation results in alignment and divergence, why logsumexp is crucial to InfoNCE, why "crop&color" is the best composition of data augmentation, etc. I really enjoy reading this paper. I believe that this work can enhance people's understanding of contrastive learning, and provide guidance on selecting augmentations and improving existing contrastive learning methods. Self-supervised learning field is usually algorithm-driven and hence seeks more theoretical understanding. Therefore, I strongly recommend to accept this paper.

---

> ### Author Response · Authors · 2022-11-16
> **Response to Reviewer qUte (2/2)**
>
> **Q4.** I wonder if the theory can be extended to other self-supervised algorithms such as BYOL, MAE. Is the theory applicable to multi-modal contrastive algorithms such as CLIP or only applicable to vision data?
>
> **A4.** Thanks for this question about the potential applications of our theoretical framework! Indeed, with minor modifications, **our theory can apply to MAE and CLIP** as well. For BYOL, more effort is needed for understanding its divergence property.
>
> ---
>
> __MAE.__
> MAE learns representations by recovering the original image from its randomly masked views.
> By regarding random masks as data augmentations, a recent work [7] has shown that **MAE implicitly aligns positive pairs as contrastive learning.**
>
> Specifically, let $g$ and $f$ be the decoder and encoder of MAE. Then under mild conditions, using Theorem 3.4 in [7], we have
> $\mathcal{L}\_{\text{MAE}}(g \circ f) \geq C_1 \cdot \mathcal{L}\_{\mathrm{align}}(f) + \mathrm{const}$,
> where $\mathcal{L}\_{\text{MAE}}(g\circ f)=\underset{\boldsymbol{x}}{\mathbb{E}}\underset{\boldsymbol{x}\_1 \in A(\boldsymbol{x})}{\mathbb{E}}\left\\|g(f(\boldsymbol{x}\_1))-\boldsymbol{x}\right\\|^2$, and $A(x)$ denotes random masks of $x$.
>
> **Based on this result, our framework applies to MAE naturally.** We can use the $(\sigma,\delta)$-formulation to characterize the concentration property of random mask, and then use Theorem 2 to study how MAE ensures alignment.
> For the divergence term, we can derive the following **new result**:
>
> $$
> \\|\mu\_{\ell} - \mu\_{k}\\|^2 \leq C\cdot \left[\mathcal{L}\_{\text{MAE}}(g\circ f) +
> \underset{\boldsymbol{x}\_1\in C\_{\ell}}{\mathbb{E}}\underset{\boldsymbol{x}\_2 \in C_k}{\mathbb{E}}\left\\|\boldsymbol{x}\_1-\boldsymbol{x}\_2\right\\|^2 \right].
> $$
>
> Therefore, the divergence bound for MAE contains both the MAE loss and the second term, which measures the class distances between original images. **Its detailed proof is also added to the appendix of our revised paper.** More refined results need additional effort.
>
> ---
>
> __CLIP.__ CLIP first constructs positive samples by image-text pairs, and then minimizes InfoNCE loss. If we **regard texts as augmented data of images**, our theory can be applied directly. To be specific, let $T(x)$ denote the set of all possible texts corresponding to image $x$. In this case, the augmented distance between images (Eq (1) in our paper) can be defined by
> $
> d_{T}(x_1, x_2) = \min_{t_1\in T(x_1),t_2\in T(x_2)} \\|t_1 - t_2\\|,
> $
> where $\\|\cdot\\|$ is some norm of the text space. Then the **$(\sigma, \delta)$-formulation can also be extended**: we say the image-text pair is $(\sigma,\delta)$-concentrated, if there exists $C_k^0\subset C_k$ such that
> $$
> P(x\in C_k^0) \geq \sigma P(x\in C_k)\ \text{ and }\sup_{x_1,x_2\in C_k^0} d_T(x_1,x_2) \leq \delta.
> $$
> Note that **we treat texts and images asymmetrically**, i.e., we regard texts as the augmentation of images. The reason is that the information density in texts is larger than that in images, namely, images contain more redundant information. Therefore, **for two images in the same class, their corresponding texts are expected to be close to each other**, but not vice versa. Based on this model, **all of our theoretical results about InfoNCE** can directly apply to CLIP.
>
> ---
>
> __BYOL/SimSiam.__
> BYOL and SimSiam adopt training strategies to avoid feature collapse instead of involving an explicit regularization term in its loss function. Besides, its network architecture contains a predictor, so its loss can not be formulated to the common alignment loss. For these reasons, how BYOL and SimSiam can optimize alignment and divergence cannot be answered directly by our theory. Nevertheless, **our $(\sigma,\delta)$-formulation and Thm 1 can still apply to them, since they are algorithm-independent**. Meanwhile, our experiments for SimSiam provide some empirical verifications:
> - Tables 1 and 2 show that the performance of SimSiam also gets better as the concentration of augmented data gets sharper, which agrees with our Thm 1 and related conclusions.
> - The new experimental results in Fig 4 show that SimSiam is ordered between MoCo and Barlow Twins in terms of both alignment and divergence, and its downstream performance is also in the middle, when the concentration is fixed. That means **the downstream error has the exact same order as alignment and divergence** for SimSiam.
>
> ---
>
> **References:**
>
> [2] Wang, Tongzhou, and Phillip Isola. "Understanding contrastive representation learning through alignment and uniformity on the hypersphere." International Conference on Machine Learning. PMLR, 2020.
>
> [6] Yang, Kaiwen, et al. "Identity-disentangled adversarial augmentation for self-supervised learning." International Conference on Machine Learning. PMLR, 2022.
>
> [7] Zhang, Qi, Yifei Wang, and Yisen Wang. "How Mask Matters: Towards Theoretical Understandings of Masked Autoencoders." *arXiv preprint arXiv:2210.08344* (2022).

---

> ### Author Response · Authors · 2022-11-16
> **Response to Reviewer qUte (1/2)**
>
> **Q1.** Compared with [2], the authors claim that the divergence condition can be loosened by better alignment and concentration properties. It seems this is a key difference from [2] but lacks of explanations. Can you provide more explanations on this to help me understand correctly?
>
> **A1.** Thanks for the question. The divergence requirement comes from the condition of Thm 1:
> $$
> \mu_{\ell}^{\top} \mu_k < r^2\left(1-\rho\_{\max }(\sigma, \delta, \varepsilon)-\sqrt{2 \rho_{\max }(\sigma, \delta, \varepsilon)}-\frac{\Delta_\mu}{2}\right).
> $$
> The right hand side contains $\rho_{\max}(\sigma,\delta,\varepsilon)$ and $\Delta_{\mu}$, **whose values depend on alignment and concentration.** In particular, these two quantities are defined as:
> $$
> \rho\_{\max }(\sigma, \delta, \varepsilon):=2(1-\sigma)+\frac{R_{\varepsilon}}{\min\_{\ell}  p_{\ell}}+\sigma\left(\frac{L \delta}{r}+\frac{2 \varepsilon}{r}\right),
> \Delta\_\mu:=1-\min\_{k \in[K]}\left\||\mu_k\right\||^2 / r^2.
> $$
> As a consequence of Theorem 2 and Lemma C.1 in the appendix, **both of the two quantities become smaller when alignment and concentration get better**. Therefore, the above requirement of $\mu_{\ell}^\top \mu_k$ becomes easier to satisfy. That is what we mean by "the divergence condition can be loosened by better alignment and concentration properties". We remark that the "uniformity" proposed in [2] does not have this property.
>
> As an intuitive example, we consider the case with perfect alignment (discussed below Theorem 1), where all samples from the same latent class are embedded to a **single point** on the hypersphere. In this case, **an arbitrarily small positive angle (arbitrarily small divergence)** is enough to distinguish them, while "uniformity" (all data uniformly distributed on the hypersphere) is not needed.
>
> ---
>
> **Q2.** In Table 1 & 2, if I further increase the strength of augmentation, will the performance decrease? If so, can your theory explain it?
>
> **A2.** If the strength of augmentation increases further, the performance may decrease. This has been verified empirically in [6]. However, **this does not contradict our theory.** This is because our theory assumes that samples from different classes never transfer to the same augmented sample, i.e., $A(C_k)\cap A(C_\ell)=\varnothing$ for any $k\not=\ell$. This assumption does not hold any longer when the augmentation is too strong, which is beyond the scope of this paper.
>
> In fact, **our theory can be extended to this case easily** by considering the **correctly augmented part**, which can be defined as
> $\tilde{C}\_k := \\{ x\in C\_k: A(x)\subset C\_k \\}$.
> We also define $t := 1 - P(\cup\_{k=1}^K \tilde{C}\_k).$ By this definition, clearly $A(\tilde{C}\_{\ell}) \cap A(\tilde{C}\_k)=\varnothing$ for any $k\neq \ell$. Therefore, **for the part of $\cup_{k=1}^K \tilde{C}_k$, our theory applies directly.** On its complement, correct classification can not be ensured since data augmentation introduces wrong signals on it. Therefore, $t$ appears as an additional term on the classification error bound, i.e., under the conditions of Theorem 1 we have a more general result:
> $
> \operatorname{Err}\left(G_f\right) \leq (1-\sigma)+R_{\varepsilon} + t.
> $
> Note that some definitions are modified here to apply to the correctly augmented parts $\tilde{C}_k$. Detailed proof of this generalized result is provided in the appendix of the revised paper.
>
> With the above result, an interesting **trade-off between $t$ and $(\sigma,\delta)$** emerges. Increasing the strength of data augmentation leads to better concentration, but also a larger $t$. For extremely strong augmentation, $t$ could be extremely large and dominate the error bound, hence the performance would decrease.
>
> ---
>
> **Q3.** Additional experiments on large-scale datasets such as ImageNet (especially Fig 3 setting) would be better. But the existing experiments are okay for a theory paper.
>
> **A3.** We conduct **additional experiments on larger dataset CIFAR-100** and observe similar results to Fig 3 that the classification error rate $Err(G_f)$ is also highly related to $(1-\sigma)$ for the larger dataset (see Fig 5 in the appendix).
>
> * Similar to the experiments on CIFAR-10, we compose transformations (a)-(e) in pairs to construct a total of 10 augmentations, and observe the correlation between classification error rate $\text{Err}(G_f)$ and $(1-\sigma)$ under different $\delta$ on CIFAR-100, based on the SimCLR model trained with 200 epochs. We find that **downstream performance is also highly correlated to the concentration level on CIFAR-100**, which has the similar result to Fig 3.
>
> Moreover, we would like to remark that computing $(\sigma,\delta)$ is very time-consuming since it is NP-hard and even hard to approximate ([ref](https://en.wikipedia.org/wiki/Vertex_cover)). Therefore, it is hard to run experiments with Fig 3 setting on large-scale datasets such as ImageNet.

---

### Official Review · Reviewer_LX5M · 2022-10-28

**Confidence:** 3
**Correctness:** 3
**Technical Novelty And Significance:** 3
**Empirical Novelty And Significance:** 2
**Recommendation:** 6

**Clarity, Quality, Novelty And Reproducibility:**

Clarity: The paper is overall well written with good readability.

Quality: The theorem looks solid with experimental justifications. But I didn't check the proof thoroughly.

Novelty: The idea of alignment and uniformity has been studied in existing works. But the work focus on the effect of augmentation and has some novelty.

Reproducibility: n/a

**Strength And Weaknesses:**

### Strengths

(+) The studied problem of contrastive learning generalization is very interesting and significant.

(+) The theorems look solid and insightful. Examples and intuitive explanations are provided for a better understanding.

(+) The paper is well-organized and easy to follow.

### Weakness

(-) The empirical contribution may be limited since many results are very intuitive and have been (empirically) discovered by existing works. It would be better if some new augmentation approaches or practical guidance based on the theorems can be provided. E.g, are there any optimization approaches to generate augmentations that guarantee the highest concentration subject to that divergence and alignment are achieved?

### Suggestions and questions

- The authors briefly discuss existing theoretical works in contrastive learning. Although they are based on different groundings, I can see there may be some common conclusions or insights. The authors discuss the difference between their theory and the “alignment and uniformity”-based studies. It would be a great additional contribution if the author can include some discussions or analyses to align more works from different grounding, e.g., MI-based and expansion-based.

- Would it be possible to give a formal definition or description of the latent classes? Are they simply based on the semantics of data?

- Can the (\sigma, \delta)-augmentation cover all kinds of augmentations in practice, i.e., for some augmentations, there is no such a set and \sigma for some \delta? What would happen to the conclusion in this case, if exist?

**Summary Of The Paper:**

The paper studies and quantifies the generalization capability of contrastive learning approaches. Specifically, the paper derives the upper bound of downstream error by introducing the $(\sigma, \delta)$-augmentation. The theorems quantify the generalization capability regarding the alignment, divergence, and augmentation concentration.

**Summary Of The Review:**

I overall enjoyed reading this paper. The theoretical part is informative and solid. But there are some limitations in the empirical contribution and the practical guidance of the theorems.

---

> ### Author Response · Authors · 2022-11-16
> **Response to Reviewer LX5M (2/2)**
>
> **Q2.** It would be a great additional contribution if the author can include some discussions or analyses to align more works from different grounding, e.g., MI-based and expansion-based.
>
> **A2.** Thanks for the suggestion. We would like to give a detailed discussion of the MI-based and expansion-based works below, which has been added to the revised version.
>
> - MI-based: Since InfoNCE loss is a lower bound of mutual information (MI) [1,2,3], i.e.,
>      $$
>      I(X ; Y) \geq \mathbb{E}\left[\frac{1}{K} \sum_{i=1}^K \log \frac{e^{f\left(x_i, y_i\right)}}{\frac{1}{K} \sum_{j=1}^K e^{f\left(x_i, y_j\right)}}\right],
>      $$
>     some works explain the success of contrastive learning as it implicitly maximizes the mutual information between positive samples. While this is an intuitive interpretation, **a rigorous relationship between MI and the downstream classification error has not been established**. It is also **unclear how to understand the role of data augmentation** in this framework. Besides, [1] empirically showed that directly **maximizing a sharper bound of MI does not imply a better representation.** Therefore, MI may not fully explain the success of contrastive learning.
>
> - Expansion-based: [4] introduced the $(a,c)$-expansion framework to study unsupervised learning and self-training. This notion is used to characterize the continuity of latent classes, which can be considered as a **local quantification**. In contrast, our $(\sigma, \delta)$-augmentation is used to characterize the concentration of augmented data, which can be regarded as a **global quantification**. Besides, their goal is to propose some **new loss functions** with theoretical guarantees. Instead, our focus is to explain the success of **popularly used contrastive learning algorithms**.
>
> ---
>
> **Q3.** Would it be possible to give a formal definition or description of the latent classes? Are they simply based on the semantics of data?
>
> **A3.** The latent classes are directly determined by the **unobserved labels**. Our paper focuses on the i.i.d. setting, where training and downstream data have the same distributions (e.g., both self-supervised training and evaluation are on ImageNet). Therefore, the labels of training data exist but cannot be accessed during the training. They naturally define the so-called latent classes.
>
> ---
>
> **Q4.** Can the $(\sigma, \delta)$-augmentation cover all kinds of augmentations in practice, i.e., for some augmentations, there is no such a set and $\sigma$ for some $\delta$? What would happen to the conclusion in this case, if exist?
>
> **A4.**
> Firstly, we would like to remark that $(\sigma,\delta)$ is not unique. **One can always choose a large enough $\delta$ to make $\sigma>0$**.
> Secondly, some extremely "bad" augmentation does exist such that $\sigma=0$ for some $\delta$. An extreme example is the "identity" transformation: the augmentation only maps $x$ to itself, i.e., $A(x)=\{x\}$. If the domain of $x$ is discrete, for some sufficiently small $\delta$, the corresponding $\sigma$ will become $0$.
> Nevertheless, our theory **still applies by directly setting $C_k^0 = \varnothing$ and $\sigma=0$**. In this case, our classification bound is larger than $1$, which is vacuous. **This does not mean that our bound is not sharp.** In fact, in this case, the extremely "bad" data augmentation **can not bring any useful signal** resulting in bad concentration. Therefore, the performance of the learned encoder cannot be guaranteed. This still **agrees with our conclusion** that the sharper concentration of augmentation indicates better performance.
>
> ---
>
> **References:**
>
> [1] Tschannen, Michael, et al. "On mutual information maximization for representation learning." *arXiv preprint arXiv:1907.13625* (2019).
>
> [2] Poole, Ben, et al. "On variational bounds of mutual information." *International Conference on Machine Learning*. PMLR, 2019.
>
> [3] Oord, Aaron van den, Yazhe Li, and Oriol Vinyals. "Representation learning with contrastive predictive coding." *arXiv preprint arXiv:1807.03748* (2018).
>
> [4] Wei, Colin, et al. "Theoretical analysis of self-training with deep networks on unlabeled data." *arXiv preprint arXiv:2010.03622* (2020).

---

> ### Author Response · Authors · 2022-11-16
> **Response to Reviewer LX5M (1/2)**
>
> **Q1.** The empirical contribution may be limited since many results are very intuitive and have been (empirically) discovered by existing works. It would be better if some new augmentation approaches or practical guidance based on the theorems can be provided.
>
> **A1.** Thanks for your suggestion.
>
> First of all, we would like to remark that our empirical results **still provide some interesting insights** beyond the existing works (e.g., SimCLR paper):
>
> 1. The downstream performance monotonously increases with both the richness and strength of augmentations. The above observation can be directly explained by our theory that **richer** and **stronger** transformations lead to **sharper concentration** of augmentations (according to Def 1), hence the better downstream performance (according to Thm 1).
>
> 1. Color **dropping** and **distortion** have a great impact on the performance. **This is also predictable**, since our theory suggests that these two enable the augmented data to vary in a very wide range, making the augmented distance (Eq 1) largely decrease. Note that largely reduced augmented distance (Eq 1) indicates the **much sharper concentration** (Def 1), hence the much better downstream performance (according to Thm 1).
>
> 1. The above two empirical results are all **observed among different contrastive algorithms** (i.e., SimCLR, MoCo, Barlow Twins, SimSiam). This is because concentration of augmented data is pre-defined and can not be optimized by algorithms (Thm 1). Thus, for a given contrastive algorithm including the algorithms preventing collapse by training strategies instead of losses such as **SimSiam**, sharper concentration indicates better downstream performance.
>
>
> 1. Fig 3 shows that the concentration of augmented data is **highly correlated** to real downstream performance. This also provides a theoretical explanation for Figure 5 in SimCLR paper of **why the composition of "crop & color" performs the best**.
>
> Secondly, if we are allowed to access a few extra labeled data from the same data distribution, we can have a rough estimate of concentration for different kinds of augmentations over this data distribution. Therefore, our theory suggests choosing the (combinations of) augmentations with the sharpest concentration to do contrastive learning. **This could be an example of practical guidance based on the theorems**. A specific implementation of the above idea could be future work.
>
> Thirdly, as suggested by the reviewers, **we run additional experiments to empirically study the alignment and divergence factors**, which bring informative results. We choose the models with three different loss functions (i.e., Barlow Twins, SiamSiam and MoCo) and observe how alignment and divergence change during the training procedure when the setting of data augmentation is fixed (see Fig 4 of Appendix H in the revised pdf).
>
> We have the following interesting observations:
>
> - At the end of the training, both the alignment and divergence are ordered as MoCo < SimSiam < Barlow Twins, from small to large (i.e., good to bad). We also observe that MoCo, Simsiam, and Barlow Twins achieve a KNN accuracy of 90.33, 89.28, and 86.94, respectively. This suggests that **better alignment and divergence result in better performance** when the setting of data augmentation is fixed. This empirical result is as expected by our Thm1: as long as good alignment and divergence are achieved, **no matter** whether it is due to the loss functions (e.g., SimCLR and Barlow Twins, proved in Section 4) or other unknown reasons (e.g., SimSiam), the generalization error should be small according to our Thm 1.
>
> - During the training procedure, the **divergence factor always gets better** (i.e., smaller) for all three kinds of algorithms. It decreases more quickly at the early stage of training. Meanwhile, the **alignment factor starts to continuously get better after several training epochs** for MoCo and Barlow Twins. But for SimSiam, its alignment gets worse. This is because SimSiam does not directly minimize the alignment, instead, a predictor is involved to transform the feature of one view and matches it to the other view.

---

### Decision · Program_Chairs · 2023-01-20

**Decision:**

Accept: poster

**Justification For Why Not Higher Score:**

Some limitations are mentioned by reviewers and in meta-reviews. I won't mind if the paper is pushed up to be a spotlight paper, though.

**Justification For Why Not Lower Score:**

An interesting topic with solid theoretical results.

**Metareview: Summary, Strengths And Weaknesses:**

The paper provides theoretical understanding of the generalization ability of contrastive self-supervised learning, introduce the (\sigma, \delta)-augmentation measure and derives the upper bound of downstream classification error using KNN classifiers. The work shows that the generalization ability can be described by three key factors: alignment of positive samples, divergence of class centers, and concentration of augmented data. While alignment and divergence are optimized by contrastive learning algorithms, data augmentation determines concentration. Several learning algorithms including InfoNCE and cross-correlation are studied. The theoretical work is further supported and extended with experimental results. The paper is solid and well-written. It would be interesting to see further investigations, such as more extensive experiments, a larger set of contrastive learning algorithms, and tightness of the bound in practice. I agree with the reviewers to accept the paper.

**Note From Pc:**

if the above contains the word "oral" or "spotlight" please see: "oral" presentation means -> notable-top-5% and "spotlight" means -> notable-top-25%. As stated in our emails, we are disassociating presentation type from AC recommendations